J Physiol 603.19 (2025) pp 5827–5849

# Oxytocinergic signalling in the respiratory parafacial region increases the activity of chemosensitive neurons and respiratory output

Emmanuel V. Araujo[1,2], Phelipe E. Silva[1], Luiz M. Oliveira[3] , Yingtang Shi[4], Ana C. Takakura[5] , Daniel K. Mulkey[2] and Thiago S. Moreira[1]

[1]*Department of Physiology and Biophysics, Institute of Biomedical Science, University of São Paulo, São Paulo, São Paulo, Brazil*

[2]*Department of Physiology and Neurobiology, University of Connecticut, Storrs, Connecticut, USA*

[3]*Center for Integrative Brain Research, Seattle Children's Research Institute, Seattle, Washington, USA*

[4]*Department of Pharmacology, University of Virginia, Charlottesville, Virginia, USA*

[5]*Department of Pharmacology, Institute of Biomedical Science, University of São Paulo, São Paulo, São Paulo, Brazil*

Handling Editors: Harold Schultz & Daniel Zoccal

The peer review history is available in the Supporting Information section of this article (https://doi.org/10.1113/JP287845#support-information-section).

*The Journal of Physiology*

**Abstract figure legend** Oxytocinergic modulation of breathing in the retrotrapezoid nucleus (RTN). The paraventricular nucleus of the hypothalamus (PVN), which serves as the primary source of oxytocin (Oxt) in the brainstem, projects to the RTN. In vivo, pharmacological (TGOT) or optogenetic activation of oxytocinergic receptors and varicosities within the RTN increases breathing amplitude without affecting respiratory frequency. Selective activation of RTN by the oxytocin agonist Thr4, Gly7-oxytocin (TGOT) involves a mechanism partially dependent on KCNQ channels. Identifying the signaling pathways linking oxytocin receptor activation to changes in chemoreceptor activity may offer new therapeutic targets for treating respiratory central disorders.

The Journal of Physiology

**Abstract** The retrotrapezoid nucleus, located in the parafacial medullary region (RTN/pFRG), is crucial for respiratory activity and central chemoreception. Recent evidence suggests that neuro-modulation, including peptidergic signalling, can influence the $CO_2/H^+$ sensitivity of RTN neurons. The paraventricular nucleus of the hypothalamus (PVN) projects to the ventral medullary surface, including the RTN, and is considered the primary source of oxytocin to the brainstem. However the physiological significance of oxytocin signalling in RTN neurons has not been determined. To investigate this further we employed neuroanatomical techniques, slice-patch electrophysiology and *in vivo* pharmacological and optogenetic tools to characterize the effects of oxytocin on breathing. We found that a subset of PVN excitatory neurons (VGlut2-positive) that project to the RTN (Ctb-positive) are also immunoreactive for oxytocin, suggesting the RTN is a down-stream target of these neurons. Exogenous application of the selective oxytocin agonist (TGOT) activates RTN chemoreceptors in a dose-dependent manner ($EC_{50}$ = 3 nM), and this response is blunted by the blockade of KCNQ channels (ML252; 10 μM). In urethane-anaesthetized mice pharmacological (TGOT) or optogenetic activation of oxytocinergic receptors/varicosities in the RTN increases breathing amplitude without changing respiratory frequency. These results identify oxytocin signalling to RTN neurons as a novel regulator of respiratory activity and further demonstrate the importance of KCNQ channels in the modulation of RTN neurons.

(Received 10 October 2024; accepted after revision 18 August 2025; first published online 5 September 2025)

**Corresponding authors** T. S. Moreira: Department of Physiology and Biophysics, Institute of Biomedical Science, University of Sao Paulo, Sao Paulo 05508-000, Brazil. Email: tmoreira@icb.usp.br

D. K. Mulkey: Department of Physiology and Neurobiology Pharmacy Building, Rm216, University of Connecticut, 75 North Eagleville Road, Unit 3156, Storrs 06269-3156, CT, USA. Email: daniel.mulkey@uconn.edu

### Key points

- Oxytocin is an important modulator of breathing, including at the level of the retrotrapezoid nucleus (RTN).
- The paraventricular nucleus of the hypothalamus (PVN), considered as the primary source of oxytocin in the brainstem, projects to the RTN.
- Selective oxytocin agonist ($Thr^4, Gly^7$-oxytocin, TGOT) activates RTN by a mechanism partly dependent on KCNQ channels.
- *In vivo*, selective activation of oxytocinergic signalling within the RTN increases breathing amplitude without changing respiratory frequency.
- Identifying components of the signalling pathway that couples oxytocin receptor activation to changes in chemoreceptor activity may provide new potential therapeutic targets for treating central respiratory disorders.

## Introduction

Substantial evidence supports the involvement of the glutamatergic neurons of the retrotrapezoid nucleus within the parafacial medullary region (RTN/pFRG) in breathing regulation and central chemoreceptor control (Abbott et al., 2009; Gourine, 2005; Mulkey et al., 2004; Nattie, 2011; Souza et al., 2018; Stornetta et al., 2006;

**Emmanuel V. Araujo** earned his degree in nutrition from Federal University of Paraiba (UFPB), Brazil, in 2017. He is currently a PhD student in the Physiology Program at the Institute of Biomedical Sciences, University of São Paulo (ICB-USP), under the supervision of Dr. Thiago S. Moreira. His research focuses on investigating the mechanisms and extent of oxytocin signalling in modulating respiration at the ventral medullary surface.

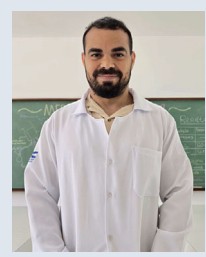

Takakura et al., 2006). RTN neurons are intrinsically activated by $CO_2/H^+$ through mechanisms involving the inhibition of TASK-2 channels and activation of GPR4. The RTN serves as an integration centre for other respiratory areas, as well as arousal and metabolic centres, including hypothalamic neurons (Fukushi et al., 2019; Gestreau et al., 2010; Guyenet et al., 2019; Kumar et al., 2015; Li et al., 2020; Moreira et al., 2025; Mulkey et al., 2004; Wang et al., 2013). Furthermore recent evidence indicates that RTN chemosensitivity can be shaped by neuromodulatory drive, including inputs from hypothalamic neurons (Fukushi et al., 2019; Moreira et al., 2025).

We are particularly interested in oxytocinergic neurons in the paraventricular nucleus of the hypothalamus (PVN), because oxytocin plays an important role in various physiological and behavioural processes, including social bonding, maternal behaviours, stress regulation, reproductive functions and cardiorespiratory coupling (Gimpl & Fahrenholz, 2001).

The PVN is a key hypothalamic structure involved in co-ordinating neuroendocrine and autonomic responses to a wide range of stimuli. Oxytocin produced in the PVN is released both into the bloodstream, where it acts as a hormone, and within the brain, where it functions as a neuromodulator (Grinevich & Ludwig, 2021). This dual role allows oxytocin to influence both peripheral and central systems, making it a pivotal molecule in mediating complex behaviours and physiological processes such as breathing.

Oxytocin modulation of breathing may involve its influence on the ventral respiratory column (VRC) and chemoreceptor neurons (Mack et al., 2002). To address the possibility that oxytocin signalling modulates RTN neuronal activity we assessed the neuroanatomical connections of oxytocin-expressing cells with projections to the RTN/pFRG region. We also used receptor pharmacological and optogenetic stimulation to examine whether oxytocin signalling modulated either RTN neuronal activity *in vitro* or breathing output *in vivo*. Furthermore at the cellular level we showed that chemosensitive RTN neurons are strongly activated by a selective oxytocin receptor agonist, and this response is reduced by blocking downstream KCNQ channels. These results identify key components of the signalling pathway that couples oxytocin receptor activation to changes in chemoreceptor excitability and suggest an important role for this pathway in regulating breathing activity.

# Methods

## Ethical approval

Experiments were performed on mice following procedures adhering to Brazil or USA National Institutes of Health Animal Care and Use Guidelines and are approved by the Animal Care and Use Committees of the University of São Paulo (Protocol # CEUA-ICB/USP: n° 9 750 170 720), the University of Connecticut (A22-049), the University of Virginia (Protocol no. 2454) and *The Journal of Physiology*'s policies regarding animal experiments. All efforts were made to minimize pain and suffering. The investigators acknowledge the ethical principles upheld by the journal and confirm that the present work complies with the animal ethics checklist.

## Animals

Mice were housed in ventilated racks and steam-sterilized caging (up to 5 per cage), with *ad libitum* access to food and water. Animals were maintained on a 12 h light/dark cycle in a vivarium maintained at 22–24°C and ∼40%–50% relative humidity. To characterize neuroanatomical projections we crossed VGlut2[Cre/+] mice (Slc17a6[tm2(cre)Lowl]/J (JAX # 01 6963) or VGat[Cre/+] mice (Slc32a1[tm2(cre)Lowl]/J (JAX # 01 6962) with a Cre-dependent reporter line (B6.Cg-Gt(ROSA)26Sortm6(CAG-ZsGreen1)Hze/J; Ai6 JAX # 0 07906). The offspring from these crosses are referred to as VGlut2[Cre/+]::Ai6 and VGat[Cre/+]::Ai6 mice, respectively.

To target RTN neurons for slice-patch recording we used wild-type mice on a C57BL/6J background (JAX # 000664) or offspring from a cross between Phox2b[Cre/+] (B6(Cg)-Tg(Phox2b-cre)3Jke/J; JAX # 01 6223) and Ai14 mice (B6;129S6-Gt(ROSA)26Sor[tm14(CAG-tdTomato)Hze]/J; JAX # 0 07908). For optogenetic experiments we crossed Oxt[Cre/+] mice (B6;129S-Oxt[tm1.1(cre)Dolsn]/J; JAX # 02 4234) with Ai32 mice (B6.Cg-Gt(ROSA)26Sor[tm32(CAG- COP4∗H134R/EYFP)Hze]/J; JAX # 02 4109) to express in the offspring the channelrhodopsin-2 (Ch2R)/eYFP fusion protein in a Cre-dependent manner (Madisen et al., 2012). To characterize oxytocin receptor expression in the RTN/pFRG we used a Phox2b::GFP BAC transgenic mouse line (Jx99, mixed background) developed by the GENSAT project and previously characterized (Lazarenko et al., 2009).

A total of 88 mice of both sexes were used: 56 neonates (P7–P12) for *in vitro* electrophysiological recording and 21 adults (8–15 weeks) for *in vivo* studies, including 11 for anatomical and physiological assessments (7 Ctb and 4 AAV).

## Pharmacological and optogenetic experiments *in vivo*

**Physiological preparation.** General anaesthesia was induced in mice with 5% isoflurane delivered in 100% $O_2$.

The isoflurane concentration was reduced to 1.4%–1.5% until the end of the surgical process. All mice (C57BL/6J or Oxt$^{Cre/+}$::Ai32) used for anaesthetized experiments were subjected to the following surgical procedures: (1) placed in dorsal supine position in the stereotaxic apparatus (model Kopf 1760); (2) removal of the occipital plate for the insertion of a pipette for drug injection or a fibre optic for laser stimulation directly into the RTN/pFRG region; (3) implantation of electrodes for external intercostal muscle (Int$_{EMG}$) recording. For Int$_{EMG}$, two thin Teflon-coated silver wires with bared tips forming a 2 mm hook were inserted through the lateral edge of the intercostal on the right side of the mice using 25 gauge. The electrode tips were inserted no more than 2–3 mm apart to minimize the electrocardiogram (EKG) artifact.

Upon completion of the surgical procedures isoflurane was replaced with urethane (1.4 g/kg, I.P.), administered slowly. The adequacy of anaesthesia was monitored during a 20 min stabilization period by testing for the absence of withdrawal responses, changes in Int$_{EMG}$ to a firm toe pinch. Approximately hourly supplements of one-third of the initial dose of urethane were administered as needed to maintain an adequate level of anaesthesia. Int$_{EMG}$ signal was digitized with a micro1401 (Cambridge Electronic Design), stored on a computer and processed offline using Spike2 software (version 7.2, Cambridge Electronic Design, Cambridge, UK). Integrated intercostal muscle activity ($\int$Int$_{EMG}$) was obtained after rectification and smoothing ($\tau = 0.015$ s) of the original signal, which was acquired with a 30–300 Hz bandpass filter. Int$_{EMG}$ amplitude and IntE$_{MG}$ frequency were evaluated before and after pharmacological administration and during photostimulation, and values were expressed as a percentage of baseline to allow for comparison across conditions.

## Photostimulation and pharmacological injection in the RTN/pFRG region

The light source was a diode pumped 473 nm blue laser (Thorlabs laser Model S1FC473MM; Newton, NJ, USA) controlled by a function generator (Grass Technologies/Astro-Med Inc., Warwick, RI, USA) to generate 10 ms light pulses. Stimulation trials generally consisted of 100 s trains for 10 ms pulses delivered at 10 Hz. Five photostimulation trials were conducted for each animal, and the data presented represent average values from these trials. Trials were included in the analysis if they met predefined criteria: a stable baseline breathing pattern before stimulation and a clear response during stimulation, minimizing variability due to factors like movement artifacts or irregular baseline states. The actual power output measured at the end of the fibre with a light metre (Thorlabs) was close to 9 mW. The same fibre optic was used for all experiments. Photostimulations were made using the following co-ordinates to target the RTN/pFRG of mice: 5.35 mm below the dorsal surface of the brain, 1.3 mm lateral to the midline and 1.4 mm caudal to lambda.

Injections of [Thr$^4$, Gly$^7$] oxytocin (TGOT) (1 µM) or saline into the RTN/pFRG region were performed using nitrogen pressure (8–15 ms pulses) via glass micropipettes (0.5 mm i.d., Sutter Instrument Co, CA, USA) coupled with a PicoSpritzer II pneumatic pump (General Valve Corporation, Fairfield, NJ, USA). The injection volume was maintained at approximately 30 nl, as indicated by graduated rule coupled with scope used to observe the surface of animal's head during the surgery. In addition TGOT or saline contained 1% fluorescent latex microspheres (Lumafluor, New York City, NY, USA) for subsequent histological analysis. Injections were made using the following co-ordinates to target the RTN/pFRG of mice: 5.5 mm below the dorsal surface of the brain, 1.3 mm lateral to the midline and 1.4 mm caudal to lambda.

After the completion of pharmacological or optogenetic stimulation, the depth of anaesthesia was carefully verified using urethane (1.4 g/kg, I.P.) to ensure that animals were adequately anaesthetized. The animals were then immediately perfusion-fixed for subsequent histological analysis, as described below.

## Tracer injections

To identify oxytocinergic projections to the RTN/pFRG we injected 1% of the cholera toxin B subunit (1% in 0.2 M phosphate buffer, pH 7.35; List Biological, Campbell, CA, USA) into the RTN/pFRG of VGlut2$^{Cre/+}$::Ai6 or VGat$^{Cre/+}$::Ai6 mice. To selectively trace oxytocinergic projections we injected a Cre-dependent virus (AAV5-EF1a-double-floxed-hChR2(H134R)-mCherry-WPRE-HGHpA, titre for injection: $1 \times 10^{13}$ vg/ml, Addgene plasmid 20297) into the PVN of the Oxt$^{Cre/+}$ mice. For both protocols mice were anaesthetized with ketamine (100 mg/kg, I.P.) and xylazine (10 mg/kg, I.P.) and fixed in a stereotaxic apparatus (Kopf model 1760). For tracer experiments 30 nl of Ctb was unilaterally injected into the RTN/pFRG region. For tracing and selectivity experiments two 50 nl injections of virus (AAV5-EF1a-double-floxed-hChR2(H134R)-mCherry-WPRE-HGHpA) were delivered unilaterally at two sites spanning the rostrocaudal length of the PVN region. All the injections were made using a glass micropipette with an internal tip diameter of around 20 µm. These surgeries were performed using standard aseptic techniques, and the injections in the RTN/pFRG and PVN regions were placed in the following co-ordinates: pFRG/RTN (1.4 mm caudal to lambda, 1.3 mm lateral to the midline and 5.5 mm below the brain surface and PVN): 0.3 mm

caudal to bregma, 0.2 mm lateral to the midline and 4.5 mm below the brain surface. The incisions on the mice's heads were closed and subsequently treated with ampicillin (100 mg/kg, I.M.) and ketorolac (0.6 mg/kg, S.C.). The animals were returned to the vivarium at the end of the surgical procedures. After 7–10 days of the Ctb injections or 4 weeks of the vector injections, the animals were deeply anaesthetized with ketamine and xylazine (100/10 mg/kg, I.P.) and immediately perfusion-fixed, as described below.

### Single-cell transcriptome of RTN neurons

In the present study we utilized a previously published single-cell RNA-sequencing dataset (GEO: GSE163155), with the transcriptome analysis performed according to the methodology described by Shi and colleagues (Shi et al., 2017). Briefly JX99 (Phox2b::GFP BAC transgenic) mice were anaesthetized with ketamine (375 mg/kg) and xylazine (25 mg/kg) via I.M. injection, followed by rapid decapitation. The brainstems were immediately extracted and sliced (300 μm thick). Individual RTN neurons were collected in a recording chamber mounted on a Zeiss Axio Imager FS fluorescence microscope using HEPES-based solution (containing 140 mM NaCl, 3 mM KCl, 2 mM MgCl$_2$, 2 mM CaCl$_2$, 10 mM HEPES and 10 mM glucose). eGFP-labelled cells from the RTN/pFRG region were aspirated into glass pipettes filled with sterile HEPES-based solutions. The contents were expelled into 0.2 ml RNase-free PCR tubes containing ice-cold lysis mix. RNA was reverse-transcribed, and cDNA was amplified using the SMART-Seq version 4 kit (Clontech, catalogue #634 896), following the manufacturer's instructions; 1 ng of cDNA was used to prepare dual-indexed libraries using the Nextera XT DNA Library Kit (Illumina, catalog #FC-131-1096; RRID: SCR_01 0233). Library quality was assessed using Agilent High Sensitivity D1000 TapeStation. Libraries (12–40 per batch) were pooled and sequenced on an Illumina NextSeq500 to generate 75 bp paired-end reads. FASTQ files were quasi-mapped to the mouse reference transcriptome (Ensembl GRCm38) using (Patro et al., 2017) protocols. Read counts per cell ranged from 3 to 14 million (50%–80% mapped). Transcript abundance was normalized to transcripts per million (TPM) according to Soneson et al. (2016).

### Histology, analysis and cell counts

Immunohistochemical procedures were performed as previously described (Miranda et al., 2023; Shi et al., 2021). Mice were first deeply anaesthetized with ketamine/xylazine (100/10 mg/kg, I.P.). After confirming the absence of the toe pinch reflex, 50 units of heparin were injected transcardially. The mice were then perfused via the ascending aorta and pulmonary artery with 50 ml of phosphate-buffered saline (PBS) (pH 7.4) followed by 100 ml of 4% formaldehyde (Electron Microscopy Sciences, Fort Washington, PA, USA) in 0.1 M phosphate buffer (PB, pH 7.4). The brains were then removed and stored in fixative for 24–48 h at 4°C. A series of coronal sections (1:4 series, 40 μm thick) were cut along the rostrocaudal axis using a microtome (SM2010R; Leica Biosystems, Buffalo Grove, IL, Vibratome) and stored at −20°C in cryoprotectant solution (20% glycerol, 30% ethylene glycol in 50 ml PB) for later histological processing. All histochemical procedures were performed using free-floating sections, in accordance with the previously described protocols (Miranda et al., 2023; Shi et al., 2021).

For the immunofluorescence technique oxytocin (Oxt) was detected using a polyclonal guinea pig anti-oxytocin antibody (BMA Biomedicals H0UWV4; 1:200 000). The m-cherry protein was detected using a polyclonal rabbit anti-DsRed antibody (Takakara 632496; 1:1000). Cholera toxin B (Ctb) was identified with a polyclonal goat anti-Ctb antibody (List Biological Laboratories; 1:10 000). Green fluorescent protein (GFP) was detected using a polyclonal chicken anti-GFP (Aves Lab, Inc; 1:800). For the *in vitro* experiments phox2b was detected using a polyclonal goat anti-mouse phox2b antibody (BRID:AB 10 889 846; 1:100) and rabbit polyclonal anti-lucifer yellow antibody (BRID:AB 2 536 190; 1:500). Each sample was diluted in PBS containing normal donkey serum (017-000-121; 1%; Jackson Immuno Research Laboratories) and 0.3% triton and then incubated for 24 h. The sections were subsequently washed in PBS and incubated for 2 h with donkey anti-guinea pig Alexa 488 or Alexa 594 (706-545-148; Jackson Immuno Research Laboratories; dilution 1:400) and donkey anti-rabbit Alexa 594 (711-585-152; Jackson Immuno Research Laboratories; dilution 1:500) for the anterograde tracer experiments. For the retrograde tracer experiments donkey anti-goat Alexa 594 (705-586-147; Jackson Immuno Research Laboratories; dilution 1:500) was used. For the GFP expression donkey anti-chicken Alexa 488 (703-545-155; Jackson Immuno Research Laboratories; dilution 1:400) was used. *In vitro* experiments used donkey anti-goat Alexa 594 (705-586-147; Jackson Immuno Research Laboratories; dilution 1:500) and donkey anti-rabbit Alexa 488 (711-545-152; Jackson Immuno Research Laboratories; dilution 1:500).

The brain sections were mounted onto slides, dried and covered with DPX (Aldrich, Milwaukee, WI, USA), and coverslips were affixed with nail polish. Brain sections were analysed under blind conditions using StereoInvestigator software (MBF Bioscience) with a confocal fluorescence microscope with High Content Imaging in Cell Analyzer 2200 GE (Zeiss LSM 780-NLO, Oberkochen, Germany). Images were acquired using a

Hamamatsu C11440 Orca-Flash 4.0LT digital camera (resolution: 2048 × 2048 pixels), resulting in TIFF files. A technical illustration software package (Adobe Illustrator, version 28.1, USA) was used for line drawings, figure assembly and labelling, following the guidelines of Franklin and Paxinos (2019). Representative images that were pseudo-coloured and optimized for presentation, brightness and contrast were adjusted equally in all pixels of the image. Images were manually cropped to the relevant anatomical region, a Gaussian blur was applied, the default auto-thresholding was applied (Ridler & Calvard, 1978), despeckled and then the default particle analysis tool was used to collect signal intensities. Axonal varicosities were drawn using the StereoInvestigator software. The files were exported into the Canvas drawing program (version 9, Deneba Systems Inc., Miami, FL, USA) for text labelling and final presentation.

The sections were counted uni- or bilaterally (depending on the experiment), and the numbers reported in the results section correspond exactly to the counts of one-in-four sections in a series. Section alignment between brains was completed relative to a reference section, as previously described (Oliveira et al., 2024; Souza et al., 2023; Wang et al., 2019). Briefly to align sections around the RTN/pFRG level the most caudal section that contained an identifiable cluster of facial motor neurons was identified in each brain and assigned to the level of 6.48 mm caudal to the bregma (bregma = −6.48 mm). To align sections around the PVN level the most caudal section was identified in each brain and assigned to the level 1.22 mm caudal to the bregma (bregma = −1.22 mm). Rostral or caudal levels to this reference section were determined by adding or subtracting the number of intervening sections (40 μm intervals). The analysis was performed as follows: (1) RTN/pFRG: four sections rostral from the caudal end of the facial nucleus (bregma: −6.00 to −6.48 mm); (2) PVN: five sections rostral from the caudal end of the most caudal PVN section (bregma: −0.58 to −1.22 mm). For the RTN/pFRG level the mapping was limited to the ventral half of the brainstem, which contains the distinctive and isolated parafacial cluster of $Phox2b^+$-expressing neurons.

### Electrophysiology in brainstem slices

Slices were prepared from the brainstem of neonatal ($N = 56$; P7-P12 mixed sexes) C57BL/6J background (Jax Stock #000664) or $Phox2b^{Cre/+}$::Ai14 mice. Briefly pups were anaesthetized with ketamine and xylazine (375 mg/kg and 25 mg/kg, I.P.). After verifying the lack of response to a firm toe pinch, pups were rapidly decapitated, and medullary slices (250 μm) were cut in ice-cold sucrose-substituted solution, containing (mM): 260 sucrose, 3 KCl, 5 $MgCl_2$, 1 $CaCl_2$, 1.25 $NaH_2PO_4$, 26 $NaHCO_3$, 10 d-glucose and 1 kynurenic acid. Slices were incubated for 30 min at 37°C and subsequently at room temperature in normal Ringer's solution containing (mM): 130 NaCl, 3 KCl, 2 $MgCl_2$, 2 $CaCl_2$, 1.25 $NaH_2PO_4$, 26 $NaHCO_3$ and 10 d-glucose. Cutting, incubation and recording solutions were bubbled with 95% $O_2$ and 5% $CO_2$ (pH = 7.3).

Individual slices containing the RTN/pFRG were transferred to a recording chamber mounted on a fixed-stage microscope (Olympus BX5.1WI) and perfused continuously (~2 ml/min) with normal Ringer's solution. Recordings (1 cell/mouse) were made at room temperature or ~22°C, as indicated, with an Axopatch 200B patch-clamp amplifier, digitized with a Digidata 1322A A/D converter and recorded using pCLAMP version 11.0.3. software (Molecular Devices, Sunnyvale, CA, USA). Cellular excitability was measured in the cell-attached (seal resistance >1 GΩ) voltage clamp mode using a pipette solution containing (in mM): 120 $KCH_3SO_3$, 4 NaCl, 1 $MgCl_2$, 0.5 $CaCl_2$, 10 HEPES, 10 EGTA, 3 Mg-ATP, 0.3 GTP-Tris and 0.2% Lucifer yellow (pH 7.30). To test for $CO_2/H^+$ sensitivity the gas mixture was switched to one containing 10% $CO_2$ (balance $O_2$; pH 7.0) for at least 5 min or when a plateau of firing activity was achieved for at least 2 min. [$Thr^4,Gly^7$] oxytocin (TGOT) (0.25–128 nM) was bath-applied at different concentrations. Where indicated slices were incubated in a cocktail of synaptic blockers consisting of CNQX (10 μM, Sigma-Aldrich) to block AMPA/kainate receptors, strychnine (2 μM, Sigma-Aldric) to block glycine receptors, gabazine (10 μM, Sigma-Aldric) and picrotoxin (50 μM, Sigma-Aldric) to block GABA-A receptors (Almado et al., 2012; Gonye et al., 2024; Wenker et al., 2012). In some experiments we also used a KCNQ channel blocker (ML252 – 10 μM, Sigma-Aldric) (Kanyo et al., 2023; Soto-Perez et al., 2023). Exposure to ML252 consistently increased neural activity; therefore, to minimize potential confounding effects of activity saturation (ceiling effect) during assessment of TGOT sensitivity in ML252, a negative DC current was used to return firing rates near control levels before testing TGOT sensitivity. To test for involvement of the desensitization of oxytocin receptors in RTN chemosensitive neurons the firing response of RTN neurons to TGOT before and following bath application of a selective G protein–coupled receptor kinase 2 (GRK2) inhibitor (CCG258208 – 500 nM) (Bouley et al., 2020). During exposure to CCG258208 a DC current was also applied to restore the firing rate to near-control levels prior to testing TGOT sensitivity. In addition in the whole-cell configuration we filled cell types of interest with Lucifer Yellow for *post hoc* immunohistochemical identification using the Phox2b marker.

## Statistics

All data used in inferential statistical tests were assessed for normality using the Shapiro-Wilk test (based on sample size) and confirmed to follow a normal distribution. For normally distributed data Student's *t* test or two-way repeated-measures ANOVA, followed by Bonferroni's multiple comparisons test, was used. Paired *t* tests were used for within-subject comparison and unpaired *t* tests for between-group comparisons. Unpaired *t* test was used to examine the following: (1) differences in breathing activity, expressed as percentage change from baseline, following saline or TGOT injection, as well as after optogenetic stimulation of oxytocin terminals, in the RTN/pFRG region; (2) RTN neuron firing rate response to 10% $CO_2$ or TGOT bath application in $CO_2$-sensitive and $CO_2$-insensitive neurons located in the RTN/pFRG region; (3) RTN activity of bath application of TGOT before and after GRK2 blocker. Paired *t* test was used to examine the following: (1) TGOT-induced change in firing rate of RTN under control condition, and in the presence of synaptic blocker cocktail; (2) RTN neuron firing rate during the first and second bath applications of TGOT (200 nM for 2 min or 3 min); (3) TGOT-induced change in firing rate of RTN under control condition, and in the presence of KCNQ blocker (ML252). Note that we typically get one slice containing the region of interest (RTN/pFRG) per mouse pup and only record from one cell per slice. Therefore the number of cells equals the number of animals used for each dataset included in this study. The F- and *P*-values for every effect and interaction between effects are reported in the figure legends. Results are shown as mean ± SD and presented in box and whiskers format (the box bisected by the median and bounded by the 25th and 75th percentiles, with whiskers indicating the range). Statistical analyses were performed using GraphPad Prism (version 9).

## Results

### Oxytocinergic paraventricular nucleus pathway to the ventral respiratory parafacial region

Oxytocinergic (Oxt) neurons are predominantly located in the PVN and supraoptic nucleus (Swanson & Sawchenko, 1980). To confirm that Oxt neurons of the PVN project to the ventral respiratory parafacial region near RTN neurons the Cre-dependent AAV-EF1a-double-floxed-hChR2(H134R)-mCherry-WPRE-HGHpA was injected into the PVN in five $Oxt^{Cre/+}$ mice. Four out of five AAV-double-floxed-hChR2-mCherry injections were correctly placed into the PVN of $Oxt^{Cre/+}$ mice (Fig. 1*A* and *B*). Based on a systematic inspection of four mice in which a one-in-four series of 40 μm sections (5 sections/mouse; bregma level: −0.58 to −1.22 mm) were reacted for immunohistochemical detection of Oxt and mCherry, the majority of the Oxt neurons express ChR2 (79 ± 2.5%; 141.2 ± 3.2 $mCherry^+$/$Oxt^+$ *vs.* 178.5 ± 3.4 $Oxt^+$; $N = 4$) (Fig. 1*B–D*). More specifically the majority of oxytocinergic neurons transfected ($mCherry^+$/$Oxt^+$) were located into the lateral magnocellular aspect of the PVN (PaLM – bregma level: −0.74 to −1.06 mm) (Fig. 1*C* and *D*). We found only 9.76 ± 1.2 $mCherry^+$/$Oxt^-$-expressing neurons throughout the PVN region (Fig. 1*D*). Brightly labelled axons and varicosities (terminal-like endings) were observed in regions that contain the medullary dorsal and ventral respiratory column (Fig. 1*E–J*). Labelled fibres from PVN Oxt neurons were also detectable in most subregions of the VRC, with the heaviest labelling located ventral to the nucleus ambiguous, a region corresponding to the pre-Bötzinger complex and rostral ventral respiratory group (Baertsch et al., 2019; Picardo et al., 2013) (Fig. 1*G*, *Gi* and *H*). Labelled fibres are also evident in the dorsal respiratory group, particularly the nucleus of the solitary tract (Fig. 1*E*, *Ei* and *F*). PVN Oxt neurons also send projections to the medial and lateral aspects of the ventral respiratory parafacial region (Fig. 1*I*, *Ii* and *J*). Although we did not perform a quantitative analysis, we qualitatively demonstrated the efferent projections of PVN Oxt neurons.

Next to complement our anterograde tracing results we performed retrograde tracing experiments where cholera toxin b (Ctb – 1%) was injected into the RTN/pFRG region of $VGlut2^{Cre/+}$::Ai6 or $VGat^{Cre/+}$::Ai6 reporter mouse lines along with Oxt immunohistochemistry to identify neurons (Fig. 2*A* and *B*). The Ctb injections ($N = 7$) were centred 50–150 μm dorsal to the ventral medullary surface, 100–300 μm rostral to the caudal end of the facial motor nucleus and 1.3–1.4 mm lateral to midline. Our analysis focused on the PVN region (bregma level: −0.58 to −1.22 mm; 1 series of 40 μm-thick coronal sections:160 μm apart) (Fig. 2*C* and *D*). As illustrated in Fig. 2 we found that a portion of Ctb-labelled excitatory neurons (VGlut2) were also immunoreactive for Oxt ($Ctb^+$/$VGlut2^+$/$Oxt^+$: 16.5 ± 5.6, *vs.* $Ctb^+$/$VGlut2^+$: 39.3 ± 18.8 neurons), suggesting that the RTN is one of several downstream targets of Oxt neurons (Fig. 2*C*, 2*Ci–Civ* and 2*E*). Some Ctb-labelled PVN neurons express Oxt ($Ctb^+$/$Oxt^+$: 20.8 ± 5.4 *vs.* $Ctb^+$: 55.3 ± 23.7 neurons) (Fig. 2*E* and *F*). Few back-labelled neurons showed VGat signal ($Ctb^+$/$VGat^+$: 3.7 ± 1.5 *vs.* $Ctb^+$: 51.7 ± 19.6 neurons) (Fig. 2*D*, *Di–Div* and *F*). In addition only a few of the Oxt neurons express VGat ($VGat^+$/$Oxt^+$: 9.7 ± 7.7 *vs.* $Oxt^+$: 145 ± 11.8 neurons) (Fig. 2*D* and *F*). We also found the majority of PVN projections to the ventral parafacial region are ipsilateral, whereas contralateral labelling was extremely rare (data not shown).

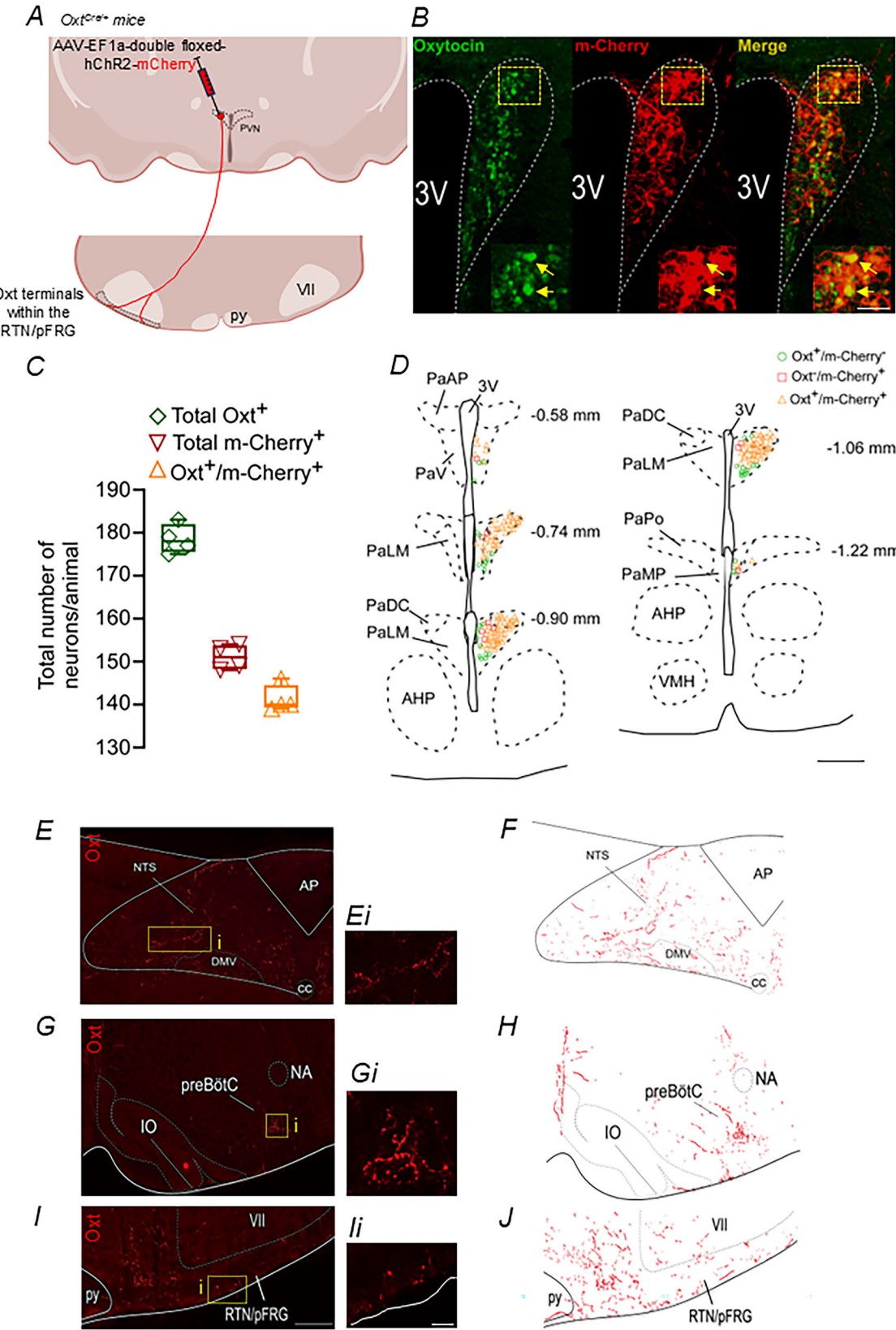

**Figure 1. Oxytocinergic PVN pathway to RTN/pFRG**
*A*, experimental approach for labelling the efferent projections from PVN oxytocinergic neurons to the RTN/pFRG in a coronal view. *B*, photomicrography of ipsilateral injection of AAV-EF1a-double-floxed-hChR2(H134R)-mCherry-WPRE-HGHpA into the PVN of OxtCre/+ mice. *C*, cell counts

were performed on 1:4 sections at the level of the PVN region to quantify the total number of oxytocin-positive (Alexa 488, green; diamond symbol), total mCherry-positive (Alexa 594, red, inverted triangle) and double-labelled (Oxytocin$^+$/mCherry$^+$, orange, triangle) cells in Oxt$^{Cre/+}$ mice ($N$ = 4 mice). D, computer-assisted plot of oxytocin and m-Cherry marker combinations detected in PVN neurons (bregma levels from −0.58 to −1.22 mm). E, image of projections of PVN oxytocinergic neurons to NTS (bregma = −7.32 mm). Ei, inset. F, axonal drawing showing the PVN projection to the NTS region. G, image of projections of PVN oxytocinergic neurons to preBötC (bregma = −7.08 mm). Gi, inset. H, axonal drawing showing the PVN projection to the preBötC region. I, image of projections of PVN oxytocinergic neurons to RTN/pFRG region (bregma = −6.36 mm). Ii, inset. J, axonal drawing showing the PVN projection to the RTN/pFRG region. AHP, posterior aspect of the anterior hypothalamic area; AP, area postrema; DMV, dorsal motor nucleus of the vagus; NA, nucleus ambiguous; IO, inferior olive; PaAP, anterior parvicellular aspect of the paraventricular nucleus of the hypothalamus; PaDC, dorsal cap aspect of the paraventricular nucleus of the hypothalamus; PaLM, lateral magnocellular aspect of the paraventricular nucleus of the hypothalamus; PaMP, medial parvicellular aspect of the paraventricular nucleus of the hypothalamus; PaPo, posterior aspect of the paraventricular nucleus of the hypothalamus; PaV, ventral aspect of the paraventricular nucleus of the hypothalamus; PVN, paraventricular nucleus of the hypothalamus; py, pyramid tract; RTN/pFRG, retrotrapezoid nucleus/parafacial respiratory group; Sp5, spinal trigeminal nucleus; VMH, ventromedial hypothalamic nucleus; VII, facial motor nucleus; XII, hypoglossal motor nucleus; 3V, third cerebral ventricle. Scale bar = 20 μm for panel B higher magnification, 500 μm for panel D, 100 μm for panel I (applied to E–J), 10 μm for panel Ii (applied to Ei, Gi and Ii.

## Oxytocinergic signalling in the RTN/pFRG increases respiratory activity

To determine whether oxytocin receptors are expressed by RTN neurons we analysed a previously published single-cell RNA-sequencing dataset (Shi et al., 2017; GEO: GSE163155). We found that Oxt transcripts were detectable in 95% of RTN neurons at levels nearly identical to those of the substance P receptor (Tacr1), another neuropeptide receptor known to be expressed in these cells (Fig. 3A). In comparison both Oxtr and Tacr1 were expressed at lower levels than the 5-HT2C receptor (Htr2c), which largely mediates serotonin effects on RTN neurons, and at much lower levels than Gpr4, the most highly expressed G protein–coupled receptor (GPCR) and a key contributor to the intrinsic $CO_2/H^+$ sensitivity of RTN neurons (Fig. 3A) (Cleary et al., 2021; Kumar et al., 2015).

To investigate whether oxytocinergic signalling to the ventral medullary surface influences breathing activity we conducted *in vivo* experiments informed by the presence of oxytocin receptor transcripts (Fig. 3A) and the neuroanatomical organization of oxytocinergic projections (Figs 1 and 2). Using two distinct approaches we aimed to assess the functional contribution of this signalling pathway to respiratory modulation. First we evaluated the effect of unilateral injection of oxytocin agonist [Thr$^4$,Gly$^7$] oxytocin (TGOT – 1 μM–30 nl), an analogue of oxytocin highly selective for the rodent oxytocin receptors (Fig. 3B–G). Second we used selective optogenetic stimulation (473 mm – pulses of 10 ms/10 Hz) of the RTN/pFRG region in Oxt$^{Cre/+}$::Ai32 mice on baseline breathing in urethane-anaesthetized mice (Fig. 3H–N). We found that the injection of TGOT into the RTN/pFRG ($N$ = 5 out of 11 injections were correctly placed) elicited an increase in Int$_{EMG}$ amplitude (16.7 ± 9.5 vs. saline: 4.4 ± 2.5%, $t$ = 2.8; d$F$ = 8; $P$ = 0.0229), without changing Int$_{EMG}$ frequency (3.5 ± 13.4 vs. saline:

−3.7 ± 6.6%, $t$ = 1; d$F$ = 8; $P$ = 0.311) (Fig. 3B–F). TGOT also increased the amplitude-frequency product (20 ± 9.7 vs. saline: 0.38 ± 5.3%, $t$ = 3.98; d$F$ = 8; $P$ = 0.0041) (Fig. 3G).

In urethane-anaesthetized Oxt$^{Cre/+}$::Ai32 mice, photo-stimulation (10 Hz) of the RTN/pFRG region produced a significant increase in Int$_{EMG}$ amplitude (18.8 ± 8.3 vs. Oxt$^{−/−}$::Ai32: 6.4 ± 4.2%, $t$ = 3.01; d$F$ = 9; $P$ = 0.0147), and Int$_{EMG}$ ampl × freq (22.7 ± 5.8 vs. Oxt$^{−/−}$::Ai32: 7.9 ± 9.3%, $t$ = 3.22; d$F$ = 9; $P$ = 0.0104), without changing Int$_{EMG}$ frequency (3.6 ± 6.9% vs. Oxt$^{−/−}$::Ai32: 1.3 ± 7.5%, $t$ = 0.52; d$F$ = 9; $P$ = 0.6094) (Fig. 3K–N). The effects of photostimulation on breathing were highly reproducible within each animal, and we did not observe a significant rundown of the effect over the course of an experiment. Photostimulation in control mice (Oxt$^{−/−}$::Ai32) had no effect on breathing ($t$ = 2.138; d$F$ = 4; $P$ = 0.099) (Fig. 3K–N).

Given that Cre-lox recombination does not always ensure exclusive expression in the target population, we performed immunohistochemistry to assess the selectivity of ChR2 expression in Oxt$^{Cre/+}$::Ai32 mice. We observed colocalization of ChR2-expressing fibres (GFP fluorescence) with oxytocin (Oxt) immunoreactivity in the RTN/pFRG region. Representative confocal image is shown in Fig. 3J, demonstrating the presence of ChR2-labelled axons colocalized with Oxt immuno-reactivity in the target area.

## Oxytocinergic signalling increases activity of RTN chemosensitive neurons

Cell-attached recordings were used to identify RTN neurons in acute brainstem slices from C57BL/6J or Phox2b reporter mice (Phox2b$^{Cre/+}$::Ai14) by their firing response to $CO_2/H^+$; RTN neurons were spontaneously active under control conditions (1.6 ± 0.5 Hz, 5% $CO_2$)

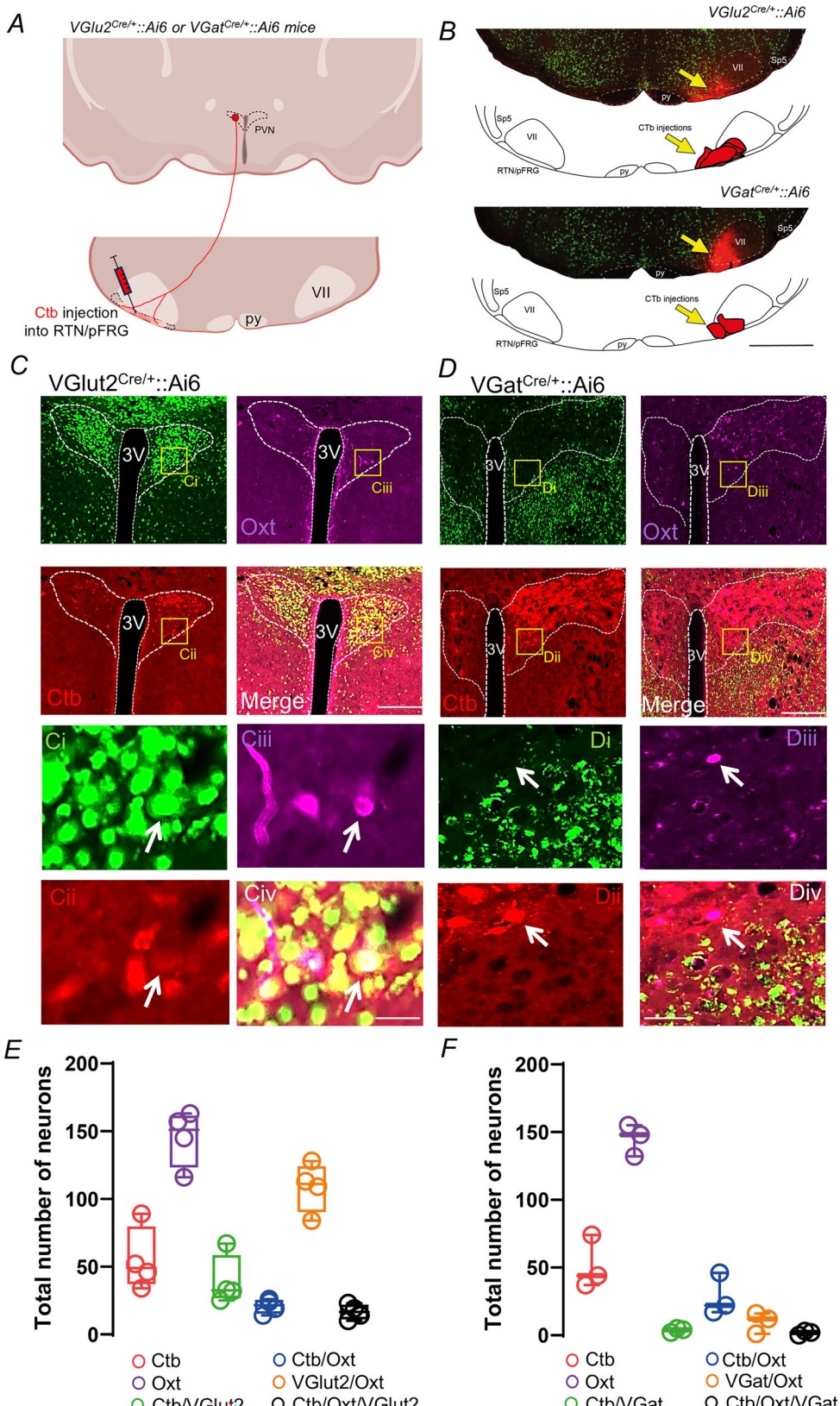

**Figure 2. Oxytocinergic and glutamatergic PVN inputs to the RTN/pFRG region**
*A*, experimental approach to retrogradely label PVN glutamatergic and GABAergic neurons using cholera toxin B (Ctb 1%) in a coronal view. *B*, photomicrography showing the location of an individual Ctb injection into

the RTN/pFRG region using a VGlut2$^{Cre/+}$::Ai6 (*N* = 4 mice) or VGat$^{Cre/+}$::Ai6 (*N* = 3 mice) reporter mouse. Below the photomicrographs are the corresponding computer-assisted plot showing the locations of all Ctb injections in VGlut2$^{Cre/+}$::Ai6 (*N* = 4 mice) or VGat$^{Cre/+}$::Ai6 (*N* = 3 mice) reporter mice. C, Ctb injection into the RTN/pFRG-labelled neurons (red) that are immunoreactive for VGlut2 (green) and oxytocin (magenta) in the PVN region. *Ci–Civ*, represents a higher magnification of the yellow square depicted in C. D, Ctb injection into the RTN/pFRG-labelled neurons in the PVN that are immunoreactive for oxytocin (magenta) but not for VGat (green). *Di–Div*, represents a higher magnification of the yellow square depicted in D. E, total number of PVN neurons (*N* = 4 mice) detected in five sections per brain (Ctb, all Ctb-ir neurons irrespective of other markers; Oxt, all Oxt-ir neurons irrespective of other markers). F, total number of PVN neurons (*N* = 3 mice) detected in five sections per brain (Ctb, all Ctb-ir neurons irrespective of other markers; Oxt, all Oxt-ir neurons irrespective of other markers). PVN, paraventricular nucleus of the hypothalamus; py, pyramid tract; RTN/pFRG, retrotrapezoid nucleus/parafacial respiratory group; Sp5, spinal trigeminal nucleus; VII, facial motor nucleus, 3V, third cerebral ventricle. Scale bar = 1 mm in *B*; 200 μm for panel *D* applied to *C*; 20 μm for panel *Div* applied to *Ci–Civ* and *Di–Div*.

and robustly activated by 10% $CO_2$ (3.5 ± 0.4 Hz, *N* = 16) (Fig. 4*C*, *E* and *G*). This level of $CO_2/H^+$-sensitivity is similar to what has been previously reported for RTN neurons (Shi et al., 2022; Sobrinho et al., 2014, 2016, 2017, 2022; Wenker et al., 2012). Furthermore, in most cases, recorded neurons were confirmed to be Phox2b-immunoreactive, as expected for RTN neurons (Fig. 4*A* and *B*). Neurons that showed <1 Hz firing response to 10% $CO_2$ were considered $CO_2$-insensitive (non-RTN neurons) (Fig. 4*E*).

To determine whether and how oxytocin signalling modulates the activity of RTN neurons, we characterized effects of TGOT on activity of RTN chemoreceptors under control conditions and during synaptic blockade. Under control conditions, we found that bath application of TGOT increased the activity of RTN neurons in a dose-dependent manner with an EC$_{50}$ of 3 nM (Fig. 4*C* and *D*). For subsequent experiments, we used a dose of 2 nM (1 min exposure) (3.9 ± 1.5 *vs.* baseline: 1.8 ± 1.5 Hz; t = 12.9; d*F* = 20, *P* < 0.0001) because it elicited a robust, reversible and repeatable firing response (Fig. 4*G*). These factors helped to ensure consistent magnitude of each TGOT response under varying conditions within the same experiment, thereby preventing desensitization, as illustrated in Figs 4*C* and 5*A*. We did not find any sex-based differences in the TGOT response (male: Δ = 1.8 ± 0.5 *vs.* female: Δ = 2.3 ± 0.9 Hz; t = 1.51; d*F* = 19; *P* = 0.1473) (data not shown). The observation that neurons unresponsive to $CO_2$ – when compared to their own baseline activity (t = 2.381; d*F* = 4; *P* = 0.076) (see methods for more detail) – also failed to respond to TGOT relative to baseline (t = 3.13; d*F* = 3; *P* = 0.052) supports the possibility that oxytocin activates only $CO_2$-sensitive neurons in the RTN/pFRG region (Fig. 4*E* and *F*).

To test whether Oxt activation of RTN chemoreceptors is dependent on synaptic input we exposed neurons to repeated bouts of TGOT in the presence of a cocktail of neurotransmitter receptor blockers. The blocker cocktail consisted of CNQX (10 μM) to block AMPA/kainate receptors, gabazine (10 μM) and picrotoxin (50 μM) to block GABA receptors and strychnine (2 μM) to block glycine receptors. As noted above under control conditions bath application of TGOT (2 nM) increased the activity of RTN neurons by 1.8 ± 0.5 Hz. After TGOT was washed out, exposure to the blocker cocktail (∼7 min incubation) had negligible effects on baseline activity (3.4 ± 0.6 *vs.* baseline: 3.9 ± 0.4 Hz; t = 1.189, d*F* = 6, *P* = 0.279) (Fig. 4*G* and *H*). In the continuous presence of the blocker cocktail a second exposure to TGOT (2 nM) increased firing rate by an amount similar to control (Δ = 2.0 ± 0.9 *vs.* TGOT without synaptic blockade: Δ = 1.8 ± 0.5 Hz; t = 0.607; d*F* = 6; *P* = 0.56) (Fig. 4*G* and *H*). These results are consistent with the possibility that chemosensitive RTN neurons respond directly to oxytocin.

## Oxytocin agonist-induced desensitization of oxytocin receptors in the RTN/pFRG

Oxytocin receptors, members of the GPCR family, are prone to desensitization (Rajagopal & Shenoy, 2018; Smith et al., 2006). Although RTN neurons exhibit a reversible and repeatable excitatory response to low concentrations of TGOT (0.25–16 nM) with brief exposure (1 min) (Fig. 4*C* and *D*), higher doses of TGOT (>16 nM) did not produce a proportional increase in activity, suggesting desensitization of oxytocin receptors. Therefore we chose to expose RTN neurons to two identical applications of TGOT, using the same concentration and exposure duration. Specifically we tested the response of RTN neurons to two repeated exposures of TGOT (200 nM, 2 min duration), with a 20 min interval between exposures. RTN neurons exhibited similar peak firing responses to both applications (TGOT change in activity first exposure: 3.6 ± 1.4 *vs.* second exposure: 3.0 ± 1.4 Hz; t = 2.066; d*F* = 7; *P* = 0.077) (Fig. 5*C*). Next we tested the same TGOT concentration with a longer exposure duration of 3 min. Under these conditions we observed a reduction in the firing response to the second TGOT application (Δ = 3.4 ± 1.2 Hz) compared to the first

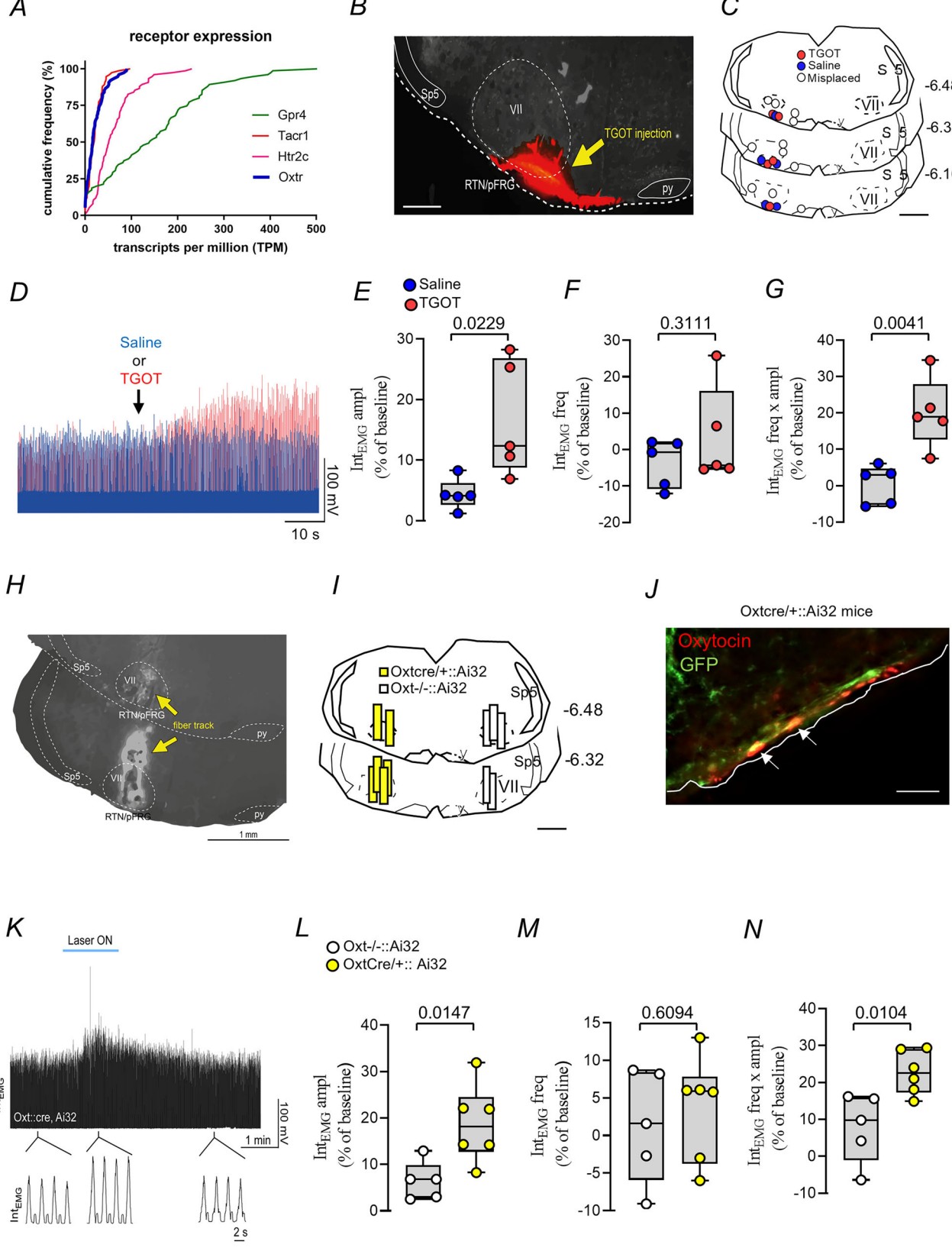

**Figure 3. Selective activation of oxytocinergic signalling in the RTN/pFRG increases respiratory activity**
*A*, cumulative probability distributions of the indicated transcripts (Oxtr, Tacr1, Htr2c, Gpr4) in GFP-positive neurons harvested from the RTN/pFRG of Phox2b::GFP mice, as determined by single-cell RNA-Seq (*n* = 75

cells, $N = 22$ mice; data from Shi et al. (2017); GEO: GSE163155). *B*, photomicrography showing the injection site of the oxytocin agonist [Thr$^4$, Gly$^7$] oxytocin (TGOT – 1 µM – 30 nl) into the RTN/pFRG region. *C*, computer-assisted drawing of the centre of the TGOT injection site revealed by the presence of fluorescent micro-beads included in the injectate. *D*, representative recordings of external intercostal electromyography (Int$_{EMG}$) from a urethane-anaesthetized mouse under baseline and after the injection of saline or TGOT in the RTN/pFRG region. *E–G*, group data ($N = 5$ mice for the saline group and $N = 5$ mice for the TGOT group) showing the percentage change from baseline in Int$_{EMG}$ activity of (*E*) amplitude (Int$_{EMG}$ ampl, unpaired $t$ test, $P = 0.0229$), (*F*) frequency (Int$_{EMG}$ fr, unpaired $t$ test, $P = 0.3111$) and (*G*) the product of amplitude and frequency (Int$_{EMG}$ fr × ampl, unpaired $t$ test, $P = 0.0041$). *H*, photomicrograph showing typical unilateral fibre-optic tracts in the RTN/pFRG region. *I*, computer-assisted drawing of the location of the fibre optics in the RTN/pFRG region. *J*, image of the expression of channel rhodopsin ChR2 (GFP immunohistochemistry, green) in oxytocinergic varicosities (Alexa 647, red) (bregma = −6.32 mm). *K*, representative recordings of external Int$_{EMG}$ from an anaesthetized Oxt$^{Cre/+}$::Ai32 mouse under baseline and after photostimulation (10 Hz, 10 ms light pulse) of the oxytocin-expressing varicosities in the RTN/pFRG region. *L* and *N*, group data ($N = 5$ mice for the Oxt$^{−/−}$ group and $N = 6$ for the Oxt$^{Cre/+}$ group) showing the percentage change from baseline in Int$_{EMG}$ activity of (*L*) amplitude (Int$_{EMG}$ ampl, unpaired $t$ test, $P = 0.0147$), (*M*) frequency (Int$_{EMG}$ fr, unpaired $t$ test, $P = 0.6094$) and (*N*) the product of amplitude and frequency (Int$_{EMG}$ fr × ampl, unpaired $t$ test, $P = 0.0104$) in Oxt$^{−/−}$::Ai32 and Oxt$^{Cre/+}$::Ai32. py, pyramid tract; RTN/pFRG, retrotrapezoid nucleus/parafacial respiratory group; Sp5, spinal trigeminal nucleus; VII, facial motor nucleus, 3V. Scale bar = 100 µm in *B* and *J*; 1 mm in *C*, *H* and *I*; 20 µm for panel *Ji*.

($\Delta = 5.0 \pm 1.8$) ($t = 5.039$; d$F = 8$; $P = 0.001$) (Fig. 5*A*, *B* and *D*). Recent evidence suggests that after ~3 min exposure, oxytocin receptors appear to be internalized (George et al., 2024). Evidence from other brain regions suggests that GPCR desensitization is mediated by GRKs (George et al., 2024). To test whether this mechanism contributes to TGOT desensitization in RTN neurons we measured the peak firing response of RTN neurons to 3 min applications of TGOT (200 nM) following incubation in a selective GRK2 inhibitor (CCG258208, 500 nM) or DMSO (vehicle – control). In the presence of the GRK2 inhibitor TGOT produced only modest desensitization, as indicated by a slight reduction in the peak firing rate response during the second TGOT application (Fig. 5*D*). This ratio was similar under control conditions and in the presence of CCG258208 ($F_{1,12} = 1.865$; $P = 0.1971$) (Fig. 5*E*). Therefore under our experimental conditions we could not confirm a primary role for GRK2 in mediating TGOT-induced desensitization in RTN neurons.

### KCNQ channels contribute to oxytocinergic modulation of RTN chemoreceptors

KCNQ channels serve as a point of convergence for multiple Gq-coupled modulators in RTN neurons (Hawryluk et al., 2012; Soto-Perez et al., 2023) and in other brain regions. TGOT has been shown to stimulate neural activity by inhibition of KCNQ channels (Tirko et al., 2018). Therefore we tested whether KCNQ channels contributed to oxytocin-mediated activation of RTN neurons. We found that pharmacological blockade of KCNQ with ML252 (10 µM) increased baseline firing by ~1 Hz and attenuated TGOT responsiveness by ~45% ($\Delta = 1.7 \pm 0.7$ Hz *vs*. ML252: $\Delta = 0.8 \pm 0.5$ Hz; $t = 2.93$; d$F = 5$; $P = 0.032$) (Fig. 6*A* and *B*). Note that in the continued presence of ML252 cell activity was adjusted close to control levels by negative DC current

injection (Fig. 6*A* – arrows). Note also that ML252 exhibits greater selectivity for KCNQ type 1, 2, 2/3 and 4 subunits (Kanyo et al., 2023), and of these KCNQ2 is preferentially expressed by RTN neurons (Soto-Perez et al., 2023).

## Discussion

Oxytocin is recognized as a significant physiological modulator of breathing, with evidence indicating a direct projection from the PVN to the VRC, which may influence respiratory function (Geerling et al., 2010; Mack et al., 2002). Despite this potentially important physiological role, little is known regarding the mechanisms underlying oxytocinergic modulation of RTN/pFRG chemoreceptor neurons and respiratory control. Here we showed that at both the integrative and cellular levels oxytocin increases RTN chemoreceptor activity, at least in part, through the downstream inhibition of KCNQ channels. Although blocking the GRK2 subunit did not produce a significant change in desensitization, our data suggest that oxytocin can enhance breathing; however this effect may be time limited due to receptor desensitization. Further studies are needed to clarify the molecular mechanisms involved, which are critical for understanding the role of oxytocin in respiratory physiology and may inform the development of novel therapeutic approaches for respiratory disorders.

### Caveats and experimental limitations

Some experiments were conducted in the medullary brain slice preparation because it allows considerable control of the neuronal environment and easy access to neurons for patch-clamp recording. However it should be recognized that the tissue used in this preparation has been traumatized, subjected to oxidative stress in the form of hyperoxic incubation conditions (95% $O_2$), and neural network connections have been disrupted (Mulkey et al.,

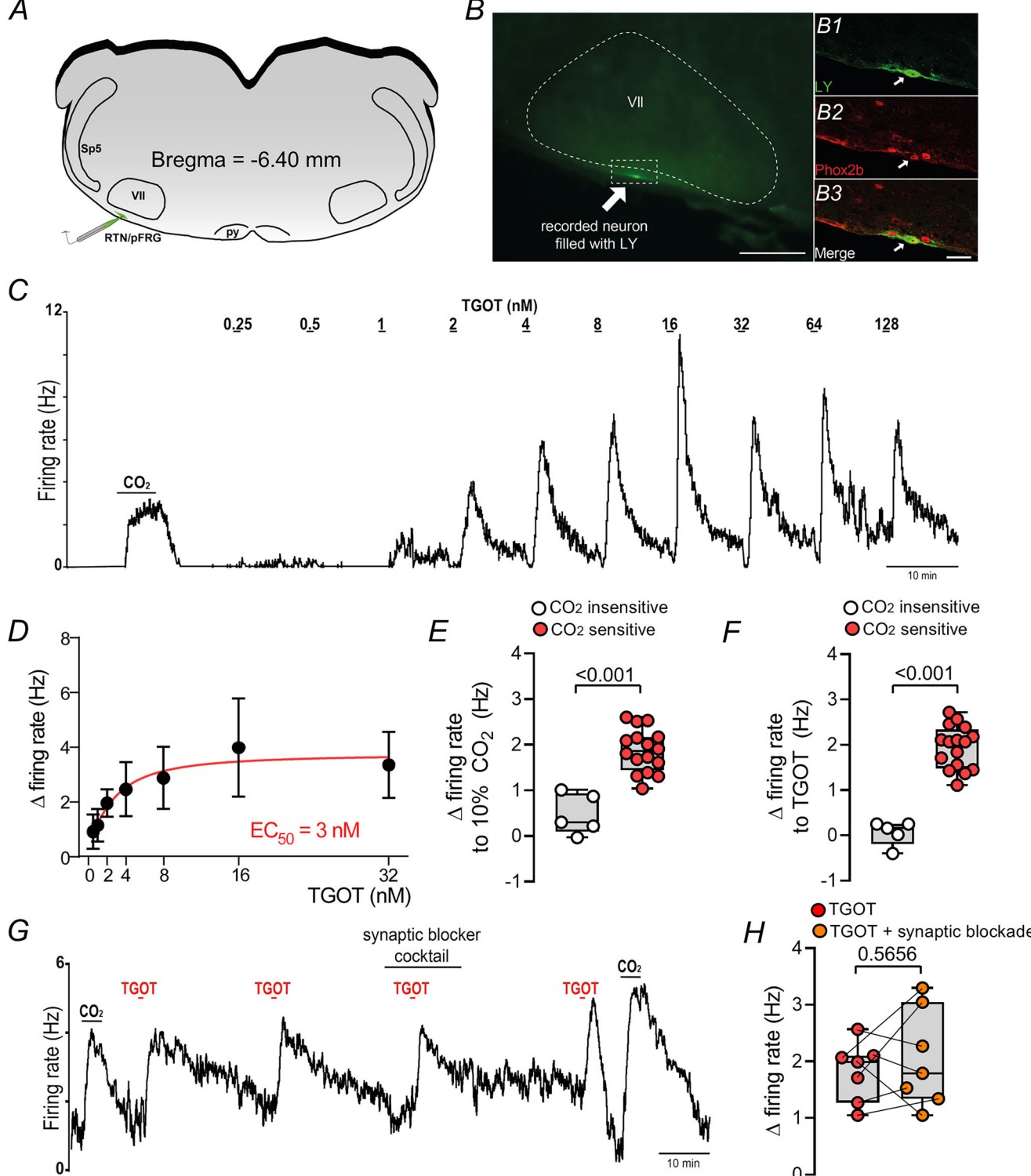

**Figure 4. RTN chemoreceptors are activated by oxytocin agonist**
*A*, schematic drawing of experimental approach to record RTN chemoreceptors activity. *B*, photomicrography showing the location of RTN neurons recorded. Double-immunolabelling shows a Lucifer Yellow (LY)-filled (green) $CO_2$/$H^+$-sensitive RTN neuron recorded in slice is immunoreactive for Phox2b (red). *C*, trace of firing rate showing the characteristic response of a chemosensitive RTN neuron to 10% $CO_2$ and to graded increases in oxytocin agonist (TGOT). *D*, non-linear of the dose–response curve showing average firing rate at each concentration plotted (black dots) were fitted (continuous red line) to a logistic equation of the form: $y = (a - c)/[1 + ([TGOT]/EC50)b] + c$, where a and c are the theoretical minimum and maximum, respectively, and b is a slope function (Li et al., 1998).

Each point represents data from 3 to 11 neurons from an equal number of mice. The calculated TGOT $EC_{50}$ was 3 nM. *E*, group data showing firing rate response to 10% $CO_2$ in chemosensitive ($CO_2$-sensitive; 16 cells from 16 slices and an equal number of mice) and non-chemosensitive ($CO_2$-insensitive; 5 cells from 5 slices and an equal number of mice) neurons located in the RTN/pFRG region (unpaired *t* test, $t = 5.87$, $dF = 19$, $P < 0.0001$). *F*, Group data showing firing rate response to bath application of TGOT (2 nM) in chemosensitive (16 cells in 16 slices and an equal number of mice) and non-chemosensitive neurons (5 cells from 5 slices obtained from an equal number of mouse pups) located in the RTN/pFRG region (unpaired *t* test, $t = 8.43$, $dF = 19$, $P < 0.0001$). *G*, trace of firing rate showing the response of an RTN chemoreceptor to TGOT (2 nM) under control conditions in a neurotransmitter receptor blocker cocktail containing CNQX (10 μM), gabazine (10 μM), strychnine (2 μM) and picrotoxin (50 μM) synaptic block solution. *H*, summary data (7 cells in 7 slices from an equal number of mouse pups) showing the TGOT-induced change in firing rate under control condition and in the presence of the blocker cocktail (paired *t* test, $t = 0.607$, $dF = 6$, $P = 0.5656$). py, pyramid tract; RTN/pFRG, retrotrapezoid nucleus/parafacial respiratory group; Sp5, spinal trigeminal nucleus; VII, facial motor nucleus. Scale bar = 100 μm in *B*; 20 μm in *B3*.

2001). However it is important to point out that we also ran experiments using an *in vivo* preparation, confirming our main hypothesis about the role of oxytocin in the modulation of breathing activity through the RTN/pFRG region. Moreover the scope of our experiments was constrained by the application of exogenous drugs. This is a potential issue because bath application of oxytocinergic agonists may not mimic the discrete and transient nature of endogenous neurotransmitter release, and differences in the spatiotemporal profile of oxytocinergic receptor activation can result in divergent neural responses (Carter et al., 2020).

We also performed optogenetic experiments to selectively stimulate oxytocinergic terminal-like endings and assess the contribution of oxytocin to breathing activity in anaesthetized adult mice. For this we used the Oxt$^{Cre/+}$::Ai32 transgenic line. Before we discuss the data it is important to address a key methodological consideration by verifying and quantifying ChR2 expression, specifically in oxytocinergic neurons, as a critical control. In our study we provide a representative confocal image (Fig. 3*J*) showing ChR2-labelled axons (GFP immunohistochemistry) colocalized with Oxt immunoreactivity in the RTN region. Although we did not quantify the total number of labelled neurons or terminals, we are confident that ChR2 expression in this line is largely restricted to oxytocin-producing neurons and their projections. This transgenic line has been extensively characterized for robust Cre-dependent ChR2 expression, with the Ai32 construct enabling strong light-evoked neuronal spiking at low light intensities (Li et al., 2024; Madisen et al., 2012; Zhang et al., 2024). Nonetheless recent studies have reported potential 'leaky' expression (Cre-independent ChR2-EYFP) in Ai32 mice, which could lead to off-target activation in some brain regions (Prabhakar et al., 2019). As shown in Fig. 3*J*, our histological data confirm ChR2 expression in oxytocinergic projections to the RTN. Future studies, combining functional recordings and histological verification in the same animals, will further ensure specificity and strengthen causal interpretations of behavioural and physiological effects.

Although our data demonstrate that optogenetic stimulation of oxytocinergic terminals influences breathing, we cannot definitively attribute this effect to oxytocin release. It is well established that oxytocinergic neurons can co-release other neurotransmitters, such as glutamate (Eliava et al., 2016; Knobloch et al., 2012), which may also contribute to the observed changes in respiratory activity. However our *in vitro* data showed that the respiratory response to TGOT, a selective OXTR agonist, was unaffected by the blockade of glutamatergic receptors, suggesting that the observed effects *in vivo* are likely mediated by oxytocin signalling rather than glutamatergic co-transmission. Future studies are needed to pinpoint whether endogenous oxytocin could modulate RTN chemoreceptor activity by a similar mechanism.

Another important caveat to consider is the difference in animal age and the temperature disparity between *in vitro* (22°C) and *in vivo* (37°C) experimental conditions. The activity of RTN neurons increased with temperature (Q10 = 2.4) (Guyenet et al., 2005), so it is reasonable to expect oxytocin sensitivity to increase with temperature by a similar amount. However to facilitate comparison with previous studies that characterized the response of RTN neurons to other modulators, including orexin (Lazarenko et al., 2011), serotonin (Hawkins et al., 2015; Shi et al., 2022), acetylcholine (Sobrinho et al., 2016), norepinephrine (Kuo et al., 2016; Oliveira et al., 2016) and histamine (Sobrinho et al., 2022), we chose to perform these experiments at room temperature. Regarding age, although our slice experiments were conducted in animals aged P7-12, and *in vivo* studies involved animals aged 8–15 weeks, we recognize that developmental changes could impact the results. Oxytocin receptor expression, for instance, has been shown to vary significantly with age, which could influence the observed physiological responses. However performing reliable brainstem *in vitro* experiments in adult animals is technically challenging due to extensive myelination, which complicates tissue slicing and electrophysiological recordings. Additionally conducting *in vivo* recordings of intercostal EMG (int$_{EMG}$) in neonatal mice presents significant experimental difficulties, making it impractical

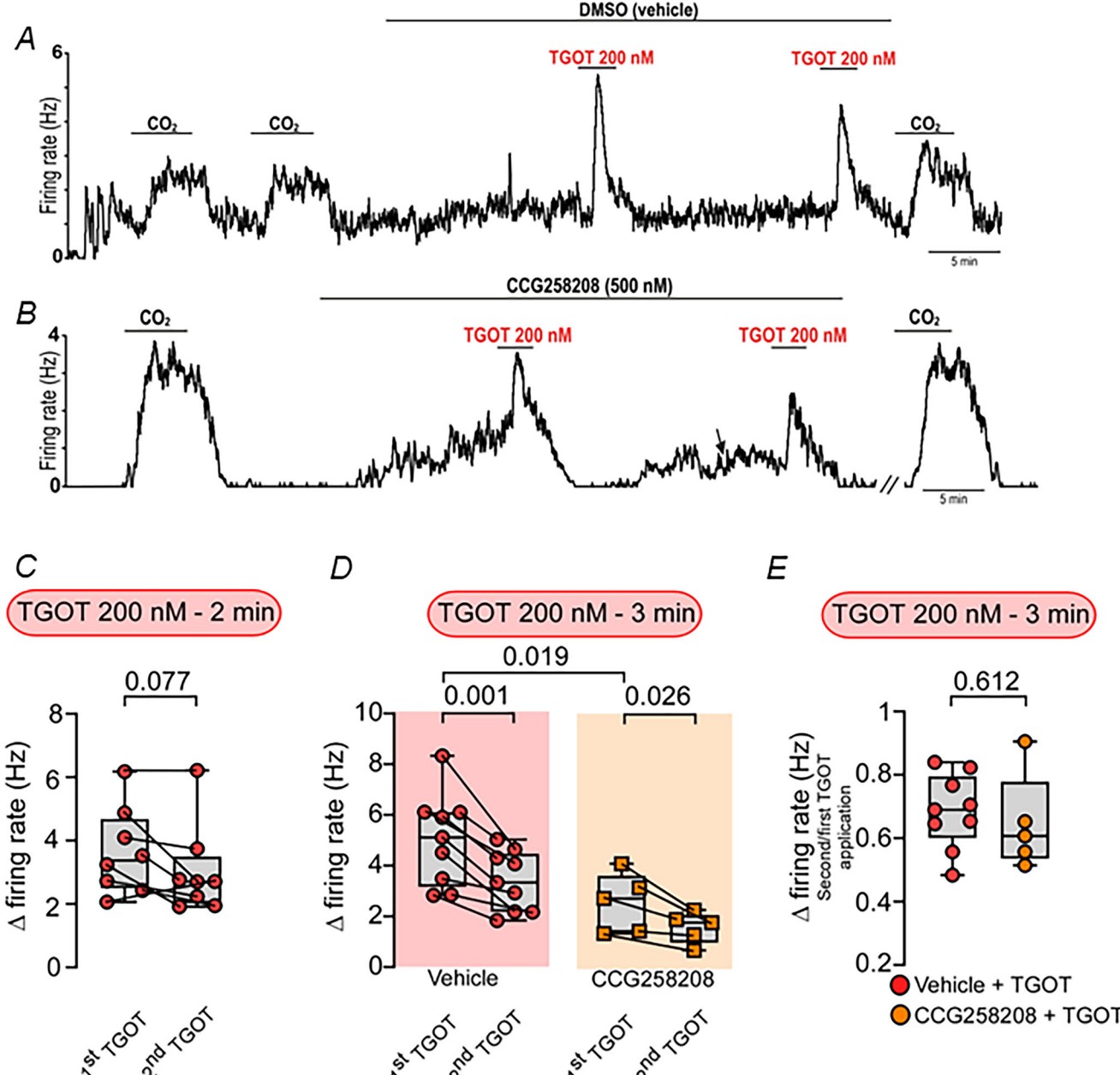

**Figure 5. Oxytocin agonist-induced desensitization of oxytocin receptor in the RTN/pFRG**
*A*, firing rate recording of RTN chemosensitive neuron showing two sequential bath applications of 200 nM TGOT for 3 min each separated by 20 min in the presence of DMSO (vehicle). *B*, firing rate recording of RTN chemosensitive neuron showing two sequential bath applications of 200 nM TGOT for 3 min each separated by 20 min in the presence of GRK2 blocker (CCG258208 – 500 nM). The arrow indicates the adjustment of the firing rate with DC current injection. *C*, summary data (8 cells in 8 slices from an equal number of mice) showing RTN activity of first and second bath application of TGOT (200 nM for 2 min, paired *t* test, $t = 2.066$, d$F = 7$, $P = 0.077$). *D*, summary data (9 cells in 9 slices from an equal number of mice (red circle) and 5 cells in 5 slices from an equal number of mice (orange square)) showing RTN activity of first and second bath application of vehicle + TGOT (200 nM for 3 min, paired *t* test, $t = 5.039$, d$F = 8$, $P = 0.001$) or CCG258208 (GRK2 blocker) + TGOT (200 nM for 3 min, paired *t* test, $t = 3.433$, d$F = 4$, $P = 0.026$). Two-way repeated-measures ANOVA, TGOT application *vs.* experimental group (vehicle + TGOT *vs.* CCG258208 + TGOT) ($F_{1,12} = 1.865$; $P = 0.1971$). *E*, change in firing rate (ratio of the second and first TGOT application) ($N = 5$–9 cells in equal numbers of slices obtained from the same number of mice) showing RTN activity of bath application of TGOT (200 nM for 3 min) before and after GRK2 blocker (CCG258208 – 500 nM, unpaired *t* test, $t = 0.521$, d$F = 12$, $P = 0.612$).

to perform such comparisons within the scope of this study.

## Oxytocinergic control of breathing function

The understanding of the hypothalamus's role in respiratory regulation has evolved significantly over the years (Burdakov et al., 2013; Mack et al., 2002). Traditionally the hypothalamus has been recognized for its control over various autonomic and endocrine functions (Fong et al., 2023). However it is now well established that the hypothalamus projects to the VRC, a crucial brainstem region involved in the generation and modulation of breathing rhythms (Del Negro et al., 2018; Fukushi et al., 2019). The hypothalamic projection to the VRC suggests a role in modulating respiratory responses under various physiological and behavioural conditions. One of the neuromodulators of the VRC can be oxytocin, which is involved in various physiological processes, including social bonding, stress responses and reproductive behaviours. Here we highlighted oxytocin's role in the modulation of respiratory functions, adding a new dimension to our understanding of this hormone.

Using a standard neuroanatomical approach we found a portion of Ctb-labelled excitatory neurons (VGlut2) were also immunoreactive for Oxt, suggesting that the RTN is one of several downstream targets of oxytocinergic neurons. These data are intended to provide a quantitative anatomical characterization of the cellular populations in the PVN projecting to the RTN. We described the presence of subpopulations with distinct neurochemical phenotypes (oxytocinergic and/or glutamatergic/GABAergic), and these data were not designed for statistical comparison among groups. This neuromodulatory effect is particularly interesting in the context of $CO_2$ sensitivity and the drive to breathe. This action of oxytocin in the RTN/pFRG can presumably modulate the responsiveness of respiratory centres to hypercapnia, which is a critical trigger for increased ventilation.

The RTN/pFRG is well established as a key site for respiratory control. Neurons (Mulkey et al., 2004; Wang et al., 2013), astrocytes (Gourine et al., 2010; Huckstepp et al., 2010; Wenker et al., 2010) and blood vessels (Cleary et al., 2020; Hawkins et al., 2017) in this region are key elements of the integrated $CO_2$-dependent drive to other components of the respiratory circuit to regulate both inspiratory and expiratory activity (Guyenet et al., 2019; Moreira et al., 2021; SheikhBahaei et al., 2024). During the prenatal period chemosensitive RTN/pFRG neurons also contribute to inspiratory rhythmogenesis (Onimaru et al., 2008; Thoby-Brisson et al., 2009). In adulthood subsets of RTN/pFRG neurons contribute to expiratory activity (Janczewski & Feldman, 2006; Magalhães et al., 2021; Pagliardini et al., 2011; Souza et al., 2020) and postinspiratory activity (Flor et al., 2024). Thus the RTN exerts a significant influence on all aspects of breathing. Despite the neuroanatomical evidence that oxytocinergic neurons project to the RTN/pFRG, we demonstrated that both pharmacological (using an oxytocin agonist) and optogenetic (using Oxt[Cre/+]::Ai32) stimulation of oxytocin receptors/varicosities produced a consistent increase in breathing activity, specifically an increase in breathing amplitude. The role of oxytocin activating RTN/pFRG to increase breathing amplitude suggests an adjustment to meet respiratory needs by taking deeper breaths rather than breathing faster. This can be beneficial in situations where deeper breaths are needed to increase oxygen intake or remove carbon dioxide more effectively without necessarily increasing the respiratory

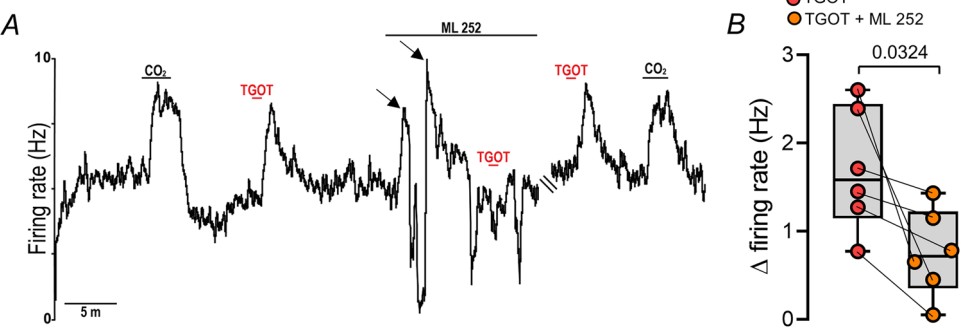

**Figure 6. KCNQ channels are the downstream target of oxytocin agonist in RTN chemoreceptors**
*A*, firing rate trace of a $CO_2$-sensitive RTN neurons showing that bath application of the KCNQ channel blocker ML252 (10 μM) increased baseline activity. In the sustained presence of ML252 the baseline activity was regulated to maintain levels close to control conditions by adjusting the electrical potential of the membrane to more negative values (adjust the firing rate with DC current injection. See arrows). Under these conditions the firing response to TGOT (2 nM) was reduced by 45% compared to control. *B*, summary data (6 cells in 6 slices from an equal number of mice) showing that ML252 decreased TGOT responsiveness of chemosensitive RTN neurons (paired *t* test, *t* = 2.937, d*F* = 5, *P* = 0.0324).

rate. Oxytocin is often associated with social bonding, stress response and emotional regulation. Therefore its role in modulating breathing amplitude could be linked to these functions, as changes in breathing patterns are often associated with emotional and psychological states. Deeper, slower breaths can be calming and may be part of the physiological responses to stress or anxiety, potentially mediated by oxytocin's effects (Kirsch et al., 2005).

Our findings demonstrate that optogenetic activation of presynaptic varicosities in the RTN/pFRG evokes breathing responses; however this does not necessarily establish glutamate release as the primary driver of these effects. Many presynaptic varicosities in the RTN are capable of co-releasing multiple neurotransmitters, including neuropeptides, which may contribute to the observed respiratory modulation. Additionally the activation of ionotropic or metabotropic receptors beyond glutamate receptors, as well as potential recruitment of local inhibitory or excitatory circuits, cannot be excluded. Importantly although our approach is sufficient to elicit breathing responses, it does not determine whether glutamate release is necessary for physiological respiratory control at the RTN/pFRG region. Future studies, using pharmacological blockade of glutamatergic receptors in the RTN/pFRG or selective manipulation of neurotransmitter release mechanisms, will be essential to clarify the specific contribution of glutamate signalling in this context.

### Oxytocinergic control of RTN chemoreceptor function

The molecular basis for the oxytocinergic control of RTN function likely involves GPCR-mediated inhibition of KCNQ channels. Specifically we showed that RTN neurons are strongly activated by TGOT (selective oxytocin agonist). This aligns with data showing oxytocin receptor expression within the RTN, suggesting that oxytocin signalling to the RTN neurons contributes to modulation of breathing in mice. Oxytocin receptors can signal though multiple types of GPCR that vary in their ligand sensitivity (Busnelli & Chini, 2018). The oxytocin sensitivity of RTN neurons ($EC_{50 = 3 \text{ nM}}$) is most consistent with signalling via Gq-coupled receptors. Once activated Gq-coupled receptors result in PLC$\beta$ activation, leading to the hydrolysis of PIP2 to generate IP3 and DAG. Considering the significant intracellular signalling roles of PIP2 depletion, IP3 and DAG (Kosenko et al., 2012), we propose these as the most likely mediators of oxytocin modulation of RTN neurons.

KCNQ channels, a classic downstream target of several peptide signalling pathways (Delmas & Brown, 2005; Tirko et al., 2018), are known to regulate RTN chemoreceptors activity (Hawkins et al., 2015; Hawryluk et al., 2012; Sobrinho et al., 2016; Soto-Perez et al., 2023). Thus we identified KCNQ channels as the primary

effector-coupling receptor activation to changes in neuronal excitability. Evidence suggests that KCNQ channels can be inhibited directly by PIP2 depletion (Delmas & Brown, 2005; Schroeder et al., 2000; Suh et al., 2006; Zhang et al., 2003), or indirectly through CaM- (Gamper & Shapiro, 2003), CK2- (Kang et al., 2014) or PKC-dependent mechanisms affecting PIP2 affinity (Kosenko et al., 2012). Considering that RTN neurons preferentially express KCNQ2 in the absence of other channel isoforms (Soto-Perez et al., 2023), and because the activity of KCNQ2 is highly dependent on PIP2, we suspect that Gq-mediated PIP2 hydrolysis is the most likely pathway linking oxytocin receptors to activation of RTN neurons (Hawryluk et al., 2012).

Consistent with this our results show that blocking KCNQ channels decreases oxytocinergic modulation of RTN chemoreceptor excitability, clearly implicating these channels as downstream targets of oxytocin. Although we do not yet fully understand the molecular mechanisms, we believe depletion of PIP2 is an important component of oxytocin modulation in RTN neurons. In summary we demonstrate that oxytocin increases the activity of RTN chemosensitive neurons in a dose-dependent manner likely through G protein–coupled signalling pathways that inhibit KCNQ channels.

### Oxytocin agonist-induced desensitization of oxytocin receptors in RTN neurons

We provide evidence of agonist-induced desensitization of oxytocin receptors in mouse RTN neurons. According to the literature the desensitization of the oxytocin receptor may occur during activation of the G protein–coupled receptor, before any desensitization effects, likely mediated by potassium channels (George et al., 2024). [Correction made on 23rd September 2025, after first online publication: This sentence has been updated to appropriately cite the work of "George et al. (2024)".] Therefore under our experimental conditions we could not confirm a primary role for GRK2 in mediating TGOT-induced desensitization in RTN neurons. In neurons desensitization of GPCRs, including oxytocin receptors, is known to occur at the level of G proteins and often involves the recruitment of multiple GRK isoforms, including GRK2, GRK3 and GRK6. This finding is somewhat unexpected, as previous studies have suggested that desensitization of oxytocin receptors may rely predominantly on a single GRK isoform – GRK2 in HEK293 cells (Smith et al., 2006) and GRK6 in uterine myometrium (Willets et al., 2009). The apparent requirement for multiple GRK isoforms in neurons suggests that oxytocin receptor desensitization, and likely that of other neuronal GPCRs, involves more diverse and redundant regulatory mechanisms. Building on the findings of George et al. (2024), future studies in RTN respiratory

neurons will focus on identifying GRK phosphorylation sites on the oxytocin receptor and substituting them with phospho-null mutants to further elucidate how GRK-dependent phosphorylation contributes to oxytocin receptor desensitization and internalization. [Correction made on 23rd September 2025, after first online publication: This sentence has been updated to appropriately cite the work of "George et al. (2024)".]

## Conclusion

Our study provides compelling evidence that oxytocin significantly modulates respiratory function by directly influencing RTN chemoreceptor neurons. We demonstrated that oxytocin increases the activity of these neurons via GPCR-mediated inhibition of KCNQ channels, likely leading to enhanced breathing amplitude. However we also observed receptor desensitization, which may reduce the respiratory effects of oxytocin over time. Understanding the molecular mechanisms underlying this desensitization is important for advancing knowledge of oxytocin's role in respiratory physiology and for developing potential therapeutic approaches for respiratory disorders.

Moreover our research highlights the broader implications of oxytocinergic modulation within the brainstem, particularly in respiratory control. The interplay between the effect of oxytocin on respiration and its other roles in the stress response and emotional regulation presents intriguing possibilities for future investigation. By elucidating the molecular pathways and cellular mechanisms involved, our study lays the groundwork for developing targeted interventions to enhance respiratory function in conditions like sleep apnoea, and for exploring the interconnectedness of the diverse physiological roles of this important neuropeptide.

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

## Additional information

### Data availability statement

The data that support the findings of the study are available from the corresponding author upon reasonable request.

## Competing interests

The authors declare that they have no competing interests.

## Author contributions

E.V.A., D.K.M., A.C.T. and T.S.M. designed the research; E.V.A., P.E.S., L.M.O, Y.S. and T.S.M. performed experiments and analysed data; E.V.A., P.E.S., L.M.O, Y.S., A.C.T., D.K.M. and T.S.M. interpreted results of experiments; E.V.A., A.C.T., D.K.M. and T.S.M. prepared figures and drafted manuscript; all authors reviewed and edited the manuscript. All authors have read and approved the final version of this manuscript and agreed to be accountable for all aspects of the work in ensuring that questions related to the accuracy or integrity of any part of the work are appropriately investigated and resolved. All persons designated as authors qualify for authorship, and all those who qualify for authorship are listed.

## Funding

Supported by public funding from São Paulo Research Foundation (FAPESP) (grants: 2021/09768-5 to A.C.T. and 2021/05299-0 to T.S.M.), Conselho Nacional de Desenvolvimento Científico e Tecnológico (CNPq) fellowship (306580/2023-3 to A.C.T. and 306418/2023-1 to T.S.M.) and São Paulo Research Foundation (FAPESP) fellowship (2020/08620-1 and 2022/15 609-0 to EVA and 2021/0 9377-6 to PES). This work was also funded by grants from the National Institutes of Health, National Heart, Lung, and Blood Institute: HL104101, HL137094 to DKM; and HL108609 to DAB.

## Acknowledgements

The authors wish to thank Dr Douglas Bayliss (University of Virginia) for sharing the single-cell RNA-seq dataset.

The Article Processing Charge for the publication of this research was funded by the Coordenação de Aperfeiçoamento de Pessoal de Nível Superior - Brasil (CAPES) (ROR identifier: 00x0ma614).

## Keywords

breathing, central chemoreceptor, oxytocinergic signalling, RTN

## Supporting information

Additional supporting information can be found online in the Supporting Information section at the end of the HTML view of the article. Supporting information files available:

**Peer Review History**

