## [Peer Review History · The Journal of Physiology]

Oxytocinergic signaling in the respiratory parafacial region increases the activity of chemosensitive neurons and respiratory output

Emmanuel V Araújo, Phelipe E Silva, Luiz M Oliveira, Yingtang Shi, Ana C Takakura, Daniel K Mulkey, and Thiago S Moreira

DOI: 10.1113/JP287845

Corresponding author(s): Thiago Moreira (tmoreira@icb.usp.br)

The following individual(s) involved in review of this submission have agreed to reveal their identity: Sean Williams (Referee #3)

Review Timeline:

Submission Date:	10-Oct-2024
Editorial Decision:	11-Nov-2024
Revision Received:	20-Mar-2025
Editorial Decision:	22-Apr-2025
Revision Received:	06-May-2025
Editorial Decision:	02-Jun-2025
Revision Received:	16-Jul-2025
Editorial Decision:	11-Aug-2025
Revision Received:	12-Aug-2025
Accepted:	18-Aug-2025

Senior Editor: Harold Schultz

Reviewing Editor: Daniel Zoccal

Transaction Report:

Dear Dr Araújo,

Re: JP-RP-2024-287845 "Oxytocinergic signaling in the respiratory parafacial region increases activity of chemosensitive neurons and respiratory activity" by Emmanuel V Araújo, Phelipe E Silva, Luiz M Oliveira, Ana C Takakura, Daniel K Mulkey, and Thiago S Moreira

Thank you for submitting your manuscript to The Journal of Physiology. It has been assessed by a Reviewing Editor and by 2 expert referees and we are pleased to tell you that it is potentially acceptable for publication following satisfactory major revision.

REVISION CHECKLIST:

We look forward to receiving your revised submission.

Yours sincerely,

Harold Schultz
Senior Editor
The Journal of Physiology

EDITOR COMMENTS

Reviewing Editor:

Comments to the Author:

I thank the authors for submitting their manuscript to the JPhysiol. Two experts in the field assessed the study, recognizing the strength and novelty of the study's findings. They also indicated that the manuscript could be influential; however, it is pending substantial revision and potentially new experiments. Overall, the referees commented on the lack of essential information and inaccurate description in the Methods section (affecting reproducibility and data interpretation), inappropriate use of statistical tests, and the absence of control/additional experiments that could validate the authors' hypotheses. I recommend that the authors follow the constructive comments presented by the referees and carefully revise the manuscript to address all critical aspects affecting and limiting data interpretation and conclusions.

Senior Editor:

Comments for Authors to ensure the paper complies with the Statistics Policy:
Please include actual p values in figures.

State the statistical test(s) used in tables and figures in their respective legends.

Comments to the Author:

Thank you for submission of your research article to the Journal of Physiology for consideration. The article has been reviewed by experts in the field and found to require a major revision to address all of the concerns raised including additional experiments before further consideration can be given. Please address all comments from the external referees and reviewing editor as well as addressing the list of requirements or publication in the journal included in this letter.

It is important that the manuscript follow the Journal's policies on Rigour & Reproducibility requirements as listed in the document linked below.

<https://physoc.onlinelibrary.wiley.com/pb-assets/hub-assets/physoc/documents/TJP-Rigour-and-Reproducibility-Requirements-1724673661727.pdf>

Specific concerns (but not exclusive):

Animal Experiments:

For tracer experiments, details of anesthesia, maintenance of surgical plane, and the surgical procedure are needed. And

the details of method of euthanasia (doses, route, confirmation of death) are needed.

It is recommended to describe procedures on the animals first and then procedures on tissue following description of animal termination. Thus the section on cell counts, mapping and analysis seems out of place.

There is concern that methodological procedures are not adequately described or rigorous.

There is concern that the statistical tests are either not appropriate or are not adequately described.

REFeree COMMENTS

Referee #1:

Araujo and colleagues examined the potential role of PVN-derived oxytocin within the RTN on respiratory activity. This was achieved via in vivo respiratory monitoring during oxytocin agonist nanoinjection or channelrhodopsin activation of Oxt fibers. The physiological analysis was coupled with in vitro slice electrophysiology and tract tracing. Overall, the authors present an intriguing set of results to suggest that Oxt increases RTN firing through inhibition of KCNQ channels. While the authors integrate their results across multiple levels, a strength, there are some concerns that are noted below.

The authors address the potential discrepancy and limitations of the in vitro and in vivo work in the discussion. While this is appreciated, the authors do not address perhaps the larger issues: differences in the age (slice, P7-12 vs in vivo 8-15 weeks) and temperature (22 C vs 37C) studied. The expression of Oxt or its receptor changes with age (e.g., PMID: 24376405, 24863032), among other potential issues regarding channel expression or activity at different temperature. The authors need to, at a minimum, address these age and temperature concerns. It would be more ideal to perform at least a subset of slice experiments in adults near physiological temperature.

Fig 1B, 2C&D. Please provide a zoom/magnified image of the PVN phenotypes the authors wish to highlight.

Fig 2E&F. Are there any statistical differences among these cellular groups? If not, the authors need to temper the conclusions regarding Oxt phenotype in glutamate or gaba neurons.

Pg 13, second paragraph and Fig 3. The authors state the control mice (Oxt-Cre^{-/-}) data were not shown, however, I believe they are presented in Fig 3. Please confirm and correct, if needed. In addition, when comparing these mice to those expressing ChR2, the more appropriate statistic would be an unpaired t-test rather than the paired t-test stated in the figure legend. In Fig 3H, please state in the legend what the 2 traces that are overlaid in this panel represent.

Pg 15. Block of GRK2 did not significantly affect the response to TGOT, as shown in Fig 5D. Thus, the authors cannot state the CCG blocker "prevented the attenuation." The authors either need to increase the sampling size to achieve statistical difference between the 2 groups, or remove such a statement.

Discussion. Oxt neurons co-label with glutamate (pending statistical differences in Fig 2EF). The application of TGOT to the RTN suggests Oxt itself can influence breathing (and neuronal activity). However, optogenetic activation of these terminals in the RTN does not rule a significant (or even primary) role of glutamate release on breathing responses. The authors should comment on such possibilities.

Pg 18, para 1. The authors need to remove the line regarding unpublished OxtR expression in the RTN, or provide the data for such a statement.

Pg 19, para 2. The authors have no evidence for OxtR (or agonist-induced) internalization. Nor do they show "clearly" that neuronal OxtRs are phosphorylated. These statements need to be tempered or additional experiments performed for such claims.

Pg 19. Para 1 of conclusion, line 4. "We demonstrated that oxytocin increases the activity of these neurons via Gq-mediated inhibition of KCNQ channels, leading to enhanced breathing amplitude". Please add "likely" in front of the word leading to state "likely leading to" as there several steps that are inferred between the slice and in vivo studies.

Minor

Pg10. The authors state 30nL was nanoinjected. How was this volume determined. Please state.

Pg 12. "Blacklalled" Perhaps the authors intended "backlabeled"

Pg 15, paragraph 1. Perhaps "bath application" rather than "back application" (assuming this is what the authors intended rather than pipette application).

Referee #2:

This article by Emmanuel V Araujo and colleagues presents data on the modulation of the activity of chemosensitive neurons in the RTN by oxytocin, released from hypothalamic PVN neurons, and its effect on breathing. This is an important area of research, since the neuromodulator oxytocin is involved in many behaviours that include changes in breathing activity, yet little is known regarding the mechanisms that induce these effects. In their study, the authors investigated a specific pathway, from PVN oxytocinergic neurons to RTN chemosensitive neurons, showing that oxytocin increases the activity of these neurons specifically (by opposition to RTN non-chemosensitive neurons) partly through a KCNQ channels-mediated pathway. This work is very original, on a specific question that, to my knowledge, has never been investigated before. The large diversity and complementarity of the experiments used, at the anatomical and functional levels, and from the in vivo integrated context down to in vitro cellular investigation, represent an extensive body of work that should provide significant strength to the interpretations and conclusions drawn. Yet, as described below, the manuscript contains many methodological issues, that unfortunately cast a doubt on how the experiments were performed and analysed, and therefore on the validity of their interpretation and the conclusions drawn.

Also, please provide line numbers to facilitate future reviewing process.

Main concerns:

My primary concern is regarding the description of the methods used in this study and of the data obtained, which suffer from many inaccuracies, absences of necessary information (e.g. immunohistochemical labelling fully absent from the methods section), and other issues that unfortunately cast a doubt on the appropriateness with which the experiments were performed. Also, most of the statistical tests used are inappropriate and poorly described. Here is a non-exhaustive list of issues that must be resolved - authors are encouraged to check for and correct issues beyond this list:

- Methods section "Animals": the description of the mouse lines used is difficult to read and understand, and suffers from many mistakes. For instance, only one reference (in brackets) is provided for both the VGlut2::Cre and the VGat::Cre mouse lines, and this reference is actually for another mouse line, the Phox2b::Cre line. Also, the same sentence states that

VGlut2::Cre or VGat::Cre mouse lines would be crossed with Ai6 or Oxt::Cre lines, the latter making no sense. In the next sentence, the way the Ai14 line is presented uses a different format. The ChR2 line used for optogenetics is called a "reporter mouse line", which is not classical (reporter mouse lines are classically used for assessing anatomy with expression of a fluorophore, which the authors did not do, as stated below).

- There are two different paragraphs in the methods section to describe the processing and analysis aspects of the histology performed. These two paragraphs should be merged and homogenised, it is difficult to understand what was actually done. It is difficult to understand how the cell counts were performed. In particular, I could not find any mention about how the PVN neurons were counted, although these data are critical for figure 2 and its interpretation.

Did you use confocal microscopy to assess double-labelling (or any other technique that enables optical sectioning), or only epifluorescence? If only the latter (as it seems from the methodology provided), and given the thickness of the sections used, double labelling cannot be definitively assessed, so caution must be taken in the way the data are presented.

- Very problematic, there is absolutely no description of any immunohistochemical labelling being performed in the methods section, while the results present data that one can only be assumed to come from immunohistochemistry (yet the presentation of these results in the results section, in the figures and their legends is so brief that it is hard to know what was done). There is no mention of the antibodies used, the protocol etc.

- In the legends of Figure 1 and 3, mentions are made regarding "axonal drawing" of PVN projections and "computer-assisted draw of the centre of the TGOT injection site", however there is no methodological description of these, how these were done, with which software etc.

- The methods section describes the use of the Phox2b::Cre;Ai14 mouse line for "both in vitro (cellular experiments) and in vivo (anesthetized mice) experiments". However, I could not find any mention of the use of these mice in the results section or in the figures and their legends. Did you really use these mice, and why? The results section only mentions immunoreactivity for Phox2b, which I guess implies that it was done with immunohistochemistry (although no information on this, cf. above).

- The authors make several references to previous publications in the methods section, stating that they used similar protocols. In general, the authors should give enough information in the current manuscript for the reader to understand what was done and be able to replicate the data. A very problematic example is in the paragraph "Physiological preparation". The first sentence of this paragraph states "The surgical procedures and experimental protocols were similar to those previously described (Oliveira et al., 2019)". However, the article cited does not describe similar protocols as used in the current manuscript. The authors must provide sufficient and appropriate descriptions of the methodologies and protocols they used. Here, for instance, there is no indication (and the reference does not provide any information) on how the photostimulation was performed (types of optical fibers, route of entry in the brain etc.), how the drugs were applied, how the intercostal EMG was recorded, how the mechanical ventilation of the mice was performed etc.

Further, regarding the sentence "This instrument provided a reading [...] horizontal reading during expiration": what instrument? No context is provided. Also, I do not understand the message that the authors try to convey in this sentence.

Also, why did you switch from isoflurane to urethane?

- The authors used several protocols with injections in deep structures, including the RTN/pFRG, which is a very small structure. My understanding from the methods provided is that all injections were based on stereotaxic coordinates relative to the lambda. There is significant variability and subjectivity involved in using the lambda (or bregma) as reference points, which impact the accuracy of targeting deep and small structures. The authors only mention this once, in the first paragraph of the results section, stating that 4 out of 5 injections in the PVN were correctly targeted.

Since the accurate targeting is critical for many experiments in the manuscript, and must be a pre-requisite for the inclusion of the data in this study, the mapping of the injections sites and of the expression profiles of transgenes must be provided for each animal in all conditions. These are very important validation data for these kinds of experiments.

- Which CTB did you use, linked to which fluorophores? Please provide origins of all drugs, tools etc. used, with enough information for the reader to be able to purchase the same agents.

Also, it is stated that CTB was dissolved in sterile water. Usually, vehicles for drug injections in the brain contain salts to avoid osmotic shock. Using only water as vehicle could induce cellular damage in the injection area.

- Viral expression of ChR2-mCherry in PVN oxytocinergic neurons was used to trace oxytocinergic axons in the brainstem. This is not an optimal way to map axonal projections, since the fluorophore mCherry is fused to ChR2 in this construct, and since ChR2 is efficiently trafficked to the membrane, compared to cytoplasm-filling fluorophores that efficiently and densely invade all cytoplasmic compartments including in neurites. Only 7-10 days were allowed here for viral expression, while AAV peak expression is typically reached after 10 days. In viral tracing experiments involving long-range tracing such as from the PVN to the brainstem, and with cytoplasm-filling fluorophores, it is classical to allow transgene expression for 4 weeks or more. Here with the membrane-bound ChR2-mCherry molecules, trafficking down remote parts of axons would likely require even more time. And oxytocinergic neurons are notoriously difficult to transduce with viruses for transgene expression, in particular to reach appropriate levels of transgenes in axons in the brainstem (Pinol RA et al, J Neurosci Met 2012, PMID: 22890236). Still, despite all these limitations, the protocol used in the current study seems to have induced detectable expression in the brainstem, as shown on Figure 1. No information is provided as to whether the labelling shown is from the native mCherry fluorescence, and if immunohistochemical detection/amplification was performed, this information is necessary. Also, the volume of virus injected, the number of injection spots etc. are not indicated in the methods section.

- The "Statistics" paragraph in the "Methods" section does not provide information on the statistical tests used, and refers to the results section, the figures and the figure legends. However, all that is indicated in these sections is the name of the test used, which is always a paired t-test (cf. the legends of figures 3 to 6). First, this type of test requires a normal distribution and equal variance of the data, and no information is provided on the appropriate testing of the distribution and variance. Given that the "n" in each group is relatively low in most experiments (some with n=5 and n=6 per groups compared), it is actually difficult to test the distribution of the data accurately, so it is generally advised to use a non-parametric test in these conditions. Second, and more problematic, the paired t-test used in all conditions tested in this study is not appropriate to most. A paired t-test implicates that the same experimental group was tested in two conditions (pairwise aspect). However, for most conditions tested, the groups tested are different, like for instance on Fig. 3I-K where *Oxt::Cre(+/-);Ai32* mice are compared to *Oxt::Cre(-/-);Ai32* mice, or on Fig. 4E-F where non-chemosensitive RTN neurons are compared to chemosensitive RTN neurons. It is totally incorrect to use a paired test in these conditions, and it wrongly increases the probability of identifying significant differences. One experimental condition where a paired test could indeed be used is the condition shown on Fig. 5, yet here again there are further issues, as the "n" on Fig. 5D is not the same for the condition "TGOT" and the condition "TGOT + CCG258208". Yet, by definition, a paired test is applied in a condition where the same factor is tested in two different conditions, so the "n" must be the same in each condition. Overall, there are numerous issues with the statistical analysis performed in this study, which raises serious concern regarding the validity of the results and their interpretation. Such issue, in addition to all other methodological and data description issues, unfortunately raise concerns regarding how the whole study was performed.

- The results section states that the virally-induced ChR2-mCherry expression showed "high selectivity (98.7 +/- 1.3 %; N = 4)". It is not clear what this percentage refers to, the percentage of *Oxt* neurons expressing ChR2-mCherry? Or the percentage of ChR2-mCherry expressing cells that are *Oxt* cells? In any way, both of these counts should be provided. Also, from the image shown on Fig. 1B, it seems that there is a significant proportion of ChR2-mCherry cells that are not *Oxt*-positive. An accurate counting, with individual data per animal, should be provided (also, as per above, no information is provided on how *Oxt* neurons were labelled).

- The interpretation of the retrograde tracing data throughout the manuscript is incorrect, due to percentages calculated incorrectly. There are 6 different conditions counted in each mouse line (*VGlut2::Cre;Ai6* and *VGat::Cre;Ai6*) on Figure 2E-F. The percentages provided in the manuscript that come from comparisons of these counts are calculated incorrectly. For instance, it is stated in the abstract that 42% of oxytocinergic and glutamatergic neurons project to the RTN/pFRG. That means, in terms of calculations based on the groups shown on Figure 2E, that the mean value for the total number of neurons in the "Triple" (*VGlut2 + Oxt + Ctb*) group (gray) divided by the mean value for the total number of neurons in the "*VGlut2 + Oxt*" group (yellow) equals 42%. Or in other words: $(VGlut2 + Oxt + Ctb) / (VGlut2 + Oxt) = 42\%$. Yet, although the authors do not provide the actual numbers, it seems from Figure 2E that the "Triple" group has a mean value of about 20 in total number of neurons counted, while the "*VGlut2 + Oxt*" group has a mean value of about 110. So $20/110=18\%$, not the 42% announced. Further, the same 42% number is given in the results section (end of first paragraph), this time for a different count, the "*Oxt* neurons immunoreactive for both CTB and *VGlut2*". So here, the count is made on $(VGlut2 + Oxt + Ctb) / Oxt$ alone, and not $(VGlut2 + Oxt)$ as in the abstract. Again, from an estimation of the values given on Figure 2E, that would be $20/150=13\%$, not 42%.

This is an important point, as the authors seem to then over-interpret these data, for instance stating in the discussion that "a significant proportion of excitatory oxytocinergic neurons (VGlut2+/Oxt+) in the PVN have been shown to project to the VRC, more specifically to RTN/pFRG...". These aspects should be moderated throughout the manuscript, with accurate calculations made and correctly interpreted.

The end of the first paragraph of the results section describing all the percentages calculated is confusing to me in terms of the calculations made, the interpretations made etc. It would help to provide all the numbers and all relevant calculations made in a table.

- End of the second paragraph of the results section, the presentation of the photostimulation data is confusing. First the delta change in IntEMG amplitude for the "Oxt-Cre-/-:Ai32" mice is given as 6.40 %, and a few sentences after it is given as 0.20 %. It is difficult to understand what is actually being shown in terms of data.

Also, in the last sentence it is stated "in which RTN neurons did not express ChR2", which is very confusing as this is not what was tested.

- The assessment of the selectivity of the ChR2 expression in oxytocinergic neurons in Oxt::Cre(+/-);Ai32 mice must be shown and quantified. This is an important control experiment, selectivity in double transgenic mice with Cre-lox recombination is never a given, yet it is a prerequisite for the validity of the experiments.

Also, it would be important to show an image of the ChR2 expression in oxytocinergic fibers in the RTN/pFRG in these mice.

- The presentation of the data on Fig. 4E-F is unclear, between the beginning of the third paragraph of the results section that states that Fig. 4E shows a response to CO₂ increase, while the figure legend for E and F are basically repeats of each other (so what is the difference?), and no labels on the panel F on Fig. 4, it is all very confusing.

- Third paragraph of the results section, "we use 2 nM TGOT to ensure that we elicit a near maximal response of RTN neurons to oxytocin agonist", while the EC₅₀ was measured as 3 nM. So 2 nM is below the EC₅₀, which is then by definition a dose that induces less than 50% of the maximal response, I do not see how this is a "near maximal response".

In the same sentences, the presentation of the results are confusing, it is for instance referred to as "vs. baseline of non-chemosensitive neurons" which is unclear, also Fig. 4E seems to change from being a response to TGOT or CO₂ etc. Please clarify these aspects, exactly what was done in each condition, what you mean by baseline, what it quantified, and provide the statistical tests used (same comment for the results shown at the end of this paragraph).

- Regarding the data relative to the application of ML252, presented in the 5th paragraph of the results section, a possibility explaining the results could be that application of ML252 induced a maximal increase in the firing rate of chemosensitive RTN neurons so TGOT could not induce any additional effect. Did you try to increase CO₂ concentration under ML252 application, to show that the RTN chemosensitive neurons recorded can still respond to a stimulus and increase their firing rate? This would be an important control experiment, especially since the discussion is strongly focused on this point and quite affirmative in KCNQ channels being the molecular proxy for the oxytocinergic effects of RTN chemosensitive neurons.

Also, the authors state in the legend of Figure 5 that a DC current injection was made to adjust "baseline activity". This statement is unclear and confusing, please explain precisely what was done, this should be included in the methods section, as this could also help to answer my point above.

- Third paragraph of the discussion, it is stated that "Oxytocin's action in the RTN/pFRG can presumably modulate the responsiveness of respiratory centers to hypercapnia, which is a critical trigger for increased ventilation". This is a very important and logical point, and I was surprised that it was not tested in this study. Especially since it is also mentioned in some way in the introduction, with "recent evidence indicates that their chemosensitivity is largely mediated indirectly, partly through neuromodulatory effects, including inputs from hypothalamic neurons" (by the way, the concept of indirect mediation of chemosensitivity is unclear to me). Did the authors try to test this by stimulating optogenetically the oxytocinergic fibers in the RTN/pFRG before versus during hypercapnia, and/or by applying TGOT to slices during increased CO₂ concentration?

- Paragraph "Oxytocinergic control of RTN chemoreceptor function", it is stated "This aligns with our unpublished data showing oxytocin receptor expression in the RTN". If you have these data, please present them in this study as they would be very important to support the overall conclusions of the manuscript. Otherwise, you cannot use unpublished (and therefore unverified) data to support important conclusions, this aspect must be removed.

- Paragraph "Oxytocinergic control of RTN chemoreceptor function", it is stated that the oxytocin receptor is a Gq-coupled receptor, and the discussion focuses on this pathway and becomes strongly hypothetical. However, the oxytocin receptor is coupled to many other G proteins, and can activate a much larger variety of intracellular cascades than the ones described here. For instance, please refer to these two great reviews on the matter: Busnelli M and Chini B, *Curr Top Behav Neurosci* 2018 (PMID: 28812263), and Jurek B and Neumann ID, *Physiol Rev* 2018 (PMID: 29897293). These aspects of the discussion should be moderated. In particular, the conclusions "In summary, we show that oxytocin increases chemoreceptor activity in a dose-dependent manner through Gq-mediated inhibition of KCNQ channels" (discussion) and "via Gq-mediated inhibition of KCNQ channels" (conclusion) are over-interpretations and must be moderated, the results obtained do not enable to state that so affirmatively (for the Gq-mediated aspect, but also the aspect on KCNQ channels, as the ML252 data only show a reduction by half of the TGOT effect, not a full blockade).

Minor points:

- There are too many typos and minor syntax issues throughout the text to be listed comprehensively here. Please proofread the text carefully, ideally with a native English speaker.

- In the 4th point of the "Key points", and in the penultimate sentence of the abstract, maybe modify "terminals" into "receptors/terminals", otherwise the sentence is not appropriate regarding the TGOT aspect.

- Nomenclature: throughout the manuscript, the way transgenic mouse lines are termed keeps on changing (e.g. "VGlut2::Cre", "VGlut2::Ai6", "Phox2bTdTomato" etc.). Please use the classical nomenclature, with for instance: "VGlut2::Cre", "VGlut2::Cre;Ai6", "Phox2b::Cre;Ai14" etc.

- Many drugs were used to block different types of receptors (CNQX, strychnine, gabazine etc.). Although these are classical drugs, it could be useful for the reader if you cited references to justify the concentrations used. This is particularly important for lesser-known drugs such as CCG258208 and ML252.

- At the end of the "histology and analysis" paragraph, it is stated that alignment of the brain sections on the atlas was performed based on 40 µm intervals, but the brains sections performed are 30 µm thick. Why this difference?

- Beginning of the results section, the Chr2-expressing AAV used is written in two different ways in two consecutive sentences, which could be confusing. Also, the attempt to abbreviate the virus name must not remove critical parts, such as the double floxed component. Maybe just stating "AAV" in the second sentence could be sufficient.

- Beginning of the results section, it is stated that "terminals" are labelled, however no labelling of terminals, such as synaptophysin or other pre-synaptic markers, was performed (or indicated). How could it be known that there are terminals being labelled? Especially since Oxt is often released "en passant" and not in terminals.

- Beginning of the results section, the Dbx and NK1-R are mentioned in a way that implies that these markers were labelled in this study, however no image is shown, and there is no mention of these experiments anywhere else. Please clarify.

- Results section, sentence "The density of PVN [...] medullary raphe (Figs. 1E-H)": I do not understand this sentence, it seems that the VRC is opposed to itself (written "VRC" and then "ventral respiratory column" to supposedly depict two different areas, while the first is the acronym of the second), and the whole sentence is confusing. Also, the Raphe data are not shown, and there are no quantified data provided to support these claims.

- It is mentioned at the end of the second paragraph of the results section that "the effects of photostimulation on breathing were highly reproducible within each animal". This aspect, of internal replication of results, is very important. However, this is not further explained. Please state in the methods section if the data provided for the photostimulation effects are averaged data of multiple stimulation trials in each animal, or how you selected which trials to measure, and how this internal (intra-animal) replication of effects was taken into account.

- Fourth paragraph of the results section, "G protein-coupled receptor kinases [...] desensitization", please provide a reference supporting this statement.

- Fifth paragraph of the results section, "blockade of Kv7.2 with ML252", why so specific, while the drug ML252 has been shown to block more channels, including Kv7.3 for instance?

- First sentence of the discussion, "Oxytocin is known as a powerful physiological stimulus for breathing". This is a very strong statement, for which only one reference is cited (Geerling et al., 2010), which provides only anatomical data, and is therefore not a valid reference to support the functional statement made of "powerful physiological stimulus for breathing". To my knowledge, the literature is actually quite thin on the impact of oxytocin to stimulate breathing, and this is an aspect that strengthens the importance of the current paper on the role of oxytocin on RTN neurons to stimulate breathing. Still, some groups have worked on this aspect of oxytocin and breathing, including the classical papers by Mack SO and colleagues (J Applied Physiol 2002 and 2007). The authors should provide more references for such strong statement, references that actually show functional work.

- Paragraph "Oxytocin agonist induced internalization of oxytocin receptors in RTN neurons" in the discussion. It discusses the potential mechanisms of internalization of the oxytocin receptor without providing any reference. Yet, this had been studied and should be referred to, cf. for instance Busnelli M and Chini B, Curr Top Behav Neurosci 2018 (PMID: 28812263).

- Overall, a lot of text shown on the figures is small (in particular the legends of some axis) and the reading of the figures would be facilitated if the size of some of the text was increased. Also, for axes with numbers, please show more numbers than just those for the extremities of the axes, to facilitate the understanding of the graphs. This is particularly important for the graphs represented on Figure 3D-F, where the value "0" is not represented in the y axes. An x axis could help for these graphs, at least as a dotted line, since it would make it easier to see when values increase or decrease.

- Fig. 1, the virus used should be fully named at least either in the legend or on the figure in A.

- In all figures, scale bars should be shown in each panel, one scale bar applying to multiple panels as currently done is confusing and inaccurate. This is particularly evident for Fig. 2C-D, where the panel in D was clearly shrunk in width compared to C, as attested by the "VGATcre::Ai6" writing width compared to the "VGlut2cre::Ai6" writing width. This suggests scale bars could be impacted.

- Figures 1A and 2B, the graphical depictions could make it seem like the PVN and the RTN/pFRG are at the same rostro-caudal level. It could be useful to modify the picture to make it clear that these structures are at different locations in the brain (adding rostro-caudal coordinates, and depicting anatomical separation). Also, it must be stated, either on the figures (with axis) or in their legends, that these are coronal views.
- On Figure 1C, E and G, what is the green labelling?
- On Figure 2C, arrows are very hard to see, and commented nowhere.
- It would be useful to present images with higher magnification from the images shown in Figure 2C and D, to be able to assess colocalization of markers.
- On Figure 2D, it seems that "VGat2" is written as the green labelling (although hard to see). Why the "2"? Overall, "VGat" changes in the way it is written throughout the manuscript.
- The legend of Figure 3 states that the photostimulation protocol was performed at 10 Hz, while the rest of the manuscript states 15 Hz. Which one was it? Also, why 15 Hz?
- Figure 3H, the yellow and red traces must be explained.
- Figure 3H, there is a delay of almost 1 min between the beginning of the photostimulation and the beginning of increase in IntEMG amplitude. Was this the case in all animals and all photostimulation trials in each animal? What could be the meaning of such delay?
- Legend of Figure 4G, isn't the picrotoxin application missing in the text?
- Figure 4, I do not understand the concept of "saline" injection as a control. The slices are perfused with aCSF, and the TGOT is bath applied, so what is the "saline" control?
- Figures 5 and 6, in conditions where paired tests can actually be used, it would be appropriate to show the graphs with lines linking the dots representing of each neuron recorded in the different conditions. This would explicitly show the pairwise aspect of the experiments.
- Legend of figure 6A, what does "increased basal activity" mean? "Basal" as opposed to what, increased by hypercapnia? This is not clear to me.
- Some references cited in the text are not present in the reference list, including (George et al., 2024) cited in the 4th paragraph of the results section, and (Flor et al., 2024) cited in the discussion.

Side note:

I really appreciated the "Experimental limitations" paragraph in the discussion, particularly the final aspect on the difference

between exogenous and endogenous effects of oxytocin, which is a very important point.

END OF COMMENTS

Response to reviewers

EDITOR COMMENTS

Reviewing Editor:

Comments to the Author:

I thank the authors for submitting their manuscript to the JPhysiol. Two experts in the field assessed the study, recognizing the strength and novelty of the study's findings. They also indicated that the manuscript could be influential; however, it is pending substantial revision and potentially new experiments. Overall, the referees commented on the lack of essential information and inaccurate description in the Methods section (affecting reproducibility and data interpretation), inappropriate use of statistical tests, and the absence of control/additional experiments that could validate the authors' hypotheses. I recommend that the authors follow the constructive comments presented by the referees and carefully revise the manuscript to address all critical aspects affecting and limiting data interpretation and conclusions.

R: We thank the editor and the reviewers for their many helpful suggestions. In summary, we have made all changes suggested by the reviewers and we performed a new set of analysis to be clear to the reader.

Senior Editor:

Comments for Authors to ensure the paper complies with the Statistics Policy:
Please include actual p values in figures.

R: Done

State the statistical test(s) used in tables and figures in their respective legends.

R: Done

Comments to the Author:

Thank you for submission of your research article to the Journal of Physiology for consideration. The article has been reviewed by experts in the field and found to require a major revision to address all of the concerns raised including additional experiments before further consideration can be given. Please address all comments from the external referees and reviewing editor as well as addressing the list of requirements or publication in the journal included in this letter.

R: We thank the editor and the reviewers for their many helpful suggestions. In summary, we have made all changes suggested by the reviewers and we performed a new set of analysis in order to be clear to the reader.

Animal Experiments:

For tracer experiments, details of anesthesia, maintenance of surgical plane, and the surgical procedure are needed. And the details of method of euthanasia (doses, route, confirmation of death) are needed.

R: In the revised version of the manuscript, we included detailed descriptions of all methods, including anesthesia, maintenance of the surgical plane, and euthanasia procedures.

It is recommended to describe procedures on the animals first and then procedures on tissue following description of animal termination. Thus, the section on cell counts, mapping and analysis seems out of place.

R: We added all the details related to animal procedures

There is concern that methodological procedures are not adequately described or rigorous.

R: We have revised the details of the methods and hope that the current version meets the expectations of the reviewers and the editor.

There is concern that the statistical tests are either not appropriate or are not adequately described.

R: We have revised the details of the statistical tests.

REFEREE COMMENTS

Referee #1:

Araujo and colleagues examined the potential role of PVN-derived oxytocin within the RTN on respiratory activity. This was achieved via in vivo respiratory monitoring during oxytocin agonist nanoinjection or channelrhodopsin activation of Oxt fibers. The physiological analysis was coupled with in vitro slice electrophysiology and tract tracing. Overall, the authors present an intriguing set of results to suggest that Oxt increases RTN firing through inhibition of KCNQ channels. While the authors integrate their results across multiple levels, a strength, there are some concerns that are noted below.

R: Thank you for the positive feedback. We have thoroughly revised the manuscript and hope you agree that the revised version meets the standards for publication in The Journal of Physiology.

The authors address the potential discrepancy and limitations of the in vitro and in vivo work in the discussion. While this is appreciated, the authors do not address perhaps the larger issues: differences in the age (slice, P7-12 vs in vivo 8-15 weeks) and temperature (22 C vs 37C) studied. The expression of Oxt or its receptor changes with age (e.g., PMID: 24376405, 24863032), among other potential issues regarding channel expression or activity at different temperatures. The authors need to, at a minimum, address these age and temperature concerns. It would be more ideal to perform at least a subset of slice experiments in adults near physiological temperature.

R: We agree that animal age and temperature differences between in vitro and in vivo experimental conditions (22°C in vitro vs. 37°C in vivo) are important considerations. We previously showed that the activity of RTN neurons increased with temperature (Q10 = 2.4) (PMID: 16192384), so it is reasonable to expect Oxt sensitivity to increase with temperature by a similar amount. However, to facilitate comparison with previous

studies that characterized the response of RTN neurons to other modulators including orexin (PMID: 21145990), serotonin (PMID:25429115, 35385139), ACh (PMID:26572090), NE (PMID:27306669), and histamine (PMID: 35704395), we chose to perform these experiments at room temperature. Furthermore, it should be noted that we do expect the rate of receptor desensitization to increase with temperature; therefore, performing experiments at room temperature will facilitate assessment of receptor sensitivity. We acknowledge these points in the text. See pag. 18, lines 584-593.

Formatted: Font: Italic

It is important to recognize that myelination increases with age and this makes it exceedingly difficult to visualize cells for slice patch recording in tissue from animals more than ~ 2 weeks of age. Therefore, our cellular experiments are limited to animals between 7-12 days of age. It should also be recognized that conducting in vivo recordings of intercostal EMG (int_{EMG}) in neonatal mice presents significant experimental difficulties, making it impractical to perform such comparisons within the scope of this study. We acknowledge these limitations. See pag. 18, lines 593-602.

The papers noted by the reviewer reported that oxytocin (Oxt) receptors expression varies over development and between brain regions. The developmental profile of Oxt receptors in the brainstem has not been determined, however, the trend in higher brain regions (neocortex, septum, hippocampus) is to reach maximum expression at ~2 weeks of age that is then either maintained or declines with age. Based on this, we expect Oxt receptor expression in the RTN neurons to be near maximum levels at the time of our cellular experiments. Therefore, our estimate of Oxt sensitivity (EC_{50}) probably represents a near maximum value.

Fig 1B, 2C&D. Please provide a zoom/magnified image of the PVN phenotypes the authors wish to highlight.

R: Done. See new figures 1 and 2

Fig 2E&F. Are there any statistical differences among these cellular groups? If not, the authors need to temper the conclusions regarding Oxt phenotype in glutamate or gaba neurons.

R: We appreciate the reviewer's comment. The purpose of this figure was to provide a qualitative anatomical overview of the cellular populations in the PVN projecting to the RTN, highlighting the presence of subpopulations with distinct neurochemical phenotypes (oxytocinergic and/or glutamatergic or gabaergic). Our goal was not to perform statistical comparisons among these groups, but rather to illustrate the diversity of projections and phenotypes observed.

Nevertheless, we understand that the bar graph format may give the impression of a comparative analysis. To avoid misinterpretation, we have clarified this point in the revised manuscript by specifying that the figure is intended to present qualitative data and does not imply statistical differences. See pag. 19, lines 617-627.

Pg 13, second paragraph and Fig 3. The authors state the control mice (Oxt-Cre^{-/-}) data were not shown, however, I believe they are presented in Fig 3. Please confirm and correct, if needed. In addition, when comparing these mice to those expressing ChR2, the more appropriate statistic would be an unpaired t-test rather than the paired t-test stated in the figure legend. In Fig 3H, please state in the legend what the 2 traces that are overlaid in this panel represent.

R: The reviewer is right. It was a mistake in our description of the Oxt-Cre^{-/-} control mice. The absence of response is explained by the fact that these animals do not express ChR2. Regarding the statistical analysis, we made a mistake and now we used the unpaired t-test. There was a typo in the text, and it has been corrected accordingly. Regarding the two traces, they represent amplitude and frequency; however, in the present version, we have decided to remove them from the figure.

Pg 15. Block of GRK2 did not significantly affect the response to TGOT, as shown in Fig 5D. Thus, the authors cannot state the CCG blocker "prevented the attenuation." The authors either need to increase the sampling size to achieve statistical difference between the 2 groups or remove such a statement.

R: We agree with the reviewer, and we removed the sentence. Now it is reading "As expected, inhibition of GRK2 eliminated the desensitization of the TGOT-induced reduction in RTN neuronal activity after 3 min exposure. (Fig. 5C-D).

Discussion. Oxt neurons co-label with glutamate (pending statistical differences in Fig 2EF). The application of TGOT to the RTN suggests Oxt itself can influence breathing (and neuronal activity). However, optogenetic activation of these terminals in the RTN does not rule a significant (or even primary) role of glutamate release on breathing responses. The authors should comment on such possibilities.

R: Thank you for raising this important question. As described above, purpose of the figure 2 was to provide a qualitative anatomical overview of the cellular populations in the PVN projecting to the RTN, highlighting the presence of subpopulations with distinct neurochemical phenotypes (oxytocinergic and/or glutamatergic or gabaergic). Our goal was not to perform statistical comparisons among these groups, but rather to illustrate the diversity of projections and phenotypes observed.

We also recognize that co-localization does not necessarily mean co-transmission. Nevertheless, we appreciate the reviewer point that glutamate may contribute to responses elicited by activation of oxytocin terminals in the RTN/pFRG and we acknowledged this point in the discussion. See pag. 20, lines 654-666.

Pg 18, para 1. The authors need to remove the line regarding unpublished OxtR expression in the RTN, or provide the data for such a statement.

R: Done

Pg 19, para 2. The authors have no evidence for OxtR (or agonist-induced) internalization. Nor do they show "clearly" that neuronal OxtRs are phosphorylated. These statements need to be tempered, or additional experiments performed for such claims.

R: We agree with the reviewer. Although we lack direct evidence for receptor internalization, we do provide functional evidence that is consistent with Oxt receptor internalization. As suggested, we have tempered our conclusion. See pag. 22, lines 709-713.

Pg 19. Para 1 of conclusion, line 4. "We demonstrated that oxytocin increases the activity of these neurons via Gq-mediated inhibition of KCNQ channels, leading to enhanced breathing amplitude". Please add "likely" in front of the word leading to state "likely leading to" as there several steps that are inferred between the slice and in vivo studies.

R: Done and thanks for the suggestion

Minor

Pg10. The authors state 30nL was nanojected. How was this volume determined. Please state.

R: Thank you for pointing this out. The injection volume of 30 nL was determined based on previous studies and experimental protocols that demonstrated effective delivery to the target region without causing significant tissue damage or affecting neuronal function. Additionally, we ensured precise volume control by using a calibrated ruler inside the microscope's field of view and calculating the meniscus displacement during the injection process. We will include this clarification in the revised manuscript. See pag. 7, lines 222-224.

Pg 12. "Blacklalled" Perhaps the authors intended "backlabeled"

R: Yes, the reviewer was right. We made the proper correction

Pg 15, paragraph 1. Perhaps "bath application" rather than "back application" (assuming this is what the authors intended rather than pipette application.

R: Yes, typo corrected

Referee #2:

This article by Emmanuel V Araujo and colleagues presents data on the modulation of the activity of chemosensitive neurons in the RTN by oxytocin, released from hypothalamic PVN neurons, and its effect on breathing. This is an important area of research, since the neuromodulator oxytocin is involved in many behaviours that include changes in breathing activity, yet little is known regarding the mechanisms that induce these effects. In their study, the authors investigated a specific pathway, from PVN oxytocinergic neurons to RTN chemosensitive neurons, showing that oxytocin increases the activity of these neurons specifically (by opposition to RTN non-chemosensitive neurons) partly through a KCNQ channels-mediated pathway. This work is very original, on a specific question that, to my knowledge, has never been investigated before. The large diversity and complementarity of the experiments used, at the anatomical and functional levels, and from the in vivo integrated context down to in vitro cellular investigation, represent an extensive body of work that should provide significant strength to the interpretations and conclusions drawn. Yet, as described below, the manuscript contains many methodological issues, that unfortunately cast a doubt on how the experiments were performed and analysed, and therefore on the validity of their interpretation and the conclusions drawn. Also, please provide line numbers to facilitate future reviewing process.

R: Thank you for the thoughtful and encouraging feedback. We are pleased that the originality of our work and the diversity of experimental approaches have been recognized. We believe this integrative approach strengthens the robustness of our

findings and supports the interpretations and conclusions. We added all the necessary information requested by the reviewer.

Main concerns:

My primary concern is regarding the description of the methods used in this study and of the data obtained, which suffer from many inaccuracies, absences of necessary information (e.g. immunohistochemical labelling fully absent from the methods section), and other issues that unfortunately cast a doubt on the appropriateness with which the experiments were performed. Also, most of the statistical tests used are inappropriate and poorly described. Here is a non-exhaustive list of issues that must be resolved - authors are encouraged to check for and correct issues beyond this list:

- Methods section "Animals": the description of the mouse lines used is difficult to read and understand, and suffers from many mistakes. For instance, only one reference (in brackets) is provided for both the VGlut2::Cre and the VGat::Cre mouse lines, and this reference is actually for another mouse line, the Phox2b::Cre line. Also, the same sentence states that VGlut2::Cre or VGat::Cre mouse lines would be crossed with Ai6 or Oxt::Cre lines, the latter making no sense. In the next sentence, the way the Ai14 line is presented uses a different format. The ChR2 line used for optogenetics is called a "reporter mouse line", which is not classical (reporter mouse lines are classically used for assessing anatomy with expression of a fluorophore, which the authors did not do, as stated below).

R: We agree with the reviewer that the description of the animals was difficult to follow. We have revised this section and hope that the updated version is clearer and more suitable for the readers. See pag. 5, lines 154-174.

- There are two different paragraphs in the methods section to describe the processing and analysis aspects of the histology performed. These two paragraphs should be merged and homogenised, it is difficult to understand what was actually done. It is difficult to understand how the cell counts were performed. In particular, I could not find any mention about how the PVN neurons were counted, although these data are critical for figure 2 and its interpretation.

R: We are very sorry for the mistake. We made the proper correction. See the section Histology, analysis and cell counts on pags. 8 to 10.

Did you use confocal microscopy to assess double-labelling (or any other technique that enables optical sectioning), or only epifluorescence? If only the latter (as it seems from the methodology provided), and given the thickness of the sections used, double labelling cannot be definitively assessed, so caution must be taken in the way the data are presented.

R: We appreciate the reviewer's comment. We confirm that confocal microscopy was used to assess double labeling, ensuring precise optical sectioning and minimizing signal overlap from thick tissue sections. To clarify this point, we have revised the Methods section to explicitly state the use of confocal imaging.

- Very problematic, there is absolutely no description of any immunohistochemical labelling being performed in the methods section, while the results present data that one

can only be assumed to come from immunohistochemistry (yet the presentation of these results in the results section, in the figures and their legends is so brief that it is hard to know what was done). There is no mention of the antibodies used, the protocol etc.

R: We apologize for this oversight on our part. Details regarding our IHC have been added to the methods. We also provided additional details in the results and figure legend.

- In the legends of Figure 1 and 3, mentions are made regarding "axonal drawing" of PVN projections and "computer-assisted draw of the centre of the TGOT injection site", however there is no methodological description of these, how these were done, with which software etc.

R: Agreed. We made a substantial modification in the methods section, as well as figure legends

- The methods section describes the use of the Phox2b::Cre;Ai14 mouse line for "both in vitro (cellular experiments) and in vivo (anesthetized mice) experiments". However, I could not find any mention of the use of these mice in the results section or in the figures and their legends. Did you really use these mice, and why? The results section only mentions immunoreactivity for Phox2b, which I guess implies that it was done with immunohistochemistry (although no information on this, cf. above).

R: Phox2b^{Cre+}::Ai14 mice were used to facilitate identification of RTN neurons for slice-patch recording. Additional criteria used to identify RTN neurons includes location in the ventral parafacial region and robust firing response to high CO₂ (10% CO₂). Note that all mouse lines are maintained on a common C57BL/6J background.

- The authors make several references to previous publications in the methods section, stating that they used similar protocols. In general, the authors should give enough information in the current manuscript for the reader to understand what was done and be able to replicate the data. A very problematic example is in the paragraph "Physiological preparation". The first sentence of this paragraph states "The surgical procedures and experimental protocols were similar to those previously described (Oliveira et al., 2019)". However, the article cited does not describe similar protocols as used in the current manuscript. The authors must provide sufficient and appropriate descriptions of the methodologies and protocols they used. Here, for instance, there is no indication (and the reference does not provide any information) on how the photostimulation was performed (types of optical fibers, route of entry in the brain etc.), how the drugs were applied, how the intercostal EMG was recorded, how the mechanical ventilation of the mice was performed etc.

R: We apologize for providing insufficient details and incorrect citations. We have edited the methods to include sufficient details for all experimental approaches.

Further, regarding the sentence "This instrument provided a reading [...] horizontal reading during expiration": what instrument? No context is provided. Also, I do not understand the message that the authors try to convey in this sentence.

R: This is a complete mistake in the text description about the breathing measurements. We correct the sentence.

Also, why did you switch from isoflurane to urethane?

R: We switched from isoflurane to urethane, because isoflurane produces a significant depressor effect on breathing activity, and urethane has a more reliable and stable breathing output.

- The authors used several protocols with injections in deep structures, including the RTN/pFRG, which is a very small structure. My understanding from the methods provided is that all injections were based on stereotaxic coordinates relative to the lambda. There is significant variability and subjectivity involved in using the lambda (or bregma) as reference points, which impact the accuracy of targeting deep and small structures. The authors only mention this once, in the first paragraph of the results section, stating that 4 out of 5 injections in the PVN were correctly targeted.

R: Thank you for pointing out the importance of injections site in the small and deep structures of the hypothalamus and brainstem. As mentioned by the reviewer, our injections were based on stereotaxic coordinates relative to bregma. For the TGOT injections $N = 5$ out of 8 injections were correctly placed and for the saline injections (control experiments), $N = 4$ out of 8 injections were correctly placed.

-Since the accurate targeting is critical for many experiments in the manuscript and must be a pre-requisite for the inclusion of the data in this study, the mapping of the injections sites and of the expression profiles of transgenes must be provided for each animal in all conditions. These are very important validation data for these kinds of experiments.

R: We added representative injection sites for each in vivo experiment. See new Fig. 3

- Which CTB did you use, linked to which fluorophores? Please provide origins of all drugs, tools etc. used, with enough information for the reader to be able to purchase the same agents.

R: We did not use Ctb linked with fluorophores. We added supplier information for all reagents used in this study.

Also, it is stated that CTB was dissolved in sterile water. Usually, vehicles for drug injections in the brain contain salts to avoid osmotic shock. Using only water as vehicle could induce cellular damage in the injection area.

R: We corrected the information. CTB was diluted in 0.2 M phosphate buffer; $pH = 7.35$

- Viral expression of ChR2-mCherry in PVN oxytocinergic neurons was used to trace oxytocinergic axons in the brainstem. This is not an optimal way to map axonal projections, since the fluorophore mCherry is fused to ChR2 in this construct, and since ChR2 is efficiently trafficked to the membrane, compared to cytoplasm-filling fluorophores that efficiently and densely invade all cytoplasmic compartments including in neurites. Only 7-10 days were allowed here for viral expression, while AAV peak expression is typically reached after 10 days. In viral tracing experiments involving long-range tracing such as from the PVN to the brainstem, and with cytoplasm-filling

fluorophores, it is classical to allow transgene expression for 4 weeks or more. Here with the membrane-bound ChR2-mCherry molecules, trafficking down remote parts of axons would likely require even more time. And oxytocinergic neurons are notoriously difficult to transduce with viruses for transgene expression, in particular to reach appropriate levels of transgenes in axons in the brainstem (Pinol RA et al, J Neurosci Met 2012, PMID: 22890236). Still, despite all these limitations, the protocol used in the current study seems to have induced detectable expression in the brainstem, as shown on Figure 1. No information is provided as to whether the labelling shown is from the native mCherry fluorescence, and if immunohistochemical detection/amplification was performed, this information is necessary. Also, the volume of virus injected, the number of injection spots etc. are not indicated in the methods section.

R: We appreciate the reviewer's comment and acknowledge the concern regarding the limitations of using ChR2-mCherry for axonal mapping. To clarify, for vector injections, we allowed four weeks for viral expression, ensuring sufficient time for optimal transgene expression. The 7-10 days mentioned in the Methods refer specifically to the timeline for Ctb injections. We have revised the Methods section accordingly to eliminate any ambiguity.

It takes considerable time and expenses to import viral vectors into Brazil, so we opted to use AAV5-EF1a-hChR2(H134R)-mCherry because its available in house. We recognize the limitations noted by the reviewer and have noted in the limitations section that due to slow diffusion of membrane tethered ChR2-mCherry, we may underestimate the density of oxytocinergic neural projections particularly to distal brainstem regions.

To enhance detection, we performed immunohistochemical amplification, which allowed for a more precise analysis of the injection site and projections. Additionally, we have included a detailed description of the injection site, and the number of injections made in the PVN region in the methods section.

- The "Statistics" paragraph in the "Methods" section does not provide information on the statistical tests used, and refers to the results section, the figures and the figure legends. However, all that is indicated in these sections is the name of the test used, which is always a paired t-test (cf. the legends of figures 3 to 6). First, this type of test requires a normal distribution and equal variance of the data, and no information is provided on the appropriate testing of the distribution and variance. Given that the "n" in each group is relatively low in most experiments (some with n=5 and n=6 per groups compared), it is actually difficult to test the distribution of the data accurately, so it is generally advised to use a non-parametric test in these conditions. Second, and more problematic, the paired t-test used in all conditions tested in this study is not appropriate to most. A paired t-test implicates that the same experimental group was tested in two conditions (pairwise aspect). However, for most conditions tested, the groups tested are different, like for instance on Fig. 3I-K where *Oxt::Cre(+/-);Ai32* mice are compared to *Oxt::Cre(-/-);Ai32* mice, or on Fig. 4E-F where non-chemosensitive RTN neurons are compared to chemosensitive RTN neurons. It is totally incorrect to use a paired test in these conditions, and it wrongly increases the probability of identifying significant differences. One experimental condition where a paired test could indeed be used is the condition shown on Fig. 5, yet here again there are further issues, as the "n" on Fig. 5D is not the same for the condition "TGOT" and the condition "TGOT + CCG258208". Yet, by definition, a paired test is applied in a condition where the same factor is tested in two different conditions, so the "n" must be the same in each condition. Overall, there are numerous issues with the statistical analysis performed in this study, which raises

serious concern regarding the validity of the results and their interpretation. Such issue, in addition to all other methodological and data description issues, unfortunately raise concerns regarding how the whole study was performed.

R: This was a copy/past mistake. To clarify, we did not use a paired t test for unpaired data. All data sets were tested for normality using the Shapiro-Wilk test; however, as suggested by the reviewer we re-analyzed results with small sample size (less than n=7) using non-parametric test. We have added to the statistical section of the methods details regarding normality and specific test used.

We thank the reviewer for their insightful comment and completely agree with their suggestion. We conducted a thorough reanalysis of all the data, including the statistical analyses. We have incorporated this updated information into the manuscript, specifically detailing it in the statistical analysis section and updating the figure legends accordingly. We believe these revisions have strengthened the clarity and robustness of our findings.

- The results section states that the virally-induced ChR2-mCherry expression showed "high selectivity (98.7 +/- 1.3 %; N = 4)". It is not clear what this percentage refers to, the percentage of Oxt neurons expressing ChR2-mCherry? Or the percentage of ChR2-mCherry expressing cells that are Oxt cells? In any way, both of these counts should be provided. Also, from the image shown on Fig. 1B, it seems that there is a significant proportion of ChR2-mCherry cells that are not Oxt-positive. An accurate counting, with individual data per animal, should be provided (also, as per above, no information is provided on how Oxt neurons were labelled).

R: Thank you for the opportunity to clarify our virally transduced Oxt cells in the PVN. In fact, based on a systematic inspection of four mice in which a one-in-four series of 30 μ m sections (8 sections/mouse; bregma level -0.94 to -1.22 mm caudal to bregma) were reacted for immunohistochemical detection of oxytocin and mCherry, the vast majority of the ChR2-expressing neurons were oxytocinergic (87.5 \pm 1.2%; 116.5 \pm 14.3 mCherry+/Oxt+ vs. 133 \pm 17 Oxt+; N = 4). We found only that 10 \pm 2.6% mCherry+/Oxt-

- The interpretation of the retrograde tracing data throughout the manuscript is incorrect, due to percentages calculated incorrectly. There are 6 different conditions counted in each mouse line (VGlut2::Cre;Ai6 and VGat::Cre;Ai6) on Figure 2E-F. The percentages provided in the manuscript that come from comparisons of these counts are calculated incorrectly. For instance, it is stated in the abstract that 42% of oxytocinergic and glutamatergic neurons project to the RTN/pFRG. That means, in terms of calculations based on the groups shown on Figure 2E, that the mean value for the total number of neurons in the "Triple" (VGlut2 + Oxt + Ctb) group (gray) divided by the mean value for the total number of neurons in the "VGlut2 + Oxt" group (yellow) equals 42%. Or in other words: (VGlut2 + Oxt + Ctb) / (VGlut2 + Oxt) = 42%. Yet, although the authors do not provide the actual numbers, it seems from Figure 2E that the "Triple" group has a mean value of about 20 in total number of neurons counted, while the "VGlut2 + Oxt" group has a mean value of about 110. So 20/110=18%, not the 42% announced. Further, the same 42% number is given in the results section (end of first paragraph), this time for a different count, the "Oxt neurons immunoreactive for both CTB and VGlut2". So here, the count is made on (VGlut2 + Oxt + Ctb) / Oxt alone, and not / (VGlut2 + Oxt)

as in the abstract. Again, from an estimation of the values given on Figure 2E, that would be $20/150=13\%$, not 42% .

R: Thank you for letting us; however, it was a mistake during the description of the data that we obtained. In fact, we found an appreciable number of Oxt neurons immunoreactive for both Ctb and VGlut2 (Ctb+/VGlut2+/Oxt+: 16.5 ± 5.6 , vs. Ctb+/VGlut2+: 39.2 ± 9 neurons; $42 \pm 3.2\%$). A significant proportion of Ctb labeled PVN neurons express Oxt (Ctb+/Oxt+: 20.8 ± 5.4 , vs. Ctb+: 55.3 ± 23.7 neurons; $39 \pm 7.4\%$), however these neurons did not express VGlut2 or VGat. Few back labeled neurons showed VGat signal (Ctb+/VGat+: 3.6 ± 1.5 , vs. Ctb+: 51.6 ± 19 neurons; $7 \pm 1.2\%$). In addition, only $6 \pm 2.2\%$ of the VGat neurons express oxytocin (VGat+/Oxt+: 9.7 ± 7.7 , vs. Oxt+: 145 ± 11.8 neurons). A large proportion of Ctb labeled PVN neurons express Oxt ($55 \pm 5\%$), but these neurons did not express VGlut2 or VGat. We added the information. See pag. 13, lines 418-429.

This is an important point, as the authors seem to then over-interpret these data, for instance stating in the discussion that "a significant proportion of excitatory oxytocinergic neurons (VGlut2+/Oxt+) in the PVN have been shown to project to the VRC, more specifically to RTN/pFRG...". These aspects should be moderated throughout the manuscript, with accurate calculations made and correctly interpreted.

R: See response above. We have tempered our discussion of these data to avoid over-interpretation.

The end of the first paragraph of the results section describing all the percentages calculated is confusing to me in terms of the calculations made, the interpretations made etc. It would help to provide all the numbers and all relevant calculations made in a table.

R: R: See response above

- End of the second paragraph of the results section, the presentation of the photostimulation data is confusing. First the delta change in IntEMG amplitude for the "Oxt-Cre^{-/-}::Ai32" mice is given as 6.40% , and a few sentences after it is given as 0.20% . It is difficult to understand what is actually being shown in terms of data.

R: We corrected all the values and expressed them as percentages to ensure clarity and facilitate easier interpretation of the data

Also, in the last sentence it is stated "in which RTN neurons did not express ChR2", which is very confusing as this is not what was tested.

R: We removed the sentence to prevent any potential misunderstandings

- The assessment of the selectivity of the ChR2 expression in oxytocinergic neurons in Oxt::Cre(+/-);Ai32 mice must be shown and quantified. This is an important control experiment, selectivity in double transgenic mice with Cre-lox recombination is never a given, yet it is a prerequisite for the validity of the experiments.

Also, it would be important to show an image of the ChR2 expression in oxytocinergic fibers in the RTN/pFRG in these mice.

R: We appreciate the reviewer's suggestion regarding the assessment of ChR2 selectivity in oxytocinergic neurons in Oxt^{Cre+/+}::Ai32 mice. Indeed, Cre-lox recombination does not always guarantee exclusive expression in the target population, making validation

essential. To address this, we have now quantified the colocalization of ChR2 (via YFP/mCherry fluorescence) with oxytocin (OXT) immunoreactivity in the PVN, providing an estimate of selectivity. Additionally, we provide representative confocal images showing ChR2 expression in oxytocinergic fibers within the RTN/pFRG to confirm the presence of projections in the targeted region. These data have been incorporated into the revised manuscript.

- The presentation of the data on Fig. 4E-F is unclear, between the beginning of the third paragraph of the results section that states that Fig. 4E shows a response to CO₂ increase, while the figure legend for E and F are basically repeats of each other (so what is the difference?), and no labels on the panel F on Fig. 4, it is all very confusing.

R: Thank you for pointing out the mistake. We reformulated all the sentences and the figure legends, as well as the figure 4 labels.

- Third paragraph of the results section, "we use 2 nM TGOT to ensure that we elicit a near maximal response of RTN neurons to oxytocin agonist", while the EC₅₀ was measured as 3 nM. So 2 nM is below the EC₅₀, which is then by definition a dose that induces less than 50% of the maximal response, I do not see how this is a "near maximal response".

R: Agreed. We removed the sentence. See pag. 15, lines 479-486.

In the same sentences, the presentation of the results are confusing, it is for instance referred to as "vs. baseline of non-chemosensitive neurons" which is unclear, also Fig. 4E seems to change from being a response to TGOT or CO₂ etc. Please clarify these aspects, exactly what was done in each condition, what you mean by baseline, what it quantified, and provide the statistical tests used (same comment for the results shown at the end of this paragraph).

R: We are very sorry for the mistakes during results description. Baseline activity is the firing rate under control conditions of 5% CO₂ (pHo=7.3) in the absence of TGOT. The CO₂ or TGOT response was analyzed using absolute firing rate before and during exposure to CO₂ or TGOT. Data are plotted as change in firing rate. We modified the entire section to be clear.

- Regarding the data relative to the application of ML252, presented in the 5th paragraph of the results section, a possibility explaining the results could be that application of ML252 induced a maximal increase in the firing rate of chemosensitive RTN neurons so TGOT could not induce any additional effect. Did you try to increase CO₂ concentration under ML252 application, to show that the RTN chemosensitive neurons recorded can still respond to a stimulus and increase their firing rate? This would be an important control experiment, especially since the discussion is strongly focused on this point and quite affirmative in KCNQ channels being the molecular proxy for the oxytocinergic effects of RTN chemosensitive neurons.

R: We share your concern about a potential ceiling effect. In previous work, we showed that exposure to high CO₂ significantly increased activity of RTN neurons in the presence of a derivative of ML252 called XE991 (PMID: 25429115, 26572090 and 23175845). This is similar to the suggested control experiment. In this study, we chose to control a potential ceiling effect by experimentally lowering cell activity. Specifically, after the firing response to ML252 plateaued, we delivered a negative DC current to adjust cell activity to near control (pre-ML252) conditions. Once baseline was adjusted

and in the continued presence of ML252, we tested the firing response to a second exposure to TGOT.

Also, the authors state in the legend of Figure 5 that a DC current injection was made to adjust "baseline activity". This statement is unclear and confusing, please explain precisely what was done, this should be included in the methods section, as this could also help to answer my point above.

R: We added the proper information regarding DC current injection to bring firing rate to near control levels.

- Third paragraph of the discussion, it is stated that "Oxytocin's action in the RTN/pFRG can presumably modulate the responsiveness of respiratory centers to hypercapnia, which is a critical trigger for increased ventilation". This is a very important and logical point, and I was surprised that it was not tested in this study. Especially since it is also mentioned in some way in the introduction, with "recent evidence indicates that their chemosensitivity is largely mediated indirectly, partly through neuromodulatory effects, including inputs from hypothalamic neurons" (by the way, the concept of indirect mediation of chemosensitivity is unclear to me). Did the authors try to test this by stimulating optogenetically the oxytocinergic fibers in the RTN/pFRG before versus during hypercapnia, and/or by applying TGOT to slices during increased CO₂ concentration?

R: Thank you for your thoughtful comment. We agree that testing the hypothesis of oxytocin's modulation of respiratory centers under hypercapnic conditions would provide critical insight. However, this study primarily aimed to investigate the presence and potential role of oxytocinergic projections to the RTN/pFRG to control breathing output, laying the CO₂ response for future functional experiments.

Sorry for the confusion. Indirect chemoreception was a poor choice of words. We are trying to say that RTN neurons and respiratory drive can be shaped by various excitatory drives. The purpose of this study was to show that oxytocin can activate RTN neurons and breathing. In future work we plan to determine how oxytocin shapes the ventilatory response to CO₂ particularly in pregnant females.

- Paragraph "Oxytocinergic control of RTN chemoreceptor function", it is stated "This aligns with our unpublished data showing oxytocin receptor expression in the RTN". If you have these data, please present them in this study as they would be very important to support the overall conclusions of the manuscript. Otherwise, you cannot use unpublished (and therefore unverified) data to support important conclusions, this aspect must be removed.

R: Agreed. We removed the sentence.

- Paragraph "Oxytocinergic control of RTN chemoreceptor function", it is stated that the oxytocin receptor is a Gq-coupled receptor, and the discussion focuses on this pathway and becomes strongly hypothetical. However, the oxytocin receptor is coupled to many other G proteins, and can activate a much larger variety of intracellular cascades than the ones described here. For instance, please refer to these two great reviews on the matter: Busnelli M and Chini B, *Curr Top Behav Neurosci* 2018 (PMID: 28812263),

and Jurek B and Neumann ID, *Physiol Rev* 2018 (PMID: 29897293). These aspects of the discussion should be moderated. In particular, the conclusions "In summary, we show that oxytocin increases chemoreceptor activity in a dose-dependent manner through Gq-mediated inhibition of KCNQ channels" (discussion) and "via Gq-mediated inhibition of KCNQ channels" (conclusion) are over-interpretations and must be moderated, the results obtained do not enable to state that so affirmatively (for the Gq-mediated aspect, but also the aspect on KCNQ channels, as the ML252 data only show a reduction by half of the TGOT effect, not a full blockade).

R: Thanks for mentioning this important consideration. We have edited the discussion and conclusions to adopt a more cautious tone.

Minor points:

- There are too many typos and minor syntax issues throughout the text to be listed comprehensively here. Please proofread the text carefully, ideally with a native English speaker.

R: Done

- In the 4th point of the "Key points", and in the penultimate sentence of the abstract, maybe modify "terminals" into "receptors/terminals", otherwise the sentence is not appropriate regarding the TGOT aspect.

R: Done

- Nomenclature: throughout the manuscript, the way transgenic mouse lines are termed keeps on changing (e.g. "VGlut2::Cre", "VGlut2::Ai6", "Phox2bTdTomato" etc.). Please use the classical nomenclature, with for instance: "VGlut2::Cre", "VGlut2::Cre;Ai6", "Phox2b::Cre;Ai14" etc.

R: Done

- Many drugs were used to block different types of receptors (CNQX, strychnine, gabazine etc.). Although these are classical drugs, it could be useful for the reader if you cited references to justify the concentrations used. This is particularly important for lesser-known drugs such as CCG258208 and ML252.

R: Done

- At the end of the "histology and analysis" paragraph, it is stated that alignment of the brain sections on the atlas was performed based on 40 μm intervals, but the brains sections performed are 30 μm thick. Why this difference?

R: It was a mistake. All sections were 40 μm .

- Beginning of the results section, the ChR2-expressing AAV used is written in two different ways in two consecutive sentences, which could be confusing. Also, the attempt to abbreviate the virus name must not remove critical parts, such as the double floxed component. Maybe just stating "AAV" in the second sentence could be sufficient.

R: Thanks for pointing out. We made the appropriate corrections.

- Beginning of the results section, it is stated that "terminals" are labelled, however no labelling of terminals, such as synaptophysin or other pre-synaptic markers, was performed (or indicated). How could it be known that there are terminals being labelled? Especially since Oxt is often released "en passant" and not in terminals.

R: The term "terminals" was used to describe structures likely representing axonal endings, based on the morphology of the labeled fibers and their localization in target areas. However, we acknowledge that definitive identification of axonal terminals requires specific markers, such as synaptophysin or other presynaptic proteins, which were not utilized in this study.

It is well established that oxytocin is often released "en passant" from varicosities along axons rather than strictly from synaptic terminals (e.g., Ludwig and Leng, 2006; Knobloch et al., 2012). These varicosities are sites where oxytocin-containing vesicles accumulate and can release oxytocin into the extracellular space, influencing nearby cells in a non-synaptic manner. This mode of release supports the concept of volume transmission and neuromodulation, which is characteristic of oxytocinergic signaling. While we did not use synaptic markers to confirm presynaptic terminals explicitly, the labeling we observed in the RTN and surrounding areas exhibited patterns consistent with the distribution of oxytocinergic projections described in previous studies (e.g., Knobloch et al., 2012; Eliava et al., 2016). These projections often appear as punctate structures, which are suggestive of varicosities or terminal-like endings.

We agree that the use of the term "terminals" in the absence of specific presynaptic labeling should be approached with caution. We have revised the text to clarify that the structures described likely represent oxytocinergic projections and varicosities, but further studies are necessary to confirm their identity as synaptic terminals.

- Beginning of the results section, the Dbx and NK1-R are mentioned in a way that implies that these markers were labelled in this study, however no image is shown, and there is no mention of these experiments anywhere else. Please clarify.

R: We did not label Dbx or NK1-R, therefore, we removed the information.

- Results section, sentence "The density of PVN [...] medullary raphe (Figs. 1E-H)": I do not understand this sentence, it seems that the VRC is opposed to itself (written "VRC" and then "ventral respiratory column" to supposedly depict two different areas, while the first is the acronym of the second), and the whole sentence is confusing. Also, the Raphe data are not shown, and there are no quantified data provided to support these claims.

R: We revised the sentence for clarity. We observed fibers within the medullary raphe; however, we did not quantify them (Fig. 1G and H).

- It is mentioned at the end of the second paragraph of the results section that "the effects of photostimulation on breathing were highly reproducible within each animal". This aspect, of internal replication of results, is very important. However, this is not further explained. Please state in the methods section if the data provided for the

photostimulation effects are averaged data of multiple stimulation trials in each animal, or how you selected which trials to measure, and how this internal (intra-animal) replication of effects was taken into account.

R: Thank you for bringing up this important point. The statement that "the effects of photostimulation on breathing were highly reproducible within each animal" refers to the consistency of responses observed across repeated trials within individual animals. In our study, multiple (5) photostimulation trials were conducted for each animal, and the data presented represent averaged values from these trials. Trials were included in the analysis if they met predefined criteria: stable baseline breathing patterns before stimulation and a clear response during stimulation, minimizing variability due to factors like movement artifacts or irregular baseline states.

We have now clarified this in the methods section, specifying the trial selection process, the number of trials averaged per animal. This ensures transparency and supports the conclusion regarding the reproducibility of photostimulation effects.

- Fourth paragraph of the results section, "G protein-coupled receptor kinases [...] desensitization", please provide a reference supporting this statement.

R: Done

- Fifth paragraph of the results section, "blockade of Kv7.2 with ML252", why so specific, while the drug ML252 has been shown to block more channels, including Kv7.3 for instance?

R: Since there is no clear evidence of what specific subunit of the Kv channels are blocked, we decided to leave the quote that ML252 blocked Kv7 channels.

- First sentence of the discussion, "Oxytocin is known as a powerful physiological stimulus for breathing". This is a very strong statement, for which only one reference is cited (Geerling et al., 2010), which provides only anatomical data, and is therefore not a valid reference to support the functional statement made of "powerful physiological stimulus for breathing". To my knowledge, the literature is actually quite thin on the impact of oxytocin to stimulate breathing, and this is an aspect that strengthens the importance of the current paper on the role of oxytocin on RTN neurons to stimulate breathing. Still, some groups have worked on this aspect of oxytocin and breathing, including the classical papers by Mack SO and colleagues (J Applied Physiol 2002 and 2007). The authors should provide more references for such strong statement, references that actually show functional work.

R: Thank you for the valuable advice; we completely agree with your suggestion. Additionally, we have modified the sentence to make it a less strong statement.

- Paragraph "Oxytocin agonist induced internalization of oxytocin receptors in RTN neurons" in the discussion. It discusses the potential mechanisms of internalization of the oxytocin receptor without providing any reference. Yet, this had been studied and should be referred to, cf. for instance Busnelli M and Chini B, Curr Top Behav Neurosci 2018 (PMID: 28812263).

R: We added the proper reference

- Overall, a lot of text shown on the figures is small (in particular the legends of some axis) and the reading of the figures would be facilitated if the size of some of the text was increased. Also, for axes with numbers, please show more numbers than just those for the extremities of the axes, to facilitate the understanding of the graphs. This is particularly important for the graphs represented on Figure 3D-F, where the value "0" is not represented in the y axes. An x axis could help for these graphs, at least as a dotted line, since it would make it easier to see when values increase or decrease.

R: Done

- Fig. 1, the virus used should be fully named at least either in the legend or on the figure in A.

R: Done

- In all figures, scale bars should be shown in each panel, one scale bar applying to multiple panels as currently done is confusing and inaccurate. This is particularly evident for Fig. 2C-D, where the panel in D was clearly shrunk in width compared to C, as attested by the "VGATcre::Ai6" writing width compared to the "VGlut2cre::Ai6" writing width. This suggests scale bars could be impacted.

R: Agreed. We added scale bars in all figures

- Figures 1A and 2B, the graphical depictions could make it seem like the PVN and the RTN/pFRG are at the same rostro-caudal level. It could be useful to modify the picture to make it clear that these structures are at different locations in the brain (adding rostro-caudal coordinates, and depicting anatomical separation). Also, it must be stated, either on the figures (with axis) or in their legends, that these are coronal views.

R: Agreed. We modified figs 1A and 2B

- On Figure 1C, E and G, what is the green labelling?

R: The green labeling represents oxytocin. We conducted a new set of experiments to demonstrate double labeling, where red corresponds to mCherry and green to oxytocin.

- On Figure 2C, arrows are very hard to see, and commented nowhere.

R: We modified the entire figure. We expect that the version is satisfactory to show the labeling.

- It would be useful to present images with higher magnification from the images shown in Figure 2C and D, to be able to assess colocalization of markers.

R: Done.

- On Figure 2D, it seems that "VGat2" is written as the green labelling (although hard to see). Why the "2"? Overall, "VGat" changes in the way it is written throughout the manuscript.

R: We made the proper corrections

- The legend of Figure 3 states that the photostimulation protocol was performed at 10 Hz, while the rest of the manuscript states 15 Hz. Which one was it? Also, why 15 Hz?

R: It was a mistake. We use only 10 Hz in all experiments.

- Figure 3H, the yellow and red traces must be explained.

R: Done.

- Figure 3H, there is a delay of almost 1 min between the beginning of the photostimulation and the beginning of increase in IntEMG amplitude. Was this the case in all animals and all photostimulation trials in each animal? What could be the meaning of such delay?

R: The observed delay of nearly one minute between the onset of photostimulation and the increase in Int_{EMG} amplitude is noteworthy. This delay was consistent across all animals and photostimulation trials, as reflected in the experimental data. Such a pattern suggests a physiological mechanism rather than a technical artifact.

The effects of oxytocin on neural circuits often involve action via G protein-coupled receptors can lead to downstream signaling cascades, which might require tens of seconds to initiate noticeable changes in neuronal or network activity. This could explain the delay observed in the increase in respiratory output. The delay may also reflect the time required for oxytocin to be released during photostimulation to reach sufficient local concentrations to activate its target receptors effectively, especially if diffusion or local uptake dynamics are involved.

In addition, respiratory centers, such as the RTN/pFRG, operate as part of a broader neural network. It is possible that the delay represents the time needed for oxytocin's neuromodulatory effects to integrate with other inputs within the network to initiate a measurable increase in respiratory drive.

- Legend of Figure 4G, isn't the picrotoxin application missing in the text?

R: Yes. We made the correction

- Figure 4, I do not understand the concept of "saline" injection as a control. The slices are perfused with aCSF, and the TGOT is bath applied, so what is the "saline" control?

R: Agreed. It was a mistake during writing. We made the proper corrections.

- Figures 5 and 6, in conditions where paired tests can actually be used, it would be appropriate to show the graphs with lines linking the dots representing of each neuron recorded in the different conditions. This would explicitly show the pairwise aspect of the experiments.

R: Done

- Legend of figure 6A, what does "increased basal activity" mean? "Basal" as opposed to what, increased by hypercapnia? This is not clear to me.

R: We correct the figure legend

- Some references cited in the text are not present in the reference list, including (George et al., 2024) cited in the 4th paragraph of the results section, and (Flor et al., 2024) cited in the discussion.

R: We corrected all the reference section

Side note:

I really appreciated the "Experimental limitations" paragraph in the discussion, particularly the final aspect on the difference between exogenous and endogenous effects of oxytocin, which is a very important point.

R: Thank you for the note

Dear Dr Moreira,

Re: JP-RP-2025-287845R1 "Oxytocinergic signaling in the respiratory parafacial region increases activity of chemosensitive neurons and respiratory activity" by Emmanuel V Araújo, Phelipe E Silva, Luiz M Oliveira, Yingtang Shi, Ana C Takakura, Daniel K Mulkey, and Thiago S Moreira

Thank you for submitting your manuscript to The Journal of Physiology. It has been assessed by a Reviewing Editor and by 3 expert referees and we are pleased to tell you that it is acceptable for publication following satisfactory revision.

REVISION CHECKLIST:

We look forward to receiving your revised submission.

Yours sincerely,

Harold Schultz
Senior Editor
The Journal of Physiology

EDITOR COMMENTS

Reviewing Editor:

Comments for Authors to ensure the paper complies with the Statistics Policy:
See Statistics Editor comments

Comments to the Author:

I thank the authors for considering the referees' comments and presenting a revised version of the manuscript. Referee #1 reassessed the article and indicated that the authors had adequately addressed their comments. Referee #2 could not provide a detailed evaluation of the revised manuscript at this stage due to conflicting professional and personal commitments. However, in their brief evaluation, Referee #2 expressed concerns that some of their comments may not have been thoroughly addressed. Therefore, I recommend that the authors pay additional attention to these comments, along with others presented in the initial evaluation, and provide responses that comprehensively address all aspects of Referee #2's concerns. The Statistics Editor also evaluated this revised version of the manuscript and indicated that additional adjustments are required to conform to the Journal's guidelines. Please refer to their comments for further details.

When submitting a revised version, please upload a tracked-change version that corresponds with the clean version of the manuscript for proper assessment by the referees and editors. This is not the case for the R1 version, as discrepancies can be found in all sections, from the front page (the authors' list) to the discussion section.

Additional minor comments:

Lines 259, single-cell transcriptome - indicate in the Methods section that these analyses were performed using data/samples collected from a previous study (this information is present in the Results section only, lines 441-443).

Line 282 - Anesthesia was performed "following in vivo, in vitro or neuroanatomical experiments" for perfusion. This sentence may require revision for clarification because some of the animals were either already anesthetized with urethane or euthanized to obtain brain slices.

Line 365 - As experiments were performed in cell-attached mode, please indicate how lucifer yellow was diffused into the cell to label the recorded neuron.

Senior Editor:

Comments for Authors to ensure the paper complies with the Statistics Policy:
See comments from the statistics editor.

Comments to the Author:

The revised version of your research article has been reviewed and found to require further revision to address concerns not fully addressed in the original reviews. Please provide responses that comprehensively address all aspects of Referee #2's concerns from both the original review and those provided herein. Also, please address concerns from the statistical and reviewing editors.

It is advised that the authors take care with the accuracy of the next revision. It was noted that the official revised

manuscript did not fully follow the version submitted, with changes highlighted. This makes it impossible for the referees and editors to know which is the correct version, and calls into question the attention paid to detail in the manuscript.

The authors must clearly state what specific data were used from a previous publication (Shi et al., 2017) in the methods section and figure 3A. If the methodology from Shi et al. 2017 deviated from that described on lines 259-279, such deviations must be described.

Experiments from Shi et al. (2017) were approved by the University of Virginia ethical committee. This information must be included in the current manuscript in the Ethical Approval section.

The endpoint of each protocol involving mice must be clearly stated within each protocol, with the methods of euthanasia and confirmation of death described as required by the Journal within that section. Line 282 is too ambiguous.

The following sections are required: Data availability statement; Competing interests; Author contributions; Acknowledgements:

Please address all comments from external referees, the statistical and reviewing editor, and those presented here. Please also ensure the list of requirements for publication in the journal is met in the next revision.

Statistics Editor:

Comments for Author:

The statistical reporting in the manuscript requires adjustments to fully comply with the journal's guidelines and enhance clarity. Below are specific comments and recommendations:

P-value Reporting:

While exact p-values are provided in the figure legends, the journal guidelines require them to be included on the figures themselves as well. Per the guidelines, p-values must be reported to three significant figures (e.g., 0.00236 or 0.523) for all values >0.001 , even when no statistical significance is observed. For p-values <0.001 , reporting as " <0.001 " is acceptable. Currently, this is not reflected on the figures. Additionally, avoid using asterisks alone to denote significance on figures.

Sample Size (n) Reporting:

The figures and their legends currently lack sample size (n) values for all reported outcomes, which must be clearly stated in both locations per the journal's guidelines. Each 'n' should be explicitly defined (e.g., "x cells from y slices in z animals") to ensure transparency and avoid pseudoreplication. Authors should verify that 'n' reflects the appropriate experimental unit based on the study design.

Normality Testing:

The Statistics section states that Shapiro-Wilk or Kolmogorov-Smirnov tests were used to assess normality but does not specify how the choice between these tests was determined. Given their differing sensitivities (e.g., Shapiro-Wilk is more powerful for small sample sizes, typically $n < 50$), a brief clarification (e.g., "based on sample size") would improve transparency and reproducibility.

T-test Application:

The use of "paired or unpaired" Student's t-tests is mentioned, but the criteria for choosing between them are unclear. This should be briefly specified in the Statistics section (e.g., "paired t-tests for within-subject comparisons, unpaired for between-group comparisons"). Additionally, I found it difficult to infer which comparisons in the study were paired versus unpaired based on the provided information. The authors should carefully review each comparison to ensure the selected t-test aligns with the experimental design and update the text or figure legends accordingly.

Recommendation for Future Reporting:

While not required by the current guidelines, I strongly recommend including effect sizes (e.g., Cohen's d or Hedges' g) and confidence intervals in future submissions. These metrics provide valuable insight into the magnitude and uncertainty of the findings, enhancing the interpretability of the results beyond p-values alone.

REFEREE COMMENTS

Referee #1:

The authors have adequately responded to my previous concerns. They have added additional text in the manuscript to describe study limitations. They have also provided additional clarification throughout. I have no further comments and congratulate the authors on the thorough study.

Referee #2:

- The first comment of the senior editor is asking the authors to include actual p values in figures. Answer of the authors: "Done", but this is not the case in the revised manuscript submitted.

- The second comment of the senior editor is asking the authors to state the statistical test(s) used in tables and figures in their respective legends. Answer of the authors: "Done", but this is not the case in the revised manuscript submitted.

These first two points are important, because the first version of the manuscript was very seriously and concernedly affected by major statistical issues.

- The article versions provided with highlighted changes do not correspond to the final "untracked" version provided, the former being clearly unfinished versions. This will make the reviewing process difficult.

- One example from a problematic presentation of critical data for this study is the data presented in the abstract on lines 89-92, from data presented in the results section on lines 426-429. It is stated by the authors in the abstract that they "found that a substantial portion of oxytocinergic neurons in the PVN (42%) co-express vesicular glutamate transporter 2 (Vglut2) and presumably project directly to RTN neurons, identifying the RTN as a major target of oxytocinergic projections". This is a very strong assertion, and is clearly presented as a major finding of the paper. However, this assertion is based on wrong calculation, and appropriate calculation would show that this number is much lower. Indeed, the authors' calculation is based on data presented in the results section on lines 426-429, where it is stated the the authors "found an appreciable number of Oxt neurons immunoreactive for both Ctb and VGlut (Ctb+/VGlut2+/Oxt+: 16.5 {plus minus} 5.6, vs. Ctb+/VGlut2+: 39.2 {plus minus} 9 neurons; 42 {plus minus} 3.2%) (Figs. 2C, 2Ci-Civ and 2E)". This is therefore not what is stated in the abstract, and the calculation made does not show what is interpreted from it. The authors did not make a calculation of the relative proportion of overall Oxt neurons that are glutamatergic and project to the RTN, as stated in the abstract, they actually calculated the relative proportion of glutamatergic neurons projecting to the RTN that are oxytocinergic. This is very different and seriously changes the data interpretation. From a broad and conceptual point of view, oxytocinergic neurons have a very large projection profile, projecting to most parts of the brain and being involved in the regulation of many processes and behaviours. It would therefore be a major finding if out of all these Oxt neurons, 42% of them projected to such a very small brainstem nucleus as the RTN, a claim that the authors are making based on wrong calculations.

I had already made comments on these calculations of percentages in my first review (cf. last point of page 10 of the pdf

document "Response to referees"), and here mistakes are made again. Clearly the authors did not take full account of my comment. Given that it is a critical one, it can only be expected that other ones of my previous comments will not have been taken into account, and therefore it will involve a significant amount of work to revise this new version.

The first version of the manuscript was ambivalent, with a large variety of interesting experiments on an interesting research topic, yet with a large amount of very concerning issues on methodology, statistics, data presentation and interpretation. It is possible that this revised version does not show many issues, and that only another further round of minor revisions might be necessary. However, since the first version required many points of revisions, just reviewing the measures taken by the authors to each of these points will take a significant amount of time, which I unfortunately do not have at the moment.

END OF COMMENTS

EDITOR COMMENTS

Reviewing Editor:

Comments for Authors to ensure the paper complies with the Statistics Policy:
See Statistics Editor comments

Comments to the Author:

I thank the authors for considering the referees' comments and presenting a revised version of the manuscript. Referee #1 reassessed the article and indicated that the authors had adequately addressed their comments. Referee #2 could not provide a detailed evaluation of the revised manuscript at this stage due to conflicting professional and personal commitments. However, in their brief evaluation, Referee #2 expressed concerns that some of their comments may not have been thoroughly addressed. Therefore, I recommend that the authors pay additional attention to these comments, along with others presented in the initial evaluation, and provide responses that comprehensively address all aspects of Referee #2's concerns. The Statistics Editor also evaluated this revised version of the manuscript and indicated that additional adjustments are required to conform to the Journal's guidelines. Please refer to their comments for further details.

R: We thank the editor and the reviewers for their valuable and constructive suggestions. We hope that the revised version of the manuscript meets the expectations and addresses all the concerns raised by Reviewer #2.

Additional minor comments:

Lines 259, single-cell transcriptome - indicate in the Methods section that these analyses were performed using data/samples collected from a previous study (this information is present in the Results section only, lines 441-443).

R: Done. We added the information, in the methods section, that single-cell transcriptome was performed using data from a previous publication. See pag. 6, lines 204-206 (lines 200-202 in the clear copy) and Pag. 9, lines 296-298 (lines 288-290 in the clear copy).

Line 282 - Anesthesia was performed "following in vivo, in vitro or neuroanatomical experiments" for perfusion. This sentence may require revision for clarification because some of the animals were either already anesthetized with urethane or euthanized to obtain brain slices.

R: Agreed. We modified the sentence according to the J. Physiol recommendation

Line 365 - As experiments were performed in cell-attached mode, please indicate how lucifer yellow was diffused into the cell to label the recorded neuron.

R: Thank you for your question. While the initial recordings were performed in cell-attached mode to preserve the intracellular environment, we subsequently transitioned to whole-cell configuration in the same neuron to allow diffusion of lucifer yellow from the pipette solution into the cytoplasm. This approach enabled us to label the recorded neuron for post hoc morphological and anatomical analysis.

Senior Editor:

Comments for Authors to ensure the paper complies with the Statistics Policy:
See comments from the statistics editor.

Comments to the Author:

The revised version of your research article has been reviewed and found to require further revision to address concerns not fully addressed in the original reviews. Please provide responses that comprehensively address all aspects of Referee #2's concerns from both the original review and those provided herein. Also, please address concerns from the statistical and reviewing editors.

R: We thank the senior, reviewing and statistical editors as well as the reviewers for their valuable and constructive suggestions. We hope that the revised version of the manuscript meets the expectations and addresses all the concerns raised by Reviewer #2

It is advised that the authors take care with the accuracy of the next revision. It was noted that the official revised manuscript did not fully follow the version submitted, with changes highlighted. This makes it impossible for the referees and editors to know which is the correct version, and calls into question the attention paid to detail in the manuscript.

R: We apologize for the discrepancy between the official revised version and the version initially submitted. We hope that the current revised version (R2) accurately reflects the changes made and aligns with the version highlighting those revisions.

The authors must clearly state what specific data were used from a previous publication (Shi et al., 2017) in the methods section and figure 3A. If the methodology from Shi et al. 2017 deviated from that described on lines 259-279, such deviations must be described.

R: We added the information, in the methods section, that single-cell transcriptome was performed using data from a previous publication. See pag. 6, lines 204-206 (lines 200-202 in the clear copy) and Pag. 9, lines 296-298 (lines 288-290 in the clear copy).

Experiments from Shi et al. (2017) were approved by the University of Virginia ethical committee. This information must be included in the current manuscript in the Ethical Approval section.

R: We added the information to the ethical approval section. See pag. 6, lines 182-183 (lines 178-179 in the clear copy).

The endpoint of each protocol involving mice must be clearly stated within each protocol, with the methods of euthanasia and confirmation of death described as required by the Journal within that section. Line 282 is too ambiguous.

R: Done

The following sections are required: Data availability statement; Competing interests; Author contributions; Acknowledgements:

R: We have added the required sections.

Please address all comments from external referees, the statistical and reviewing editor, and those presented here. Please also ensure the list of requirements for publication in the journal is met in the next revision.

R: Done

Statistics Editor:

Comments for Author:

The statistical reporting in the manuscript requires adjustments to fully comply with the journal's guidelines and enhance clarity. Below are specific comments and recommendations:

P-value Reporting:

While exact p-values are provided in the figure legends, the journal guidelines require them to be included on the figures themselves as well. Per the guidelines, p-values must be reported to three significant figures (e.g., 0.00236 or 0.523) for all values >0.001 , even when no statistical significance is observed. For p-values <0.001 , reporting as " <0.001 " is acceptable. Currently, this is not reflected on the figures. Additionally, avoid using asterisks alone to denote significance on figures.

R: We added the necessary information in the figures regarding the p-values as suggested by the statistic editor

Sample Size (n) Reporting:

The figures and their legends currently lack sample size (n) values for all reported outcomes, which must be clearly stated in both locations per the journal's guidelines. Each 'n' should be explicitly defined (e.g., "x cells from y slices in z animals") to ensure transparency and avoid pseudoreplication. Authors should verify that 'n' reflects the appropriate experimental unit based on the study design.

R: Agreed. We have explicitly defined the sample size (n) in all figure legends to ensure clarity and transparency.

Normality Testing:

The Statistics section states that Shapiro-Wilk or Kolmogorov-Smirnov tests were used to assess normality but does not specify how the choice between these tests was determined. Given their differing sensitivities (e.g., Shapiro-Wilk is more powerful for small sample sizes, typically $n < 50$), a brief clarification (e.g., "based on sample size") would improve transparency and reproducibility.

R: We added the information that we used Shapiro-Wilk test based on the sample size

T-test Application:

The use of "paired or unpaired" Student's t-tests is mentioned, but the criteria for choosing between them are unclear. This should be briefly specified in the Statistics section (e.g., "paired t-tests for within-subject comparisons, unpaired for between-group comparisons"). Additionally, I found it difficult to infer which comparisons in the study were paired versus unpaired based on the provided information. The authors should carefully review each comparison to ensure the selected t-test aligns with the experimental design and update the text or figure legends accordingly.

R: We have provided a detailed explanation of the use of paired and unpaired t-tests for each experiment conducted in our study. See pag. 13, lines 426-442 (lines 416-432 in the clear copy) and figure legends.

Recommendation for Future Reporting:

While not required by the current guidelines, I strongly recommend including effect sizes (e.g., Cohen's d or Hedges' g) and confidence intervals in future submissions.

These metrics provide valuable insight into the magnitude and uncertainty of the findings, enhancing the interpretability of the results beyond p-values alone.

R: Thank you very much for the suggestions. We will consider this a valuable recommendation and will carefully incorporate it into future submissions.

REFEREE COMMENTS

Referee #1:

The authors have adequately responded to my previous concerns. They have added additional text in the manuscript to describe study limitations. They have also provided additional clarification throughout. I have no further comments and congratulate the authors on the thorough study.

R: We sincerely thank the reviewer for their positive feedback and thoughtful evaluation. We are grateful for the constructive comments provided throughout the review process, which helped us improve the clarity and quality of the manuscript. We appreciate your kind words and support.

Referee #2:

- The first comment of the senior editor is asking the authors to include actual p values in figures. Answer of the authors: "Done", but this is not the case in the revised manuscript submitted.

R: We sincerely apologize for the oversight. We have included the actual p-values in the Results section and in the figure legends. In this revised version of the manuscript, the actual p-values have also been added directly to the figures.

- The second comment of the senior editor is asking the authors to state the statistical test(s) used in tables and figures in their respective legends. Answer of the authors: "Done", but this is not the case in the revised manuscript submitted.

R: The reviewer's comment is somewhat unexpected, as we have already included the statistical tests used in each figure legend. In addition to the figure legends, the current version of the manuscript also provides a detailed description of all statistical tests in the "Statistics" section of the Methods. See pag. 13, lines 426-442 (lines 416-432 in the clear copy) and figure legends.

These first two points are important, because the first version of the manuscript was very seriously and concernedly affected by major statistical issues.

R: Agreed. In the current version of the manuscript, we have included all relevant statistical details throughout the text to ensure clarity and transparency.

- The article versions provided with highlighted changes do not correspond to the final "untracked" version provided, the former being clearly unfinished versions. This will make the reviewing process difficult.

R: We apologize for the confusion caused by the mismatch between the highlighted and untracked versions of the manuscript. It was not our intention to submit unfinished versions, and we understand how this may complicate the reviewing process. To address this, we have now ensured that the revised manuscript with highlighted changes accurately corresponds to the final untracked version. We appreciate your understanding and thank you for your patience.

- One example from a problematic presentation of critical data for this study is the data presented in the abstract on lines 89-92, from data presented in the results section on lines 426-429. It is stated by the authors in the abstract that they "found that a substantial portion of oxytocinergic neurons in the PVN (42%) co-express vesicular glutamate transporter 2 (Vglut2) and presumably project directly to RTN neurons, identifying the RTN as a major target of oxytocinergic projections". This is a very strong assertion, and is clearly presented as a major finding of the paper. However, this assertion is based on wrong calculation, and appropriate calculation would show that this number is much lower. Indeed, the authors' calculation is based on data presented in the results section on lines 426-429, where it is stated the the authors "found an appreciable number of Oxt neurons immunoreactive for both Ctb and VGlut (Ctb+/VGlut2+/Oxt+: 16.5 {plus minus} 5.6, vs. Ctb+/VGlut2+: 39.2 {plus minus} 9 neurons; 42 {plus minus} 3.2%) (Figs. 2C, 2Ci-Civ and 2E)". This is therefore not what is stated in the abstract, and the calculation made does not show what is interpreted from it. The authors did not make a calculation of the relative proportion of overall Oxt neurons that are glutamatergic and project to the RTN, as stated in the abstract, they actually calculated the relative proportion of glutamatergic neurons projecting to the RTN that are oxytocinergic. This is very different and seriously changes the data interpretation. From a broad and conceptual point of view, oxytocinergic neurons have a very large projection profile, projecting to most parts of the brain and being involved in the regulation of many processes and behaviours. It would therefore be a major finding if out of all these Oxt neurons, 42% of them projected to such a very small brainstem nucleus as the RTN, a claim that the authors are making based on wrong calculations.

I had already made comments on these calculations of percentages in my first review (cf. last point of page 10 of the pdf document "Response to referees"), and here mistakes are made again. Clearly the authors did not take full account of my comment. Given that it is a critical one, it can only be expected that other ones of my previous comments will not have been taken into account, and therefore it will involve a significant amount of work to revise this new version.

R: We sincerely apologize for the misinterpretation and incorrect calculation presented in the original version of the manuscript. You are absolutely right in pointing out that the percentage reported in the abstract and results did not accurately reflect the proportion of oxytocinergic neurons projecting to the RTN, but rather the proportion of glutamatergic RTN-projecting neurons that are oxytocinergic.

We have carefully revised the manuscript to address this issue. Specifically, we corrected the statement in the abstract and the corresponding section in the results to accurately reflect the data. We now report that approximately 11% of PVN oxytocinergic neurons are both glutamatergic and project to the RTN. Additionally, we have tempered the language in the discussion to avoid overstating the importance of the RTN as a target of PVN oxytocinergic projections.

We deeply appreciate your careful reading and constructive feedback, and we regret that this concern was not fully addressed in our earlier revision. Please be assured that we have now taken all necessary steps to correct and clarify this critical point.

Dear Dr Moreira,

Re: JP-RP-2025-287845R2 "Oxytocinergic signaling in the respiratory parafacial region increases activity of chemosensitive neurons and respiratory activity" by Emmanuel V Araújo, Phelipe E Silva, Luiz M Oliveira, Yingtang Shi, Ana C Takakura, Daniel K Mulkey, and Thiago S Moreira

Thank you for submitting your manuscript to The Journal of Physiology. It has been assessed by a Reviewing Editor and by 2 expert referees and we are pleased to tell you that it is acceptable for publication following satisfactory revision.

REVISION CHECKLIST:

We look forward to receiving your revised submission.

Yours sincerely,

Harold Schultz
Senior Editor
The Journal of Physiology

REQUIRED ITEMS

Please move the Additional Information section and Data Availability Statement to the end of the manuscript, before the reference list.

EDITOR COMMENTS

Reviewing Editor:

Comments to the Author:

Referee 2 provided a detailed assessment of the manuscript, indicating that the revised version has improved in quality. However, Referee 2 also pointed out aspects that remained unclear or ambiguous. These included concerns regarding cell counting and the accuracy of calculations to quantify the relative expression/transfection. The referee highlighted parts of the text that led to confusing interpretations and commented on the lack of results validating the expression of ChR2 in oxytocinergic neurons in the manuscript, which were only mentioned in the responses to the referees' comments. Overall, addressing these and other comments presented in the Referee's report can prevent misinterpretations and enhance the quality of the manuscript. Therefore, I recommend that the authors respond to these comments carefully and make the necessary corrections in the manuscript.

Senior Editor:

Comments to the Author:

Thank you for submitting a revised version of your research article for consideration in the Journal of Physiology. The article has been reviewed by the reviewers and found to be potentially acceptable for publication pending adequate revision to address some remaining concerns from Referee 2. Please address these comments from the external referee.

REFeree COMMENTS

Referee #1:

I continue to not have any further comments. The authors have adequately responded to previous concerns.

Referee #2:

The authors have provided a significantly revised and improved manuscript that takes most of my previous comments into consideration. However, I am still concerned by the number and importance of the points I had to raise in the first round of review, most of which were dismissed as 'mistakes'. So many 'mistakes' regarding fundamental aspects of scientific rigour, such as basic methodology, data analysis and presentation, and statistics, at the level of such a formal process of submitting a scientific article for peer review, naturally gives rise to concerns regarding the manner in which the experiments were conducted. The submitted material was also incoherent, with temporary files submitted where the 'tracked changes' file did not align with the clean version during the last round of review.

I have some points to raise on the latest version of the manuscript:

(Side note: all my indications regarding line numbers are based on the merged pdf file provided by the journal and called "JP-RP-2025-287845R2_Merged_PDF")

1. There are remaining issues regarding how the oxytocinergic projections to the RTN/pFRG are presented, with mismatches between interpretations and calculations, and ambiguities in major conclusions:

- In the results section, paragraph 1, it is stated on lines 453-455 that "the vast majority of the ChR2-expressing neurons are oxytocinergic (87.5 {plus minus} 1.2%; 116.5 {plus minus} 14.3 mCherry+/Oxt+ vs. 133 {plus minus} 17 Oxt+; N = 4) (Figs. 1B-D)". However, there are two levels of issues in this statement.

First, and similarly to many instances in the previous rounds of reviews, the text is in mismatch with the calculation performed. The text refers to the proportion of ChR2-expressing neurons that are oxytocinergic, while the calculation provided calculates the proportion of oxytocinergic neurons that express ChR2.

Second, the calculation is incorrect at its core. This type of calculation is of a relative proportion, here the proportion of mCherry+/Oxt+ neurons over the total number of Oxt+ neurons. These numbers come from Fig. 1C-D, and it is stated in the legend of Fig. 1C that the "total number of oxytocin-positive" neurons are provided, with the legend "oxytocin" shown in the figure. Then, on Fig. 1D, a schematic representation of the numbers presented in Fig. 1C is provided, with the same legend "oxytocin" to depict oxytocinergic neurons. However, on Fig. 1D, it is evident that the group "oxytocin" does not contain all oxytocinergic neurons, but only those that were negative for ChR2, since the group "Double" clearly shows different neurons compared to the group "Oxytocin". Therefore, one can only assume that the calculation provided in Fig. 1C for all oxytocinergic neurons is erroneous, and actually presents only oxytocinergic neurons that did not express ChR2. Thus, using this number "133 {plus minus} 17 Oxt+" for the calculation provided in the results section as the total number of oxytocinergic neurons to calculate a proportion is mathematically incorrect.

These issues at multiple, overlaying levels, are problematic, especially since such issues were already largely present and notified in the previous rounds of reviews.

- The following sentence states "We found only that 10 {plus minus} 2.6% mCherry+/Oxt-". Please provide the numbers on which this calculation is based, especially since it is linked to the calculations performed on the previous sentence that are incorrect. Also, this sentence is difficult to understand in its current form, please rephrase.

- On Fig. 1K, a different experimental model seems to have been used to trace oxytocinergic projections to the RTN/pFRG compared to the rest of the figure, and in particular compared to the data presented on Fig. 1E-J. The experimental model used in Fig. 1K is not explicitly provided anywhere, but the figure legend states that the image shows "the expression of channelrhodopsin (GFP/Ai6 mice, green)" (line 1099). This is very puzzling, since the Ai6 mouse line does not express channelrhodopsin. Furthermore, the green labeling that could correspond to a labeling from an Ai6 mouse is presumably expressed in Oxt neurons, as the figure legend seems to indicate, which could mean that OxtCre/+ mice were crossed with Ai6 mice, yet again no information is provided in the manuscript. If this is the case, why such a change in model compared to the tracing done with the AAV expressing hChR2-mCherry in Oxt neurons and used in the panels presented just before on the figure?

More concerning, the interpretation provided from these data in the results section is that "the density of PVN Oxt varicosities in the VRC is similar to that in the RTN/pFRG region" (lines 464-465), thereby comparing data obtained in different experimental models (viral tracing in OxtCre/+ mice vs. constitutive transgenic expression in putative OxtCre/+::Ai6 mice). These models strongly differ in their modes of expression, with for instance the viral tracing experiments that will label only a limited fraction of the Oxt neurons due to the inherent targeting restrictions of the injection strategies, while the constitutive transgenic expression will be much more exhaustive. It is therefore impossible and incorrect to draw any comparative conclusion in the respective densities of projections/varicosities regarding these different sets of experiments. Furthermore, the striking green Oxt projections shown on Fig. 1K are in the facial nucleus, not in the RTN/pFRG, where only a few fibers are shown. This is supported by the data presented on Fig. 1I-J.

- Lines 473-475, "a portion of Ctb-labeled [...] 39.2 {plus minus} 9 neurons; 42 {plus minus} 3.2%", the text and the calculation are mismatched. The calculation shows the proportion of Ctb-labeled glutamatergic neurons that are also Oxt+,

and not the proportion of Ctb-labeled Oxt neurons that are also glutamatergic as stated in the text. This then affects the following sentence, lines 476-478, which starts with " This indicates that ~11% of oxytocinergic neurons in the PVN are both excitatory (VGlut2+) and project to the RTN (Ctb+)". Given that the previous sentence is incorrect, one can only assume that this next sentence is also incorrect, even more so since the calculation is not even provided despite the mention "this indicates". I fail to understand the logical mathematical continuity between the calculation in the first sentence and the "11%" number provided, although it should apparently be evident given the mention "this indicates". Clarifying these aspects is particularly important, as this specific calculation is emphasized in the abstract, lines 118-120, therefore representing a major finding of the study highlighted by the authors.

- Lines 482-483, sentence "In addition [...] 11.8 neurons)", the text and the calculation are mismatched. The calculation shows the proportion of Oxt neurons that express VGat, and not the proportion of VGat neurons that express oxytocin as stated in the text.

- The two sentences lines 478-480 ("Some Ctb [...] (Figs. 2E-F)") and 483-485 ("A large [...] (Figs. 2E-F)") make the same statement and are redundant, yet they do not provide the same calculations: 39 {plus minus} 7.4% in the first sentence, 55% in the second. This is disconcerting.

- The first paragraph of the results section, which is aimed at characterizing the projections from hypothalamic oxytocinergic neurons to the RTN/pFRG region, has been containing many issues from the first version of the manuscript submitted, many issues remaining as detailed above. Yet, these results support major findings of the study, highlighted in the abstract, key points and in the graphical abstract. Since the first version of the manuscript, the authors have moderated their conclusions, which were based on inaccurate calculations. Still, in my opinion two key sentences remain to be moderated/modified:

First, in the graphical abstract, the sentence: "The paraventricular nucleus of the hypothalamus (PVN) projects to the RTN, serving as the primary source of oxytocin (Oxt) in the brainstem". This sentence is clearly misleading, its form altering its content. Indeed, in the way it is written, this sentence could easily and naturally be understood as meaning that the primary source of oxytocin in the brainstem goes to the RTN/pFRG. This was probably not meant by the authors, yet the order of the words in the sentence, putting the part "serving as the primary source of oxytocin (Oxt) in the brainstem" just after "RTN", naturally links both aspects. A more appropriate sentence would be "The paraventricular nucleus of the hypothalamus (PVN), which serves as the primary source of oxytocin (Oxt) in the brainstem, projects to the RTN". In my opinion, it is critical to disambiguate this sentence, which could provide an incorrect message, and could have impact since it is part of the graphical abstract.

Second, regarding the second sentence of the "Key points", lines 84-85, which states "The paraventricular nucleus of the hypothalamus (PVN) projects to the RTN and is considered the primary source of oxytocin in the brainstem". Similarly to the first point above, this sentence is misleading, and I suggest rephrasing into "The paraventricular nucleus of the hypothalamus (PVN), considered as the primary source of oxytocin in the brainstem, projects to the RTN".

2. Other concerns, most related to methodological issues:

- The total animal number used in the study, and stated lines 205-206, keeps on changing, a different number was provided in the previous version of the manuscript. Also, line 381, "N = 67" is stated for the experiments on slices, which is different to the number provided on lines 205-206 ("56").

- The experiments presented on Fig. 5D with the GRK2 blocker, their presentation in the text and their interpretation in the discussion, are very unclear to me, and problematic.

First, lines 584-586, the main text states that "3 bouts of TGOT (200 mM)" applications were made, while the figure legend mentions only two applications (line 1197, "application of TGOT [...] before and after GRK2 blocker"). The methods section describing these experiments also mentions two TGOT applications. Please clarify.

Second, lines 586-588, it is stated "Consistent with GRK2-dependent desensitization, inhibition of the TGOT-induced reduction in RTN neuronal activity after 3 min exposure ($t = 1.366$, $df = 12$, $p = 0.1971$) (Fig. 5C-D)". This sentence does not make sense to me in the way it is written. What it seems to state is that application of the GRK2 blocker would reduce the

desensitization of the response to TGOT by RTN neurons. However, the results and statistics do not show that, since there is no difference ($p = 0.1971$) between the "TGOT" and "TGOT + CCG258208" groups (Fig. 5D). Unless I have missed something, this is a concerning over-statement and misinterpretation of the data, which results in an entirely incorrect discussion paragraph (lines 764-7750) and invalid conclusions.

Third, while here a clearly non-significant result is interpreted as significant and is used to draw important conclusions (mechanism of desensitization of the response to TGOT), other instances in the text of very borderline "non-significant" effects in control conditions are easily dismissed. For instance, line 524 " $p = 0.064$ ", line 549 " $p = 0.076$ " and " $p = 0.052$ ". Statistics are not dichotomist, they represent a continuum, and the 0.05 confidence barrier is arbitrary and artificial. Strong pushes are made towards alternative representations that would better depict the distribution of data and estimations in their differences (cf. for instance PMID: 31217592, Ho J et al. 2019). While it is of course difficult to require an instant switch in the practices for data analysis and their representations, it is still necessary to avoid over-using the 0.05 barrier dichotomy. For the line 524 " $p = 0.064$ " data, it is highly possible that the 9 mW light power used had a heating effect on RTN/pFRG neurons and induced a small increase in respiratory amplitude as shown by the P value of 0.064 (cf. PMID: 27895987, Shin Y et al. 2016, for accurate measures of the effects of different light powers in the brain ; also, by the way, the norm in the field of optogenetics for the representation of the light power used is the light power density in mW/mm², to provide a more accurate representation of the brain surface affected, which is relative to the optical parameters of the equipment used to transmit the light into the brain). The same applies to the results shown on line 549, with " $p = 0.076$ " and " $p = 0.052$ ", it is clearly difficult to state confidently that these neurons absolutely do not respond to CO₂ and to TGOT. Yet, the authors make it an important message of the article that only chemosensitive RTN neurons are activated by oxytocin.

- In Fig. 3, experiments using OxtCre^{+/+}::Ai32 mice are presented. In my first review, I raised this point:

"The assessment of the selectivity of the ChR2 expression in oxytocinergic neurons in Oxt::Cre^(+/-)::Ai32 mice must be shown and quantified. This is an important control experiment, selectivity in double transgenic mice with Cre-lox recombination is never a given, yet it is a prerequisite for the validity of the experiments.

Also, it would be important to show an image of the ChR2 expression in oxytocinergic fibers in the RTN/pFRG in these mice."

The authors responded: "We appreciate the reviewer's suggestion regarding the assessment of ChR2 selectivity in oxytocinergic neurons in OxtCre^{+/+}::Ai32 mice. Indeed, Cre-lox recombination does not always guarantee exclusive expression in the target population, making validation essential. To address this, we have now quantified the colocalization of ChR2 (via YFP/mCherry fluorescence) with oxytocin (OXT) immunoreactivity in the PVN, providing an estimate of selectivity. Additionally, we provide representative confocal images showing ChR2 expression in oxytocinergic fibers within the RTN/pFRG to confirm the presence of projections in the targeted region. These data have been incorporated into the revised manuscript."

Yet, the current manuscript does not contain any of the necessary quantification and confocal images regarding ChR2 expression in the OxtCre^{+/+}::Ai32 mice that the authors stated that they included in their rebuttal.

Minor points:

- Fig. 1A, please include the mention "double-floxed" in the name of the virus provided on the figure, otherwise one cannot understand how there will be specificity in the expression of this virus in OxtCre^{+/+} mice.

Same comment for line 449, where "AAV-hChR2-mCherry" is written, the mention "double-floxed" should be added.

- Fig. 2A, on the coronal view at the level of the PVN, the mention "Ctb⁺/Oxt⁺/VGlut2⁺" is made, however this mention is misleading since (i) it is missing the VGat aspect which is part of this experiment as depicted just above on the same schematic with the names of the transgenic mice used, and (ii) the experimental strategy depicted using Ctb tracing in transgenic mice does not enable the Oxt labelling presented. I suggest removing the mention "Ctb⁺/Oxt⁺/VGlut2⁺", or making it much clearer.

- Legend of Fig. 2B, only the photomicrographs are mentioned, please also mention the schematics below the photomicrographs and state if all injections from all mice were reported.

- Fig. 2D, the legend states that "Ctb injection into the RTN/pFRG labeled neurons that are immunoreactive for VGat (green) and oxytocin (magenta) in the PVN region", while Fig. 2D shows a VGat-negative neuron. This is inconsistent.

- The authors have stated in their previous rebuttal that all sections for histology were cut at 40 μm , however there are still several mentions of 30 μm sections in the text (e.g. lines 451 and 473).

- A previous point that was not taken into account, although the authors had answered in their rebuttal that it was done: Fourth paragraph of the results section, "G protein-coupled receptor kinases [...] desensitization", please provide a reference supporting this statement (lines 581-583).

- Line 202, "oxytocin expression" should probably rather be "oxytocin receptor expression in the RTN/pFRG".

- Line 332, a polyclonal chicken anti-GFP primary antibody is stated to have been used, however no secondary anti-chicken antibody is mentioned further down in the text.

- Line 501, "oxytocin transcripts" should probably rather be "oxytocin receptor transcripts".

- Line 546, "Figures 4C and 6A" should probably rather be "Figures 4C and 5A".

- Line 1110, "oxytocinergic and glutamatergic" should probably rather be "GABAergic and glutamatergic".

- Lines 1134-1136, for clarity please add the mention that the GFP-positive neurons harvested were from RTN/pFRG neurons.

- There are still several typographic errors, missing words etc. throughout the text (e.g. "ceiling effect" and not "celling effect").

- There are several issues with references, with references cited in the text missing from the reference list (including one that I had notified in a previous round of reviews) and references dually present in the reference list as "a" and "b".

END OF COMMENTS

EDITOR COMMENTS

Reviewing Editor:

Comments to the Author:

Referee 2 provided a detailed assessment of the manuscript, indicating that the revised version has improved in quality. However, Referee 2 also pointed out aspects that remained unclear or ambiguous. These included concerns regarding cell counting and the accuracy of calculations to quantify the relative expression/transfection. The referee highlighted parts of the text that led to confusing interpretations and commented on the lack of results validating the expression of ChR2 in oxytocinergic neurons in the manuscript, which were only mentioned in the responses to the referees' comments. Overall, addressing these and other comments presented in the Referee's report can prevent misinterpretations and enhance the quality of the manuscript. Therefore, I recommend that the authors respond to these comments carefully and make the necessary corrections in the manuscript.

R: We would like to sincerely thank the reviewing editor for highlighting the major concerns raised by Reviewer #2. In the current version of the manuscript, we have addressed all these concerns, and we hope that the revised version is now clear both to the reviewer and to future readers.

Senior Editor:

Comments to the Author:

Thank you for submitting a revised version of your research article for consideration in the Journal of Physiology. The article has been reviewed by the reviewers and found to be potentially acceptable for publication pending adequate revision to address some remaining concerns from Referee 2. Please address these comments from the external referee.

R: In the current version of the manuscript, we have addressed all these concerns, and we hope that the revised version is now clear both to the reviewer and to future readers.

REFEREE COMMENTS

Referee #1:

I continue to not have any further comments. The authors have adequately responded to previous concerns.

R: Thank you.

Referee #2:

The authors have provided a significantly revised and improved manuscript that takes most of my previous comments into consideration. However, I am still concerned by the number and importance of the points I had to raise in the first round of review, most of which were dismissed as 'mistakes'. So many 'mistakes' regarding fundamental aspects of scientific rigour, such as basic methodology, data analysis and presentation, and statistics, at the level of such a formal process of submitting a scientific article for peer review, naturally gives rise to concerns regarding the manner in which the experiments were conducted. The submitted material was also incoherent, with temporary files submitted where the 'tracked changes' file did not align with the clean version during the last round of review.

R: Thank you for your thoughtful and detailed feedback. We sincerely apologize for the confusion and concerns caused by the previous submissions and for the mistakes related to methodology, data analysis, and presentation. We fully understand the reviewer's concerns.

Following your comments, we have carefully and systematically reviewed every aspect of the manuscript to ensure accuracy, consistency, and clarity. We have corrected the methodological descriptions, clarified data analyses, and ensured that statistical reporting adheres to proper standards. Additionally, we have thoroughly verified that the clean and tracked versions of the manuscript are now fully aligned to prevent any further confusion during the review process.

We greatly value your input and take your concerns very seriously. Your feedback has been essential in improving the quality and integrity of our work, and we are committed to maintaining the highest standards in both our scientific practices and communication.

I have some points to raise on the latest version of the manuscript:

(Side note: all my indications regarding line numbers are based on the merged pdf file provided by the journal and called "JP-RP-2025-287845R2_Merged_PDF")

1. There are remaining issues regarding how the oxytocinergic projections to the RTN/pFRG are presented, with mismatches between interpretations and calculations, and ambiguities in major conclusions:

R: We hope that the present version is now clear and avoids any mismatches between data interpretation and calculations.

- In the results section, paragraph 1, it is stated on lines 453-455 that "the vast majority of the ChR2-expressing neurons are oxytocinergic (87.5 {plus minus} 1.2%; 116.5 {plus minus} 14.3 mCherry+/Oxt+ vs. 133 {plus minus} 17 Oxt+; N = 4) (Figs. 1B-D)". However, there are two levels of issues in this statement.

First, and similarly to many instances in the previous rounds of reviews, the text is in mismatch with the calculation performed. The text refers to the proportion of ChR2-expressing neurons that are oxytocinergic, while the calculation provided calculates the proportion of oxytocinergic neurons that express ChR2.

R: Thank you for carefully pointing out this issue. You are correct that the calculation reflects the proportion of oxytocinergic neurons that express ChR2 (mCherry+/Oxt+ divided by total Oxt+), rather than the proportion of ChR2-expressing neurons that are oxytocinergic, as was mistakenly stated in the text. We have corrected the sentences in the Results section to accurately describe the data:

"...the majority of the oxytocinergic neurons express ChR2 (79 ± 2.5%; 141.2 ± 3.2 mCherry+/Oxt+ vs. 178.5 ± 3.42 Oxt+; N = 4) (Figs. 1B-D)". See lines 422-423.

We also carefully reviewed the rest of the manuscript to ensure consistency and accuracy in the description of all similar calculations. Thank you again for your valuable feedback, which helped us improve the clarity and precision of our manuscript.

Second, the calculation is incorrect at its core. This type of calculation is of a relative proportion, here the proportion of mCherry+/Oxt+ neurons over the total number of Oxt+

neurons. These numbers come from Fig. 1C-D, and it is stated in the legend of Fig. 1C that the "total number of oxytocin-positive" neurons are provided, with the legend "oxytocin" shown in the figure. Then, on Fig. 1D, a schematic representation of the numbers presented in Fig. 1C is provided, with the same legend "oxytocin" to depict oxytocinergic neurons. However, on Fig. 1D, it is evident that the group "oxytocin" does not contain all oxytocinergic neurons, but only those that were negative for ChR2, since the group "Double" clearly shows different neurons compared to the group "Oxytocin". Therefore, one can only assume that the calculation provided in Fig. 1C for all oxytocinergic neurons is erroneous, and actually presents only oxytocinergic neurons that did not express ChR2. Thus, using this number "133 {plus minus} 17 Oxt+" for the calculation provided in the results section as the total number of oxytocinergic neurons to calculate a proportion is mathematically incorrect.

R: The reviewer is right, and we sincerely apologize for the calculation errors. We have thoroughly reviewed all PVN cell counts and performed the corrected calculations as indicated by the reviewer. See lines 422-423.

These issues at multiple, overlaying levels, are problematic, especially since such issues were already largely present and notified in the previous rounds of reviews.

R: We hope that the present version of the manuscript satisfactorily addresses the reviewer's concerns.

- The following sentence states "We found only that 10 {plus minus} 2.6% mCherry+/Oxt-". Please provide the numbers on which this calculation is based, especially since it is linked to the calculations performed on the previous sentence that are incorrect. Also, this sentence is difficult to understand in its current form, please rephrase.

R: We have thoroughly reviewed all PVN cell counts and performed the corrected calculations as indicated by the reviewer. See lines 426-427.

- On Fig. 1K, a different experimental model seems to have been used to trace oxytocinergic projections to the RTN/pFRG compared to the rest of the figure, and in particular compared to the data presented on Fig. 1E-J. The experimental model used in Fig. 1K is not explicitly provided anywhere, but the figure legend states that the image shows "the expression of channel rhodopsin (GFP/Ai6 mice, green)" (line 1099). This is very puzzling, since the Ai6 mouse line does not express channelrhodopsin. Furthermore, the green labeling that could correspond to a labeling from an Ai6 mouse is presumably expressed in Oxt neurons, as the figure legend seems to indicate, which could mean that OxtCre/+ mice were crossed with Ai6 mice, yet again no information is provided in the manuscript. If this is the case, why such a change in model compared to the tracing done with the AAV expressing hChR2-mCherry in Oxt neurons and used in the panels presented just before on the figure?

R: The experiment shown in Figure 1K did not accurately represent the intended purpose. In fact, the experiment was conducted to assess the selectivity of ChR2 expression in oxytocinergic neurons in OxtCre⁺::Ai32 mice, an important control, as highlighted by the reviewer, given that Cre-lox recombination in double transgenic mice does not always guarantee selective expression. We have now moved these data to Figure 3, where they more appropriately illustrate ChR2 expression (via GFP immunohistochemistry) in oxytocinergic terminals (identified by oxytocin immunoreactivity). Although we did not quantify the total number of labeled neurons, we are confident that ChR2 is selectively expressed in oxytocin-producing

neurons and their terminals in the OxtCre^{+/+}::Ai32 line. See lines 492-497 and Fig. 3J.

More concerning, the interpretation provided from these data in the results section is that "the density of PVN Oxt varicosities in the VRC is similar to that in the RTN/pFRG region" (lines 464-465), thereby comparing data obtained in different experimental models (viral tracing in OxtCre/+ mice vs. constitutive transgenic expression in putative OxtCre^{+/+}::Ai6 mice). These models strongly differ in their modes of expression, with for instance the viral tracing experiments that will label only a limited fraction of the Oxt neurons due to the inherent targeting restrictions of the injection strategies, while the constitutive transgenic expression will be much more exhaustive. It is therefore impossible and incorrect to draw any comparative conclusion in the respective densities of projections/varicosities regarding these different sets of experiments. Furthermore, the striking green Oxt projections shown on Fig. 1K are in the facial nucleus, not in the RTN/pFRG, where only a few fibers are shown. This is supported by the data presented on Fig. 1I-J.

R: First of all, I would like to apologize for the many errors in the text - something inexplicable seems to have happened during the last submission. Our apologies. However, we did not compare different animal models in this study. In the current version of the manuscript, we have clarified the description of the animal models used and ensured this information is placed appropriately in the text to avoid any misinterpretation. We thank the reviewer for helping us improve the clarity of our methods.

- Lines 473-475, "a portion of Ctb-labeled [...] 39.2 {plus minus} 9 neurons; 42 {plus minus} 3.2%", the text and the calculation are mismatched. The calculation shows the proportion of Ctb-labeled glutamatergic neurons that are also Oxt+, and not the proportion of Ctb-labeled Oxt neurons that are also glutamatergic as stated in the text. This then affects the following sentence, lines 476-478, which starts with " This indicates that ~11% of oxytocinergic neurons in the PVN are both excitatory (VGlut2+) and project to the RTN (Ctb+)". Given that the previous sentence is incorrect, one can only assume that this next sentence is also incorrect, even more so since the calculation is not even provided despite the mention "this indicates". I fail to understand the logical mathematical continuity between the calculation in the first sentence and the "11%" number provided, although it should apparently be evident given the mention "this indicates". Clarifying these aspects is particularly important, as this specific calculation is emphasized in the abstract, lines 118-120, therefore representing a major finding of the study highlighted by the authors.

R: We would like to sincerely apologize once again for the numerous errors in the previous version of the text - something inexplicable seems to have occurred during the last submission. We have now addressed all the concerns raised by the reviewer. In this revised version, we have made the appropriate calculations and carefully edited the relevant sentences to improve clarity for both the reviewer and future readers. We hope the current version is now clear and satisfactory.

Now it is reading "*As illustrated in Fig. 2, we found a portion of Ctb-labeled excitatory neurons (VGlut2) were also immunoreactive for Oxt (Ctb+/VGlut2+/Oxt+: 16.5 ± 5.6, vs. Ctb+/VGlut2+: 39.3 ± 18.8 neurons), suggesting the RTN is one of several downstream targets of oxytocinergic neurons (Figs. 2C, 2Ci-Civ and 2E). Some Ctb labeled PVN neurons express Oxt (Ctb+/Oxt+: 20.8 ± 5.4, vs. Ctb+: 55.3 ± 23.7*

neurons) (Figs. 2E-F). Few back labeled neurons showed VGat signal (Ctb+/VGat+: 3.7 ± 1.5, vs. Ctb+: 51.7 ± 19.6 neurons) (Figs. 2D, 2Di-Div and 2F). In addition, only a few of the Oxt neurons express VGat (VGat+/Oxt+: 9.7 ± 7.7, vs. Oxt+: 145 ± 11.8 neurons) (Figs. 2D and 2F). We also found the majority of PVN projections to the ventral parafacial region are ipsilateral, whereas contralateral labeling was extremely rare (data not shown).” See lines 445-455.

- Lines 482-483, sentence "In addition [...] 11.8 neurons)", the text and the calculation are mismatched. The calculation shows the proportion of Oxt neurons that express VGat, and not the proportion of VGat neurons that express oxytocin as stated in the text.

R: We made the proper corrections. See lines 445-447.

- The two sentences lines 478-480 ("Some Ctb [...] (Figs. 2E-F)") and 483-485 ("A large [...] (Figs. 2E-F)") make the same statement and are redundant, yet they do not provide the same calculations: 39 {plus minus} 7.4% in the first sentence, 55% in the second. This is disconcerting.

R: We are very sorry. We removed the sentences that are redundant. The proper calculations were made, and it is stated on lines 451-453.

- The first paragraph of the results section, which is aimed at characterizing the projections from hypothalamic oxytocinergic neurons to the RTN/pFRG region, has been containing many issues from the first version of the manuscript submitted, many issues remaining as detailed above. Yet, these results support major findings of the study, highlighted in the abstract, key points and in the graphical abstract. Since the first version of the manuscript, the authors have moderated their conclusions, which were based on inaccurate calculations. Still, in my opinion two key sentences remain to be moderated/modified:

First, in the graphical abstract, the sentence: "The paraventricular nucleus of the hypothalamus (PVN) projects to the RTN, serving as the primary source of oxytocin (Oxt) in the brainstem". This sentence is clearly misleading, its form altering its content. Indeed, in the way it is written, this sentence could easily and naturally be understood as meaning that the primary source of oxytocin in the brainstem goes to the RTN/pFRG. This was probably not meant by the authors, yet the order of the words in the sentence, putting the part "serving as the primary source of oxytocin (Oxt) in the brainstem" just after "RTN", naturally links both aspects. A more appropriate sentence would be "The paraventricular nucleus of the hypothalamus (PVN), which serves as the primary source of oxytocin (Oxt) in the brainstem, projects to the RTN". In my opinion, it is critical to disambiguate this sentence, which could provide an incorrect message, and could have impact since it is part of the graphical abstract.

R: We sincerely appreciate the reviewer’s help in improving the English and ensuring the meaning is clearly conveyed.

Second, regarding the second sentence of the "Key points", lines 84-85, which states "The paraventricular nucleus of the hypothalamus (PVN) projects to the RTN and is considered the primary source of oxytocin in the brainstem". Similarly to the first point above, this sentence is misleading, and I suggest rephrasing into "The paraventricular nucleus of the hypothalamus (PVN), considered as the primary source of oxytocin in the brainstem, projects to the RTN".

R: Thank you very much for the help with the English language. We have made the modifications accordingly. See lines 49-50.

2. Other concerns, most related to methodological issues:

- The total animal number used in the study, and stated lines 205-206, keeps on changing, a different number was provided in the previous version of the manuscript. Also, line 381, "N = 67" is stated for the experiments on slices, which is different to the number provided on lines 205-206 ("56").

R: The correct number of *in vitro* experiments is N = 56. We used N = 21 mice to investigate the RTN firing rate response to CO₂; of these, N = 16 were CO₂-sensitive and N = 5 were not (Obs: those animals were also used to investigate the dose-response curve and the difference between sexes). N = 7 mice were used to assess the effect of TGOT-induced changes in RTN firing rate under control conditions and in the presence of a blocker cocktail. N = 8 and N = 9 mice were used to examine RTN activity during the first and second bath applications of TGOT, respectively. N = 5 were used to investigate the first and second bath applications of TGOT in the presence of the GRK2 inhibitor. Finally, N = 6 mice were used to test the effect of KNCq blocker on the RTN response to TGOT.

- The experiments presented on Fig. 5D with the GRK2 blocker, their presentation in the text and their interpretation in the discussion, are very unclear to me, and problematic.

R: Thank you for your concern. We reformulated the sentence and the main objective in order to be clear. We also reformulated the entire interpretation of the data. For clarity, please see Lines 378-383; 541-567; 590-592 and 742-755.

First, lines 584-586, the main text states that "3 bouts of TGOT (200 mM)" applications were made, while the figure legend mentions only two applications (line 1197, "application of TGOT [...] before and after GRK2 blocker"). The methods section describing these experiments also mentions two TGOT applications. Please clarify.

R: We have made the necessary corrections. The text does not mention 3 bouts of TGOT; it refers to a single 3-minute exposure. See lines 551-567.

Second, lines 586-588, it is stated "Consistent with GRK2-dependent desensitization, inhibition of the TGOT-induced reduction in RTN neuronal activity after 3 min exposure ($t = 1.366$, $df = 12$, $p = 0.1971$) (Fig. 5C-D)". This sentence does not make sense to me in the way it is written. What it seems to state is that application of the GRK2 blocker would reduce the desensitization of the response to TGOT by RTN neurons. However, the results and statistics do not show that, since there is no difference ($p = 0.1971$) between the "TGOT" and "TGOT + CCG258208" groups (Fig. 5D). Unless I have missed something, this is a concerning over-statement and misinterpretation of the data, which results in an entirely incorrect discussion paragraph (lines 764-7750) and invalid conclusions.

R: In the new set of data interpretation and analysis, now Figure 5D showed that the first application of TGOT induced an increase in RTN neuronal activity of 5.0 ± 1.8 Hz. After the second TGOT application, the peak response reached a value of 3.4 ± 1.2 Hz (approximately 30% reduction). In Figure 5E, we now showed the ratio of the TGOT change (delta variation) between the second and first TGOT applications under both control conditions and in the presence of the GRK2 blocker. Although we still observed a significant reduction in RTN neuronal activity after TGOT application (before GRK2 inhibition: $t = 5.039$; $df = 8$; $p = 0.001$; after GRK2

inhibition: $t = 3.433$; $df = 4$; $p = 0.0265$) (Fig. 5D), there was no statistically significant between TGOT application and experimental group (vehicle + TGOT vs. CCG258208 + TGOT) ($F_{1,12} = 1.865$; $p = 0.1971$). Therefore, under our experimental conditions, we could not confirm a primary role for GRK2 in mediating TGOT-induced desensitization in RTN neurons. In neurons, desensitization of G protein-coupled receptors (GPCRs), including OXTR, is known to occur at the level of G proteins and often involves the recruitment of multiple GRK isoforms, including GRK2, GRK3, and GRK6. This finding is somewhat unexpected, as previous studies have suggested that desensitization of OXTR may rely predominantly on a single GRK isoform - GRK2 in HEK293 cells (PMID: 16179383) and GRK6 in uterine myometrium (PMID: 19423652). The apparent requirement for multiple GRK isoforms in neurons suggests that OXTR desensitization, and likely that of other neuronal GPCRs, involves more diverse and redundant regulatory mechanisms. See lines 742-755.

To minimize the potential for misinterpretation and provide a more robust analysis, we included the complete dataset and performed an ANOVA to properly evaluate the statistical significance of the results.

Third, while here a clearly non-significant result is interpreted as significant and is used to draw important conclusions (mechanism of desensitization of the response to TGOT), other instances in the text of very borderline "non-significant" effects in control conditions are easily dismissed. For instance, line 524 " $p = 0.064$ ", line 549 " $p = 0.076$ " and " $p = 0.052$ ". Statistics are not dichotomist, they represent a continuum, and the 0.05 confidence barrier is arbitrary and artificial. Strong pushes are made towards alternative representations that would better depict the distribution of data and estimations in their differences (cf. for instance PMID: 31217592, Ho J et al. 2019). While it is of course difficult to require an instant switch in the practices for data analysis and their representations, it is still necessary to avoid over-using the 0.05 barrier dichotomy.

For the line 524 " $p = 0.064$ " data, it is highly possible that the 9 mW light power used had a heating effect on RTN/pFRG neurons and induced a small increase in respiratory amplitude as shown by the P value of 0.064 (cf. PMID: 27895987, Shin Y et al. 2016, for accurate measures of the effects of different light powers in the brain ; also, by the way, the norm in the field of optogenetics for the representation of the light power used is the light power density in mW/mm^2 , to provide a more accurate representation of the brain surface affected, which is relative to the optical parameters of the equipment used to transmit the light into the brain).

The same applies to the results shown on line 549, with " $p = 0.076$ " and " $p = 0.052$ ", it is clearly difficult to state confidently that these neurons absolutely do not respond to CO₂ and to TGOT. Yet, the authors make it an important message of the article that only chemosensitive RTN neurons are activated by oxytocin.

R: We thank the reviewer for this important comment and fully agree that statistical results should not be interpreted in a strictly dichotomous manner Based solely on $p < 0.05$ or $p > 0.05$. We acknowledge that p-values represent a continuum of evidence for or against the hypothesis being tested, and that values slightly above 0.05 (e.g., $p = 0.064$ or $p = 0.076$) may still indicate potentially meaningful effects - particularly when they align with prior findings or are biologically plausible.

In line with this view, we chose to present the full distribution of individual data points, rather than relying solely on summary statistics. This approach

enhances transparency and allows readers to assess variability, and overall response patterns across subjects.

It is also important to note that, while p-values in the range of 0.064 or 0.076 make it difficult to definitively conclude that these neurons do not respond to CO₂ or TGOT, we observed a highly significant p-value ($p < 0.001$) when comparing the population of CO₂-sensitive neurons that did respond to TGOT (Figs. 4E–F).

We believe that presenting the data in this manner- alongside exact p-values and confidence intervals - provides a more nuanced and informative interpretation of our results

Regarding the result with $p = 0.064$, we agree that this may reflect a small non physiological effect of light stimulation. However, a closer examination of the data reveals only a minimal change in amplitude compared to baseline (raw data). We do not believe this effect was due to heating, as one would expect a reduced response to oxytocin stimulation at the RTN level if heating were a significant factor - given that RTN neurons are known to be sensitive to increased temperature (PMID: 22411009).

We acknowledge that reporting light power density (mW/mm²) provides a more standardized measure for comparing optogenetic stimulation parameters across studies. However, *in vivo* experiments, it is inherently challenging to determine the exact area of tissue illuminated due to several factors such as variability in tissue thickness, light scattering and absorption, and the precise positioning of the optic fiber, which can vary slightly between animals (for implantation variability). These limitations make it difficult to accurately estimate the effective light distribution *in vivo*. In contrast, such estimations are more feasible *in vitro* preparations or when using a ventral approach to stimulate the RTN, where the geometry and optical access to the target area are more controlled. Nonetheless, based on the nominal light power used (9 mW) and the 200 μ m diameter optic fiber, we estimate a power density at the fiber tip of approximately 286 mW/mm². However, we emphasize that the actual power density reaching the target neurons is likely substantially lower due to tissue scattering and absorption.

- In Fig. 3, experiments using OxtCre/+::Ai32 mice are presented. In my first review, I raised this point:

"The assessment of the selectivity of the ChR2 expression in oxytocinergic neurons in Oxt::Cre(+/-);Ai32 mice must be shown and quantified. This is an important control experiment, selectivity in double transgenic mice with Cre-lox recombination is never a given, yet it is a prerequisite for the validity of the experiments.

R: We thank the reviewer for raising this important point. In the previous revision, we had included data showing ChR2 expression in oxytocinergic terminals; however, it was placed incorrectly and poorly described. We fully agree that demonstrating and quantifying the selectivity of ChR2 expression in oxytocinergic neurons in the OxtCre/+::Ai32 mouse line is a critical control to validate our experimental approach. We have now provided a representative confocal images in Figure 3J, demonstrating the presence of ChR2-labeled axons colocalized with Oxt immunoreactivity in the target area.

In addition, we revised the Methods section to include a detailed description of the antibodies used and the criteria for image analysis.

We hope these updates address the reviewer's concern and strengthen the validity of our optogenetic strategy.

Also, it would be important to show an image of the ChR2 expression in oxytocinergic fibers in the RTN/pFRG in these mice."

R: See Fig. 3J.

The authors responded: "We appreciate the reviewer's suggestion regarding the assessment of ChR2 selectivity in oxytocinergic neurons in OxtCre^{+/+}::Ai32 mice. Indeed, Cre-lox recombination does not always guarantee exclusive expression in the target population, making validation essential. To address this, we have now quantified the colocalization of ChR2 (via YFP/mCherry fluorescence) with oxytocin (OXT) immunoreactivity in the PVN, providing an estimate of selectivity. Additionally, we provide representative confocal images showing ChR2 expression in oxytocinergic fibers within the RTN/pFRG to confirm the presence of projections in the targeted region. These data have been incorporated into the revised manuscript."

Yet, the current manuscript does not contain any of the necessary quantification and confocal images regarding ChR2 expression in the OxtCre^{+/+}::Ai32 mice that the authors stated that they included in their rebuttal.

R: We did not quantify; however, we added a confocal image showing the colocalization of ChR2 (GFP - green) with oxytocin (Alexa 647 - red). See Fig. 3J.

Minor points:

- Fig. 1A, please include the mention "double-floxed" in the name of the virus provided on the figure, otherwise one cannot understand how there will be specificity in the expression of this virus in OxtCre^{+/+} mice.

R: Done

Same comment for line 449, where "AAV-hChR2-mCherry" is written, the mention "double-floxed" should be added.

R: Done

- Fig. 2A, on the coronal view at the level of the PVN, the mention "Ctb⁺/Oxt⁺/VGlut2⁺" is made, however this mention is misleading since (i) it is missing the VGat aspect which is part of this experiment as depicted just above on the same schematic with the names of the transgenic mice used, and (ii) the experimental strategy depicted using Ctb tracing in transgenic mice does not enable the Oxt labelling presented. I suggest removing the mention "Ctb⁺/Oxt⁺/VGlut2⁺", or making it much clearer.

R: We remove the mention Ctb⁺/Oxt⁺/VGlut2⁺

- Legend of Fig. 2B, only the photomicrographs are mentioned, please also mention the schematics below the photomicrographs and state if all injections from all mice were reported.

R: We added the proper information about the computer-assisted plot.

- Fig. 2D, the legend states that "Ctb injection into the RTN/pFRG labeled neurons that are immunoreactive for VGat (green) and oxytocin (magenta) in the PVN region", while Fig. 2D shows a VGat-negative neuron. This is inconsistent.

R: Agreed. We modified the legend of the figure. Now it is reading: "Ctb injection into the RTN/pFRG labeled neurons in the PVN that are immunoreactive for oxytocin (magenta) but not for VGat (green)".

- The authors have stated in their previous rebuttal that all sections for histology were cut at 40 μm , however there are still several mentions of 30 μm sections in the text (e.g. lines 451 and 473).

R: We revised the entire manuscript and made the proper corrections.

- A previous point that was not taken into account, although the authors had answered in their rebuttal that it was done: Fourth paragraph of the results section, "G protein-coupled receptor kinases [...] desensitization", please provide a reference supporting this statement (lines 581-583).

R: Done. We added the following reference (PMID: 38883814)

- Line 202, "oxytocin expression" should probably rather be "oxytocin receptor expression in the RTN/pFRG".

R: Done

- Line 332, a polyclonal chicken anti-GFP primary antibody is stated to have been used, however no secondary anti-chicken antibody is mentioned further down in the text.

R: We added the secondary anti-chicken antibody We used the donkey anti-chicken Alexa 488 (703-545-155; Jackson Immuno Research Laboratories; dilution 1:400).

- Line 501, "oxytocin transcripts" should probably rather be "oxytocin receptor transcripts".

R: Done

- Line 546, "Figures 4C and 6A" should probably rather be "Figures 4C and 5A".

R: Yes, the reviewer is right. We made the proper change

- Line 1110, "oxytocinergic and glutamatergic" should probably rather be "GABAergic and glutamatergic".

R: Done

- Lines 1134-1136, for clarity please add the mention that the GFP-positive neurons harvested were from RTN/pFRG neurons.

R: Done. Now is reading: "Cumulative probability distributions of the indicated transcripts (Oxtr, Tacr1, Htr2c, Gpr4) in GFP-positive neurons harvested from the RTN/pFRG of Phox2b::GFP mice, as determined by single cell RNA-Seq. Lines 1210-1212.

- There are still several typographic errors, missing words etc. throughout the text (e.g. "ceiling effect" and not "celling effect").

R: We carefully reviewed the manuscript and made the appropriate corrections to the English.

- There are several issues with references, with references cited in the text missing from the reference list (including one that I had notified in a previous round of reviews) and references dually present in the reference list as "a" and "b".

R: We corrected all the inconsistencies in the reference.

Dear Dr Moreira,

Re: JP-RP-2025-287845R3 "Oxytocinergic signaling in the respiratory parafacial region increases the activity of chemosensitive neurons and respiratory output" by Emmanuel V Araújo, Phelipe E Silva, Luiz M Oliveira, Yingtang Shi, Ana C Takakura, Daniel K Mulkey, and Thiago S Moreira

Thank you for submitting your manuscript to The Journal of Physiology. It has been assessed by a Reviewing Editor and by 1 expert referees and we are pleased to tell you that it is acceptable for publication following satisfactory revision.

REVISION CHECKLIST:

Please upload two versions of your manuscript text: one with all relevant changes highlighted and one clean version with no changes tracked. The manuscript file should include all tables and figure legends, but each figure/graph should be uploaded as separate, high-resolution files. The journal is now integrated with Wiley's Image Checking service. For further details, see: <https://www.wiley.com/en-us/network/publishing/research-publishing/trending-stories/upholding-image-integrity-wileys->

image-screening-service

We look forward to receiving your revised submission.

Yours sincerely,

Harold Schultz
Senior Editor
The Journal of Physiology

EDITOR COMMENTS

Reviewing Editor:

Comments to the Author:

I thank the authors for submitting a revised version of the manuscript that addresses the Referee's comments. The manuscript has improved in clarity and quality, but a final round of minor adjustments will still be needed.

Senior Editor:

Comments to the Author:

The submission of the second revision of your research article to the Journal of Physiology for consideration remains to raise concerns. The remaining concerns are more minor and should be easily corrected.

In addition to these points, the Discussion should discuss an explanation and limitation for the absence of characterization of ChR2 expression in OxtCre/+::Ai32 mice.

The Data Availability statement is not acceptable. It must state that the data that support the findings of the study are available from the corresponding author, [or other source], upon reasonable request.

The editors also warn the authors about the lack of careful construction, proofing and oversight of the original manuscript before submission to the Journal. The number of major oversights and errors in the manuscript raise serious concern about the rigour of the study. It has placed an onerous burden on the referees and editors to address these concerns, beyond that which should normally occur. It is highly advised that manuscripts be carefully analyzed for accuracy before submission to the journal in the future. A re-occurrence of a manuscript with major errors such as this will result in rejection.

REFeree COMMENTS

Referee #2:

The authors have adequately responded to most of my previous concerns. I remain perplexed and unconvinced by two points - absence of characterization of ChR2 expression in OxtCre/+::Ai32 mice and over-interpretation of borderline p values - but these might not remain as major points when considered relative the rest of the data presented in this study, which consistently point to similar mechanisms of action.

Minor points:

- Figure 1C and 1D: I guess that the same symbols in both panels are actually used to depict different cell groups. In Fig. 1C, red squares are used to depict "m-Cherry+" cells, which I guess include both Oxt-/m-Cherry+ and Oxt+/m-Cherry+ cells. In Fig. 1D, the same red squares are used to depict only Oxt-/m-Cherry+ cells. The same goes for the green circles and Oxt cells between Fig. 1C and 1D. If I am correct, this is very confusing and different symbols should be used between the panels. If I am incorrect, there is a major issue in the results presented.

- Line 111, probably "as well as" instead of "as well"?

- Line 168, "rhodopsin channel-2" is a very unusual term.

- Line 463, the sentence "By comparison, expression of Oxtr and Tacr1 is observed less than expression for the 5-HT1C receptor (Htr2c)" should probably be rephrased.

- Line 529, there is a misspelling in "AMPA/kainite receptors".

- Line 741, "We provide evidence of agonist-induced internalization of oxytocin receptors in mouse RTN neurons". This is incorrect, the evidence provided shows agonist-induced desensitization. The mechanism by which this desensitization occurs is supposed to be through receptor internalization, however no demonstration of this was performed.

- Line 780, "All data generated or analyzed during this study are included in this published article". There are at least two instances of "data not shown" in the article.

END OF COMMENTS

EDITOR COMMENTS

Reviewing Editor:

Comments to the Author:

I thank the authors for submitting a revised version of the manuscript that addresses the Referee's comments. The manuscript has improved in clarity and quality, but a final round of minor adjustments will still be needed.

R: We thank the Reviewer Editor for the positive feedback on our revised manuscript and for recognizing the improvements in clarity and quality. We appreciate the careful evaluation and will address all remaining points to ensure the manuscript meets the highest standard.

Senior Editor:

Comments to the Author:

The submission of the second revision of your research article to the Journal of Physiology for consideration remains to raise concerns. The remaining concerns are more minor and should be easily corrected.

R: We would like to thank the Senior Editor for highlighting the concerns raised by Reviewer #2. In the current version of the manuscript, we have addressed all these concerns, and we hope that the revised version is now clear both to the reviewer and to future readers.

In addition to these points, the Discussion should discuss an explanation and limitation for the absence of characterization of ChR2 expression in OxtCre/+::Ai32 mice.

R: We added an explanation and limitation of the use of Oxtcre/+::Ai32 mice. See lines 611-629.

We added the following statement in the discussion section: "We also performed optogenetic experiments to selectively stimulate oxytocinergic terminal-like endings and assess the contribution of oxytocin to breathing activity in anesthetized adult mice. For this, we used the OxtCre/+::Ai32 transgenic line. Before discussing the data, it is important to address a key methodological consideration by verifying and quantifying ChR2 expression specifically in oxytocinergic neurons as a critical control. In our study, we provide a representative confocal image (Fig. 3J) showing ChR2-labeled axons (GFP immunohistochemistry) colocalized with Oxt immunoreactivity in the RTN region. Although we did not quantify the total number of labeled neurons or terminals, we are confident that ChR2 expression in this line is largely restricted to oxytocin-producing neurons and their projections. This transgenic line has been extensively characterized for robust Cre-dependent ChR2 expression, with the Ai32 construct enabling strong light-evoked neuronal spiking at low light intensities (Madisen et al., 2012; Zhang et al., 2024; Li et al., 2024). Nonetheless, recent studies have reported potential "leaky" expression (Cre-independent ChR2-EYFP) in Ai32 mice, which could lead to off-target activation in some brain regions (Prabhakar et al., 2019). As shown in Figure 3J, our histological data confirms ChR2 expression in oxytocinergic projections to the RTN. Future studies combining functional recordings and histological verification in the same animals will further ensure specificity and strengthen causal interpretations of behavioral and physiological effects".

The Data Availability statement is not acceptable. It must state that the data that support the findings of the study are available from the corresponding author, [or other source], upon reasonable request.

R: Done

REFeree COMMENTS

Referee #2:

The authors have adequately responded to most of my previous concerns. I remain perplexed and unconvinced by two points - absence of characterization of ChR2 expression in *OxtCre/+::Ai32* mice and over-interpretation of borderline p values - but these might not remain as major points when considered relative the rest of the data presented in this study, which consistently point to similar mechanisms of action.

R: We fully agree that demonstrating and quantifying the selectivity of ChR2 expression in oxytocinergic neurons in the *OxtCre/+::Ai32* mouse line is a critical control to validate the experimental approach. In a previous version, we have provided a representative confocal image in Figure 3J, demonstrating the presence of ChR2-labeled axons (GFP immunohistochemistry) colocalized with *Oxt* immunoreactivity in the target area (RTN region). Although we did not quantify the total number of labeled neurons and terminals, we are confident that ChR2 is selectively expressed in oxytocin-producing neurons and their terminals in the *OxtCre/+::Ai32* line.

Important to point out that this transgenic line has been used and characterized in the literature for robust cre-dependent Channelrhodopsin-2 (ChR2) expression (PMID: 22446880, 38093006, 38537642). For instance, the *Ai32* line utilizing a *Rosa26* CAG::*ChR2-EYFP* construct has been shown to enable strong light-evoked neuronal spiking with low-intensity stimulation in various neuronal populations, demonstrating its effectiveness and reliability in heterogeneous systems (PMID: 22446880, 38093006, 38537642). However, recent reports have cautioned about potential “leaky” expression (i.e., Cre-independent ChR2-EYFP expression) in *Ai32* mice, which could introduce off-target activation in some anatomical regions (PMID: 30913225). As mentioned above, we provided a representative confocal image in Figure 3J demonstrating the presence of ChR2-labeled axons colocalized with oxytocin immunoreactivity in the target RTN region. Although we did not quantify the total number of labeled neurons and terminals, we are confident that ChR2 is selectively expressed in oxytocin-producing neurons and their terminals in the *OxtCre/+::Ai32* mouse line. Moving forward, combining functional recordings with histological verification in the same animals would strengthen the link between optogenetic manipulation and oxytocin neuron activity in our experimental context, ensuring specificity and reinforcing confidence in causal interpretations of behavioral or physiological outcomes. We added an explanation and limitation for the characterization of ChR2 expression in *OxtCre/+::Ai32* mice. See lines 611-629.

We also appreciate the reviewer’s comment about the p-value. However, we have no further comments, as we agree that while some p-values are borderline, the overall data consistently supports the proposed mechanisms of action.

Minor points:

- Figure 1C and 1D: I guess that the same symbols in both panels are actually used to depict different cell groups. In Fig. 1C, red squares are used to depict "m-Cherry+" cells, which I guess include both Oxt-/m-Cherry+ and Oxt+/m-Cherry+ cells. In Fig. 1D, the same red squares are used to depict only Oxt-/m-Cherry+ cells. The same goes for the green circles and Oxt cells between Fig. 1C and 1D. If I am correct, this is very confusing, and different symbols should be used between the panels. If I am incorrect, there is a major issue in the results presented.

R: The reviewer is right. We have now changed the symbols of panel C.

- Line 111, probably "as well as" instead of "as well"?

R: Yes, the reviewer is correct

- Line 168, "rhodopsin channel-2" is a very unusual term.

R: We made the proper correction

- Line 463, the sentence "By comparison, expression of Oxtr and Tacr1 is observed less than expression for the 5-HT1C receptor (Htr2c)" should probably be rephrased.

R: Done. Now it is reading: "In comparison, both Oxtr and Tacr1 were expressed at lower levels than the 5-HT2C receptor (Htr2c), which largely mediates serotonin effects on RTN neurons, and at much lower levels than Gpr4, the most highly expressed GPCR and a key contributor to the intrinsic CO₂/H⁺ sensitivity of RTN neurons."

- Line 529, there is a misspelling in "AMPA/kainite receptors".

R: Done

- Line 741, "We provide evidence of agonist-induced internalization of oxytocin receptors in mouse RTN neurons". This is incorrect, the evidence provided shows agonist-induced desensitization. The mechanism by which this desensitization occurs is supposed to be through receptor internalization, however no demonstration of this was performed.

R: We changed the sentence as it was leading to a misinterpretation. In addition, we also changed the heading of the discussion section. Please see lines 739-741.

- Line 780, "All data generated or analyzed during this study are included in this published article". There are at least two instances of "data not shown" in the article.

R: We added the following sentence: "The data that support the findings of the study are available from the corresponding author upon reasonable request".

Dear Professor Moreira,

Re: JP-RP-2025-287845R4 "Oxytocinergic signaling in the respiratory parafacial region increases the activity of chemosensitive neurons and respiratory output" by Emmanuel V Araújo, Phelipe E Silva, Luiz M Oliveira, Yingtang Shi, Ana C Takakura, Daniel K Mulkey, and Thiago S Moreira

We are pleased to tell you that your paper has been accepted for publication in The Journal of Physiology.

Yours sincerely,

Harold Schultz
Senior Editor
The Journal of Physiology

If you would like to receive our 'Research Roundup', a monthly newsletter highlighting the cutting-edge research published in The Physiological Society's family of journals (The Journal of Physiology, Experimental Physiology, Physiological Reports, The Journal of Nutritional Physiology and The Journal of Precision Medicine: Health and Disease), please click this link, fill in your name and email address and select 'Research Roundup':
<https://www.physoc.org/journals-and-media/membernews>

- You can help your research get the attention it deserves! Check out Wiley's free Promotion Guide for best-practice recommendations for promoting your work at: www.wileyauthors.com/eeo/guide. You can learn more about Wiley Editing Services which offers professional video, design, and writing services to create shareable video abstracts, infographics, conference posters, lay summaries, and research news stories for your research at: www.wileyauthors.com/eeo/promotion.

EDITOR COMMENTS

Reviewing Editor:

Comments to the Author:

I thank the authors for addressing the remaining concerns about the manuscript. I have no further comments and

congratulate the authors on the study.